# Character, Rates, and Environmental Significance of Holocene Dust Accumulation in Archaeological Hilltop Ruins in the Southern Levant

**Bernhard Lucke [1,*], Joel Roskin [2], Kim André Vanselow [1], Hendrik J. Bruins [3],
Nizar Abu-Jaber [4], Katleen Deckers [5], Susanne Lindauer [6], Naomi Porat [7], Paula J. Reimer [8],
Rupert Bäumler [1], Tali Erickson-Gini [9] and Paula Kouki [10]**

1 Institute of Geography, Friedrich-Alexander-Universität Erlangen-Nürnberg, Wetterkreuz 15,
   91058 Erlangen, Germany; kim.vanselow@fau.de (K.A.V.); rupert.baeumler@fau.de (R.B.)
2 Geomorphology and Portable Luminescence Laboratory, Leon Recanati Institute for Maritime Studies,
   University of Haifa, Haifa 31905, Israel; yoelr@bgu.ac.il
3 Jacob Blaustein Institutes for Desert Research & Department of Geography and Environmental
   Development, Ben-Gurion University of the Negev, Beer-Sheva 8410501, Israel; hjbruins@bgu.ac.il
4 School of Natural Resources Engineering and Management, German Jordanian University, Madaba,
   Amman 11180, Jordan; nizar.abujaber@gju.edu.jo
5 Institute of Archaeological Sciences, Eberhard Karls University Tübingen, 72070 Tübingen, Germany;
   katleen.deckers@uni-tuebingen.de
6 Curt-Engelhorn-Zentrum Archäometrie gGmbH, 68159 Mannheim, Germany; susanne.lindauer@ceza.de
7 Geological Survey of Israel, Jerusalem 9692100, Israel; naomi.porat@gsi.gov.il
8 14Chrono Centre, Queen's University Belfast, Belfast BT9 6AX, UK; p.j.reimer@qub.ac.uk
9 Israel Antiquities Authority, Omer 8496500, Israel; talixgini@gmail.com
10 Ancient Near Eastern Empires Centre of Excellence, P.O. Box 3, University of Helsinki, 00014 Helsinki,
   Finland; paula.kouki@hamina.fi
* Correspondence: bernhard.lucke@fau.de; Tel.: +49-9131-85-23305

**Abstract:** Loess accumulated in the Negev desert during the Pleistocene and primary and secondary loess remains cover large parts of the landscape. Holocene loess deposits are however absent. This could be due low accumulation rates, lack of preservation, and higher erosion rates in comparison to the Pleistocene. This study hypothesized that archaeological ruins preserve Holocene dust. We studied soils developed on archaeological hilltop ruins in the Negev and the Petra region and compared them with local soils, paleosols, geological outcrops, and current dust. Seven statistically modeled grain size end-members were identified and demonstrate that the ruin soils in both regions consist of mixtures of local and remote sediment sources that differ from dust compositions deposited during current storms. This discrepancy is attributed to fixation processes connected with sediment-fixing agents such as vegetation, biocrusts, and/or clast pavements associated with vesicular layers. Average dust accretion rates in the ruins are estimated to be ~0.14 mm/a, suggesting that ~30% of the current dust that can be trapped with dry marble dust collectors has been stored in the ruin soils. Deposition amounts and grain sizes do not significantly correlate with wind intensity. However, precipitation may have contributed to dust accretion. A snowstorm in the Petra region delivered a significantly higher amount of sediment than rain or dry deposition. Snowfall dust had a unique particle size distribution relatively similar to the ruin soils. Wet deposition and snow might catalyze dust deposition and enhance fixation by fostering vegetation and crust formation. More frequent snowfall during the Pleistocene may have been an important mechanism of primary loess deposition in the southern Levant.

**Keywords:** loess; Holocene; ruin soil; archaeological sediment; vesicular layer; aeolian dust; biocrusts; clast pavements; climate change; snow

## 1. Introduction

Widespread late Quaternary loess deposits can be found in the central and northern Negev desert. These testify to Pleistocene landscape changes, documented by variations of dust deposition, accumulation, and soil development [1–12]. Loess sediments also played a role for human activities during antiquity in the arid and semi-arid southern Levant. Specific adapted land use strategies, such as terraced runoff farming, harvested rainwater and sediments, enabling flourishing ancient cities like Avdat and Petra [13,14]. Loess in southern Jordan has been postulated [15,16], but has hardly been investigated. However, much less is known about dust settling and landscape changes during the Holocene, including how human activity interacted with climate fluctuations. This is partly due to a lack of Holocene sedimentary sequences. For example, Avni et al. [17] showed how natural incision of the Negev loess deposits during the Holocene led to landscape changes, which were slowed down by ancient agricultural terraces, as these controlled and reduced runoff.

It has been suggested that Holocene dust deposits are less common in the southern Levant due to more concentrated discharges of pronounced rainfalls, which lead to erosion rather than accumulation [17]. It has also been proposed that the Pleistocene was a "dustier" period than the Holocene, with a respective higher supply of sediment [18,19]. Faershtein, G. et al. [20] elaborated that not rainfall variations but reduced amounts of dust may have led to more concentrated runoff and, thus, stronger discharges from the turn to the Holocene onwards. In this context, it was found that dust settling during the wet season is approximately twice the amount than that deposited during the dry seasons [21]. This is due to the six- to seven-fold occurrence of strong winds, which are linked with Eastern Mediterranean cyclones. However, the amount of dust washed out by raindrops is small and probably subordinate to the role of the relief in particular wind-sheltered areas in the lee [21].

It has been proposed that stronger winds during the Middle and mainly the Late Pleistocene (with its comparatively longer time frame than the Holocene) could have led to silt-sized particle production from aeolian abrasion of sand grains of dunes. Weakening wind strengths at the turn to the Holocene caused a reduction of total dust supply in the Negev [11,22]. However, it seems that processes other than aeolian abrasion governed the Holocene dust supply and silt deposition may have been a result of medium-range transport [23]. From wind tunnel experiments, Swet, N. et al. [24] reported that strong winds led to an erosion of clay coatings of sand particles and the mobilization of silt stored in-between sand grains, but less to the abrasion of quartz grain sand to coarse silt.

It is thus possible that landscape changes during the Holocene were less related to rainfall or base level changes, but more to variations in the amount of settling aeolian dust. The geomorphic response to the diminishing dust supply during the Holocene could have been delayed due to clogging of the drainage system by re-deposited Pleistocene dust [20]. Therefore, a focused study on Holocene dust records is anticipated to improve our understanding of the connection between dust supply and landscape change in the near past. Such a study is crucial for understanding current and future dust deposition character and its impact on man.

Today, dust storms occur in the Levant several times a year, albeit varying in number and intensity. The Negev has the highest dust concentrations and deposition rates in Israel, during winter storms in particular [25,26]. The mineralogy of the current dust is similar to the Pleistocene loess [11]. As climate changes were probably less than during the Pleistocene and the transition to the Holocene [27], dust deposition could have remained comparatively constant during the Holocene. Holocene dust may have been preserved in locations that are protected against aeolian and fluvial erosion, such as within archaeological ruins, which are usually covered by "debris". The latter is a mixture of rubbish, broken architecture, and fines likely representing aeolian dust [9,28–30]. The ruins could be excellent

dust traps due to their rough surface [9] and wall remains that slow down winds and trigger lee deposition [21] and protection from various types of flow.

It has already been proposed that sediments in geoarchaeological archives may include a major dust component [31]. Holocene dust deposition at Tell Brak in Syria probably took place more or less continuously, at least during the third millennium BCE [32]. However, soils formed on sediments covering ancient ruins may include significant amounts of archaeological substrate, such as the remains of mudbricks [33]. Local and remote dust sources may have mixed. Disentangling these possible sources is thus a main challenge that needs to be met in order to interpret the sediments in archaeological ruins as environmental archives.

In order to tackle dust deposition in the southern Levant during the Holocene, we studied sediments in several archaeological ruins in the northern Negev, in Israel, and in the vicinity of Petra, in southern Jordan. For Jordan, this is a pioneering study since soils derived from aeolian sediments have been postulated, but so far have not been studied. We focused on hilltop areas where fluvial re-deposition was highly unlikely. The different lithological and geomorphological conditions of the study areas allow assessing the role of local sediment sources and regional depositional patterns. In order to compare geoarchaeological sediments with current aeolian deposition, dry dust collectors were placed near the study sites and dust was collected after storms. As well, paleosols in both study areas were investigated in order to compare Pleistocene deposits with the sediments in the ruins.

Our work hypotheses are the following:

(1)　The debris inside (hilltop) ruins contains a major dust component;
(2)　Local and remote sediment sources can be differentiated;
(3)　Sediments in archaeological ruins represent the missing Holocene dust archives;
(4)　Understanding deposition processes in the ruins in the context of current dust dynamics will improve our knowledge of past landscape changes;
(5)　Ruin soils might represent a hitherto unexplored climate archive.

## 2. Study Areas and Methods

### 2.1. The Negev and Petra Areas

The study sites in the northern Negev are located at Horvat Haluquim, near Kibbutz Sede Boker and the adjacent Sede Boker campus of the J. Blaustein Institute for Desert Research (Ben-Gurion University of the Negev). Many earlier geoarchaeological investigations on this area have been published, mainly on ancient runoff-harvesting and the respective terrace systems [34–39]. In Jordan, the vicinity of the mountain Jabal Haroun near Petra (site of the pilgrimage sanctuary of Aaron/Haroun) was investigated. This area has been investigated by the archeological Finnish Jabal Haroun Project (FJHP) [40–46] (Figure 1).

The Horvat Haluqim area in the Negev is mainly built of massive to well-bedded shallow marine Turonian limestone, which includes a continental clastic unit of sandstones and paleosols from that epoch [47]. Patches of Pleistocene colluvial-aeolian aprons with loessial paleosols were preserved in some areas, which suggests that the area was once covered by extensive loess blankets [8,20]). In contrast, the region around Jabal Haroun is dominated by Cambrian continental sandstones of much more pronounced relief than in the Negev [48]. Its elevation is 900–1200 m, approximately 500 m higher than the Negev. Horst structures related to the Dead Sea transform fault form a highly diverse geology, including patches of limestones and igneous rocks [48]. Soils and sediments in the region have a significant sand fraction, derived from local fans and eroded sandstones [49]. Within archaeological structures, however, calcareous sediments have a significant silt fraction, which possibly represents long-range dust transport during the Holocene [50].

The climate of both study regions is arid and corresponds to the BWh classification of the Köppen–Geiger system [51]. Rains occur mostly during November to March. At Sede Boker, the mean annual rainfall is 93 mm (average for 1990–2000), but variations are high, as follows: A total of 188 mm,

in a wet season like 1991/92, or 34 mm, in a dry season like 1998/99. Bruins, H.J. [36] determined the current average P/PET-ratio at Sede Boker, which is a more relevant figure for agriculture than precipitation, to 0.07. In Petra and Jabal Haroun in Jordan, mean annual rainfall is 153 mm (Wadi Musa weather station, 1984–2011), with similar high variations, as follows: A total of 274 mm in the wettest season 1987/88, during the above-mentioned period, or 42 mm in the driest season, from 2010–2011.

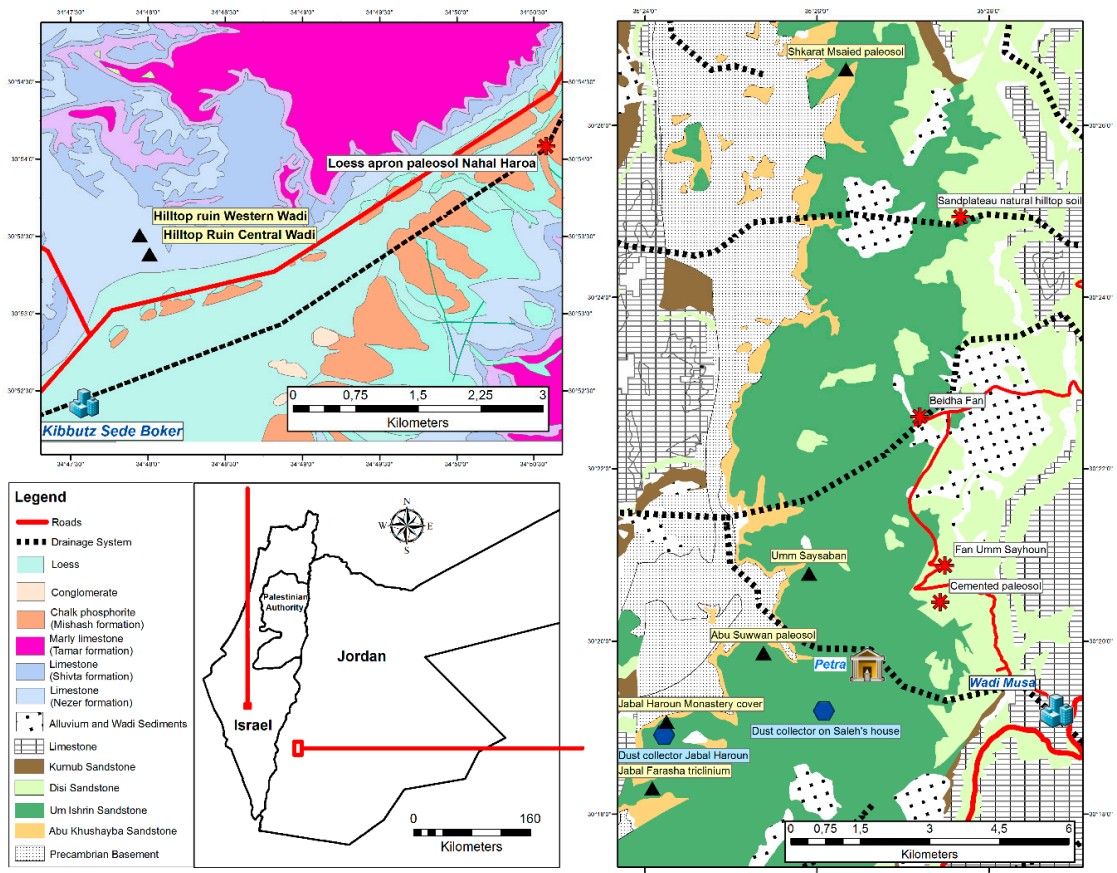

**Figure 1.** Map showing the locations of the investigation areas in Israel and Jordan and the main geological units. Black triangles mark the studied archaeological hilltop soils, orange stars mark the sampled reference sites, and blue hexagons mark the dust collectors in the Petra region. The dust collector in the Negev is located c. 4 km to the south from the sampling sites in Midreshet Ben Gurion. Rock outcrops and reference samples not marked on the map were located close to the sampled hilltop ruins.

### 2.1.1. Sampled Profiles

Two hilltop ruin soils were sampled at Horvat Haluqim in the Negev, along with two samples of paleosols preserved in a nearby loessial apron at Nahal HaRo'a. Seven reference samples that could comprise local sediment sources were collected from geological outcrops in the vicinity. At Jabal Haroun, near Petra, three soils covering hilltop ruins were sampled, along with two hilltop paleosols that were preserved below ruins, and a modern hilltop soil profile on a sandstone plateau. Eight samples were collected from rock and sediment outcrops that could comprise local sediment sources (Table 1). Figures 2 and 3 show views of the general landscapes of the investigation regions and Figures 4–6 show the sampled profiles.

**Table 1.** List of sampled soil profiles and of samples from outcrops as references of potential local sediment sources. For detailed descriptions of the sampled profiles, see Appendix A.

| Site Name | Coordinates | Number of Samples | Description |
|---|---|---|---|
| **Negev: Horvat Haluqim** | | | |
| HH-WW-R1 | N 30.89151 E 34.79909 | 2 | Soil covering a hilltop tumulus ruin near the western wadi of Horvat Haluqim |
| HH-CW-Ruin | N 30.88948 E 34.80015 | 6 | Soil covering a circular hilltop ruin, with paleosol below, near the central wadi of Horvat Haluqim |
| NH-LA | N 30.30140 E 34.84296 | 2 | Pleistocene loessial paleosol of a colluvial-aeolian apron in Nahal HaRo'a |
| Reference samples | See Appendix A for coordinates | 5 | Turonian paleosol, and various rock outcrops at Horvat Haluqim |
| **Petra Region: Jabal Haroun** | | | |
| Jabal Haroun monastery | N 30.31734 E 35.40418 | 2 | Soil covering the ruins of a monastery on a sandstone plateau at Jabal Haroun |
| Jabal Farasha triclinium | N 30.30445 E 35.40141 | 2 | Soil covering the ruins of a triclinium on a hilltop southwest of Jabal Haroun |
| Umm Saysaban | N 30.34595 E 35.43178 | 2 | Soil covering the ruins of the hilltop site of Umm Saysaban north of Jabal Haroun |
| Abu Suwwan | N 30.33064 E 35.42297 | 1 | Paleosol preserved below ruin of Neolithic hilltop site, north-east of Jabal Haroun |
| Shkarat Msaied | N 30.44372 E 35.43917 | 1 | Paleosol preserved below ruin of Neolithic hilltop site, north of Jabal Haroun |
| Sandplateau | N 30.41564 E 35.46117 | 5 | Natural, currently forming hilltop soil on sandstone plateau, north-east of Jabal Haroun |
| Reference samples | See Appendix A for coordinates | 8 | Various rock outcrops, fans, and a cemented paleosol in the Petra region |

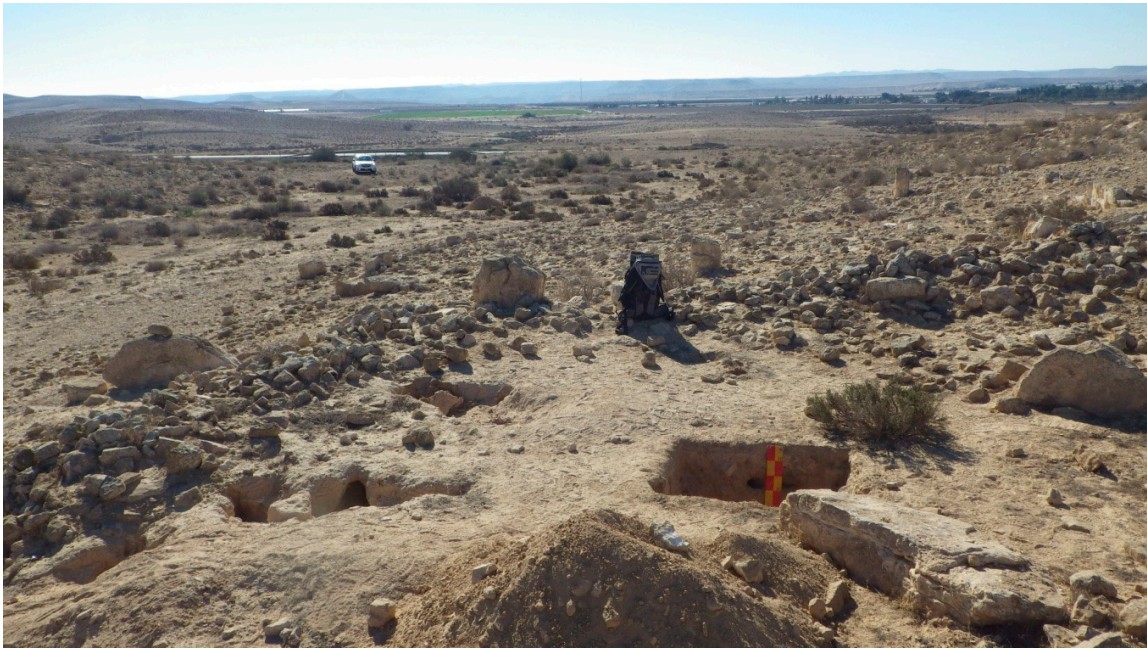

**Figure 2.** View south from the sampled ruin near the central wadi at Horvat Haluquim, showing the surrounding landscape in the Negev.

We studied soils covering ruins, with parent materials probably largely derived from aeolian sediments. The term "ruin soils" is thus meant to include sediments that gathered in the ruins. Soil types were classified according to the World Reference Base of Soil Resources WRB [52] (see Tables 2 and 3). Detailed descriptions of the profiles and reference sampling sites are provided in Appendix A. Table 1 presents the studied sites in a compressed form.

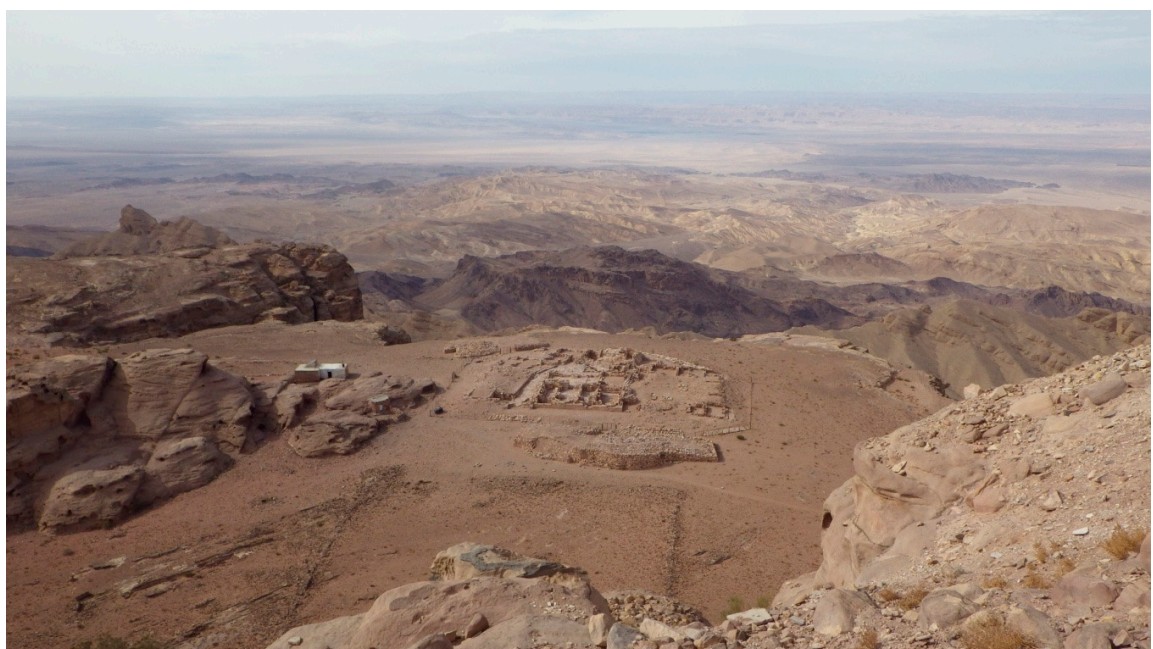

**Figure 3.** View from the top of Jabal Haroun to the south-west. The excavated ruins of the monastery of Jabal Haroun are visible in the center of the image and the Wadi Araba/Arava valley is in the background.

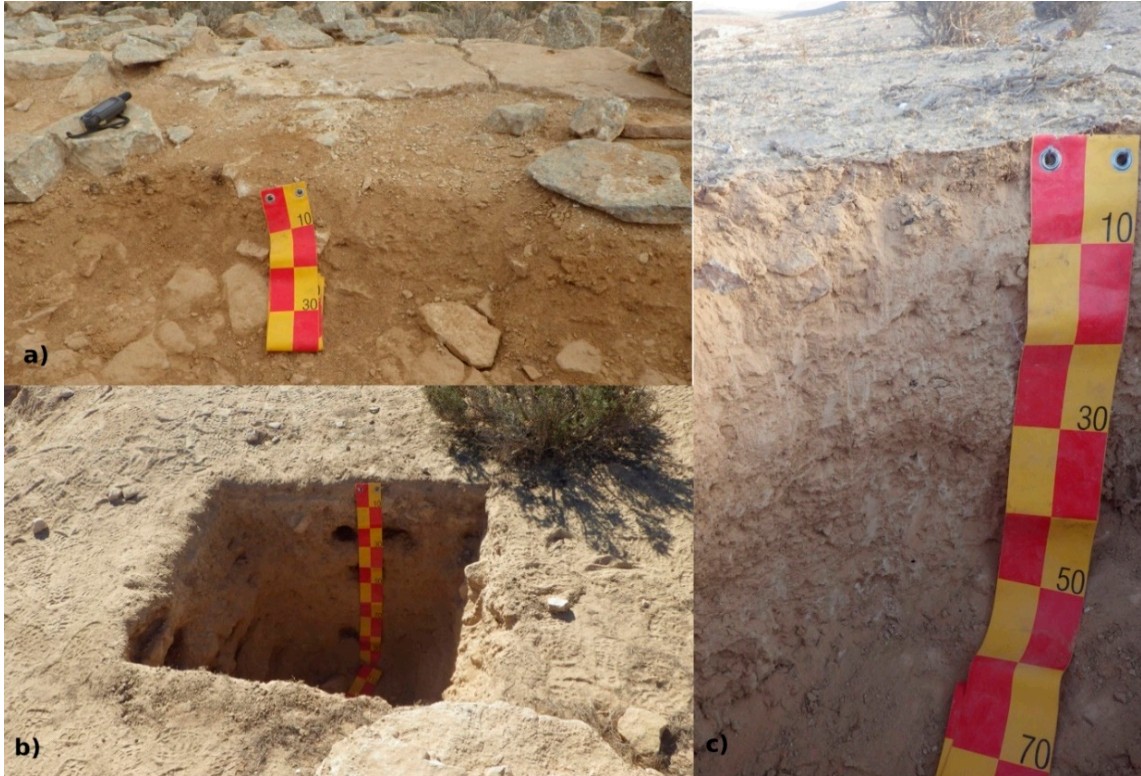

**Figure 4.** Profiles sampled at Horvat Haluqim in the Negev. (**a**) The hilltop ruin near the western wadi, (**b**) the ruin on the loess spur projecting into the central wadi, and (**c**) the paleosol of the colluvial-aeolian loessial apron in Nahal HaRo'a. The tape measure is in cm. For a detailed description of the profiles, see Appendix A.

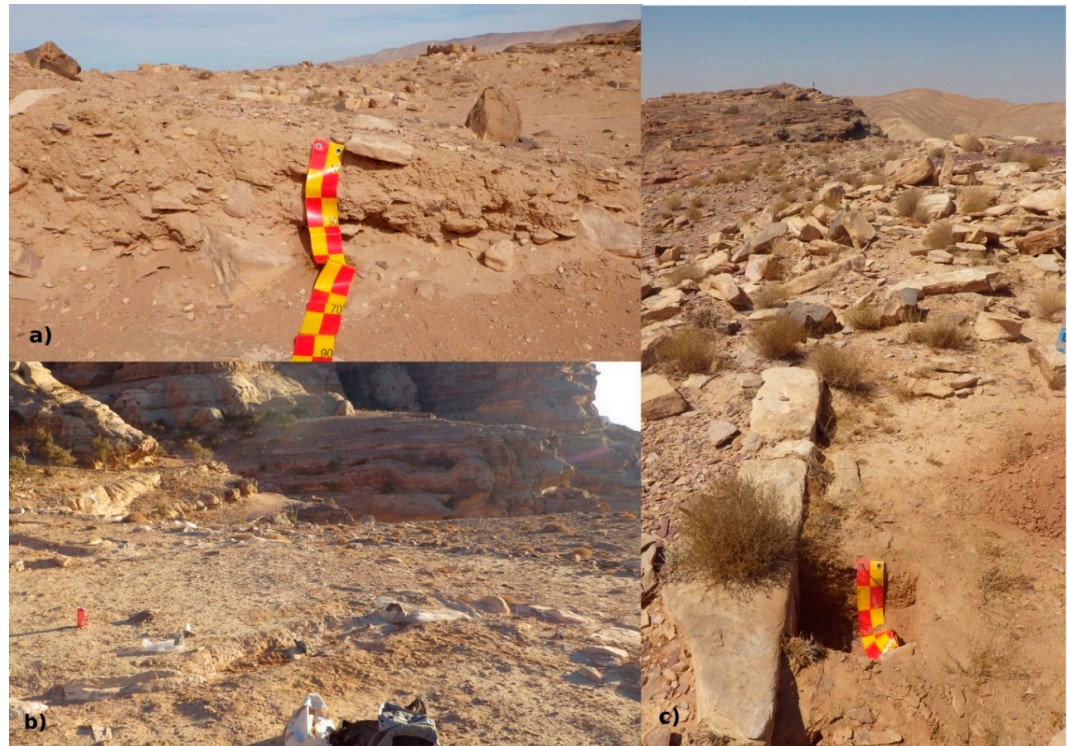

**Figure 5.** Studied profiles in the Petra region. (**a**) Soil cover on the debris in the northern courtyard of the monastery of Jabal Haroun, (**b**) Umm Saysaban (GPS and backpack give scale), and (**c**) the triclinium of Jabal Farasha. The tape measure is in cm. For a detailed description of the profiles, see Appendix A.

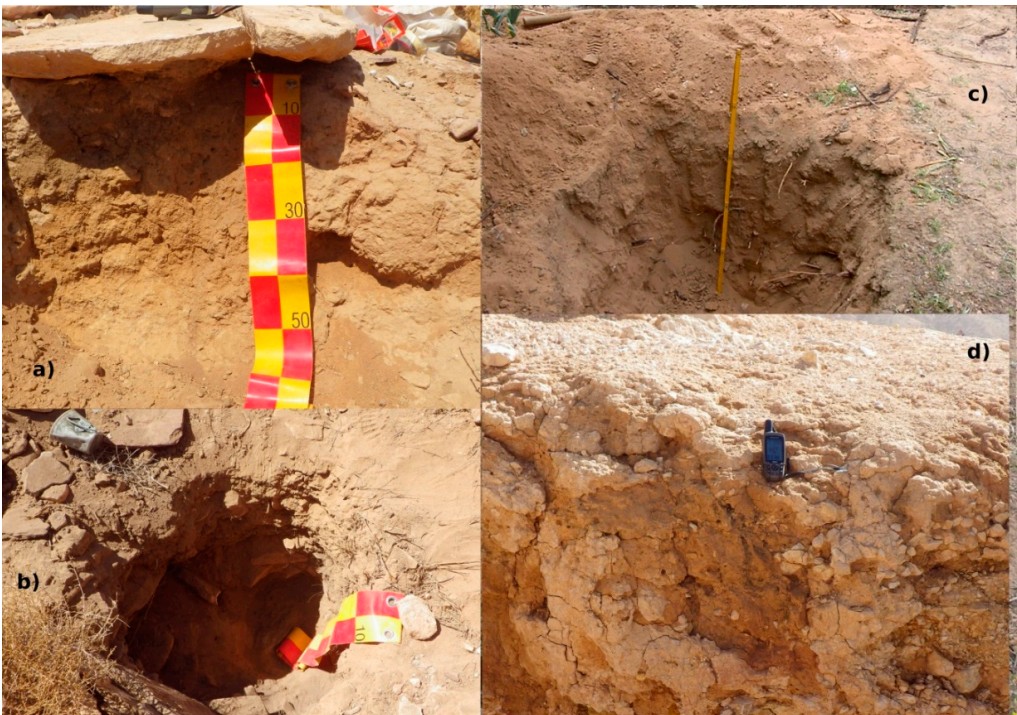

**Figure 6.** Profiles sampled in the Petra region. (**a**) The paleosol below the ruins of Shkarat Msaeid, (**b**) the sondage excavated in the ruins of Abu Suwwan, (**c**) the natural hilltop soil on a sandstone plateau (the meter represents 1 m), and (**d**) the cemented paleosol (GPS gives scale). The tape measure is in cm. For a detailed description of the profiles, see Appendix A.

**Table 2.** General properties and calibrated ages of the Negev profiles and samples.

| Sample No. | Horizon from ... to ... (cm) | Sampling Depth (cm) | Calibrated $^{14}$C Age (1-Sigma) | Cal. OSL Age | Age from Context | Cal. TL Age of Pottery | Munsell Dry | pH (H$_2$O) | Conductivity (µS/cm) | $C_{org}$ % | CaCO$_3$ [1] % |
|---|---|---|---|---|---|---|---|---|---|---|---|
| **Ruin Overlooking the Western Wadi: Calcaric Leptosol (Protic, Siltic)** | | | | | | | | | | | |
| HH-WW R1 10 | C (5–20) | 10 | | | | | 10YR 6/6 brownish yellow | 8.3 | 2300 | 1.10 | 37 |
| HH-WW-R1 20 | C (5–20) | 20 | | | 2000–2500 BCE | 970–470 BCE (15 cm) | 10YR 5/6 yellowish brown | 8.3 | 3890 | 1.30 | 38 |
| **Ruin Overlooking the Central Wadi: Protic Regosol (Siltic)** | | | | | | | | | | | |
| HH-CW-Ruin 10 | V1 (5–15) | 10 | ~ 1946 [2] CE (7 cm) 1643–1918 CE (14 cm) | | | | 10YR 7/3 very pale brown | 8.7 | 371 | 0.90 | 40 |
| HH-CW-Ruin 25 | V2 (15–35) | 25 | 1694–1918 CE (20 cm) | 870–1020 CE | | | 10YR 7/3 very pale brown | 8.4 | 2100 | 0.84 | 42 |
| HH-CW-Ruin 50 | C (40–70) | 50 | 6210–6092 BCE (charred temper) | 2570–1990 BCE | | 830–300 BCE | 10YR 7/3 very pale brown | 8.7 | 3450 | 0.76 | 40 |
| HH-CW-R-60 | C (40–70) | 60 | | | | | 10YR 7/6 yellow | 9.0 | 3620 | 1.20 | 34 |
| HH-CW-R 75 (Paleosol: buried Cambic Calcisol) | 2Bwk (70–100+) | 75 | | | | | 10 YR 7/4 very pale brown | 8.8 | 3320 | 0.30 | 36 |
| HH-CW-R 90 (Paleosol: buried Cambic Calcisol) | 2Bwk (70–100+) | 90 | | 23980–20580 BCE | | | 10YR 7/4 very pale brown | 9.0 | 3120 | 1.10 | 29 |
| **Paleosol of Loess Apron in Nahal Haroa: Cambic Calcisol (Siltic)** | | | | | | | | | | | |
| NH-LA-10cm | V (Loess apron cover) | 10 | | | | | 10YR 8/6 yellow | 8.9 | 111 | 0.43 | 28 |
| NH-LA-30cm (Paleosol: buried Cambic Calcisol) | 2Bwk (Loess apron paleosol) | 30 | | | | | 10YR 6/6 brownish yellow | 9.3 | 135 | 0.38 | 27 |
| **Reference Samples from Horvath Haluqium** | | | | | | | | | | | |
| HH-CW-Tur-Paleo | Turonian paleosol | outcrop | | | | | 2.5 Y 7/4 pale yellow | 7.6 | 1817 | | 44 |
| HH-CW-chalk | chalk | outcrop | | | | | 7.5YR 8/3 pink | | | | 98 |
| Haroa Farm chalk | chalk | outcrop | | | | | 5YR 8/2 pinkish white | | | | 98 |
| HH-WW-C2-soft limestone | soft limestone | outcrop | | | | | 5YR 8/1 white | | | | 96 |
| HH-WW-C2-hard limestone | hard limestone | outcrop | | | | | 5YR 8/1 white | | | | 96 |

[1] Average from three methods (see Appendix C for individual results). [2] Sample was outside calibration curve.

**Table 3.** General properties and calibrated ages of the ruin soils and samples from the vicinity of Petra.

| Sample No. | Horizon from … to … (cm) | Sampling Depth (cm) | Calibrated [14]C Age (1-Sigma) | Cal. OSL Age | Age from Context | Munsell Dry | pH (H$_2$O) | Conductivity (µS/cm) | C$_{org}$ % | CaCO$_3$ [1] % |
|---|---|---|---|---|---|---|---|---|---|---|
| \multicolumn Jabal Haroun Monastery Ruin: Protic Arenosol (Aridic, Aeolic) |||||||||||
| FJHP Site 1, Trench R | n.d. | 30 | | | | 7.5YR 6/3 light brown | 8.3 | 1296 | 0.33 | 7 |
| JH 1 | V (0–10) | 10 | | | ~1000 CE | 7.5YR 7/3 pink | 8.6 | 114 | 0.23 | 7 |
| \multicolumn Jabal Farasha Triclinium Ruin: Calcaric Leptosol (Protic) |||||||||||
| JF site 124/1 5 cm | V (0–10) | 5 | | | | 7.5YR 5/3 brown | 8.5 | 112 | 0.94 | 13 |
| JF site 124/1 15 cm | C (10–25) | 15 | | 870–1040 CE | | 7.5YR 5/4 brown | 8.6 | 86 | 0.53 | 10 |
| JF site 124/1 25 cm | C (10–25) | 25 | | | ~200 CE | 5YR 5/4 reddish brown | 8.5 | 79 | 0.52 | 8 |
| JF site 124/1 rock | R (25+) | bedrock | | | | 10R 5/4 weak red | n.a. | n.a. | 0.13 | 2 |
| \multicolumn Umm Saysaban Ruin: Calcaric Leptosol (Protic) |||||||||||
| Umm Saysaban 5 cm | V (0–5) | 5 | | | | 7.5YR 6/3 light brown | 8.5 | 106 | 0.58 | 12 |
| Umm Saysaban 10 cm | C (5–10) | 10 | | | ~2500 BCE | 10YR 7/3 very pale brown | 8.5 | 83 | 0.38 | 12 |
| \multicolumn Shkarat Msaied PPNB Ruins: Buried Chromic Cambisol (Protocalcic) |||||||||||
| Shkarat Msaeid 4 (cover above ruin) | 2C (70–110) | 90 | 12,385–12,131 BCE | | | | | | | |
| Shkarat Msaied 1 (paleosol below ruin) | 3BCk (45–60+) | 55 | | | >8000 BCE | 7.5 YR 7/3 pink | 8.3 | 1375 | 0.27 | 15 |
| \multicolumn Abu Suwwan PPNB Ruins: Buried Protic Regosol (Aeolic) |||||||||||
| Abu Suwwan ashy layer 30 cm | 2C (20–40) | 30 | 2014–1916 BCE | | | | | | | |
| Abu Suwwan below nw 65 (buried paleosol) | 3BC (60–70+) | 65 | | | >8000 BCE | 10 YR 5/2 grayish brown | 8.1 | 142 | 0.60 | 18 |
| \multicolumn Natural Hilltop Soil on Sandstone Plateau: Protic Regosol (Arenic) |||||||||||
| Sandplateau 1 | A (0–10) | 10 | | | | 10 YR 6/4 light yellowish brown | 8.4 | 66 | 0.28 | 2 |
| Sandplateau 2 | C (10–70) | 30 | | | | 10 YR 5/6 yellowish brown | 8.5 | 60 | 0.16 | 1 |
| Sandplateau 3 | C (20–70) | 50 | | | | 10 YR 6/6 brownish yellow | 8.2 | 60 | 0.12 | 0 |
| Sandplateau 4 | C (20–70) | 70 | | | | 10 YR 5/8 yellowish brown | 8.2 | 40 | 0.14 | 0 |
| Sandplateau Stein | R (70+) | bedrock | | | | 10 YR 8/1 white | 8.5 | 52 | 0.03 | 0 |
| \multicolumn Reference Samples from Petra Region: Rocks, Current Fans, and Cemented Paleosol: Cambic Calcisol (Loamic) |||||||||||
| Ba'ja Sandstein (rock) | | surface rock | | | | 10 YR 8/2 white | 8.3 | 81 | 0.02 | 1 |
| Disi Sandstone | | rock outcrop | | | | 7.5YR 8/1 white | | | 0.14 | 4 |
| Um Ishrin sandstone | | rock outcrop | | | | 5YR 4/6 yellowish red | | | 0.06 | 2 |
| Abu Khushayba sandstone JH | | rock outcrop | | | | 2.5YR 5/3 reddish brown | 8.7 | 97 | 0.02 | 2 |
| JH limestone outcrop | | rock outcrop | | | | 7.5YR 8/4 pink | | | 0.85 | 97 [2] |
| Beidha Fan | | surface | | | | 10YR 6/4 light yellowish brown | 8.44 | 346 | 0.46 | 16 |
| Fan Umm Sayhoun | | surface | | | | 10YR 8/4 very pale brown | 8.98 | 136 | 0.07 | 5 |
| Loess(?) Umm Seihun (US 1) | | Cemented paleosol | | | | 7.5YR 8/3 pink | 8.9 | 76 | 0.44 | 34 |

[1] Average from three methods (see Appendix C for individual results). [2] Estimate due to the presence of dolomite.

### 2.1.2. Current Dust

Dust was collected by H. J. Bruins near the campus of the J. Blaustein Institutes for Desert Research in the Negev (Midreshet Ben Gurion, N 30.85135, E 34.78099) from elevated and sheltered surfaces, including a large plastic garden table, at about 80 cm height above ground level. These surfaces were cleaned before an expected dust storm. After the dust storm, the dust was carefully brushed together and moved with a small spoon into a plastic sample bag. These collection surfaces were always sheltered from rainfall. On one occasion only, after a very severe dust storm, dust was also collected from an area where rain had fallen as well. Therefore, in most of the above cases, washout dust was not involved, but only dry dust fall.

In the Petra region, one collector, 45 × 35 cm in size (plastic box) and filled with dry standard glass marbles, was mounted on the roof of the house of Saleh Suleiman, the guard of the Islamic *weli* on Jabal Haroun (Figure 7). The house is located close to the Snake Monument of the ancient cemeteries of Petra, at the foot of the mountain (N 30.319891, E 35.435040). In addition, one dust sampler was placed very close to the mountaintop of Jabal Haroun (N 30.31520, E 35.40406). Dust sampling faced several technical difficulties on Jabal Haroun, due to the rapid disintegration of plastic boxes in the sun and the disappearance of marbles, as they were attractive for children. The solution was hiding the sampler on a secluded rock outcrop that could only be accessed after a courageous climb. That meant, however, that this sampler had to be placed on ground level on the rock. As far as possible, dust samples were collected after every major dust storm. However, simultaneous dust collection from all locations was constrained.

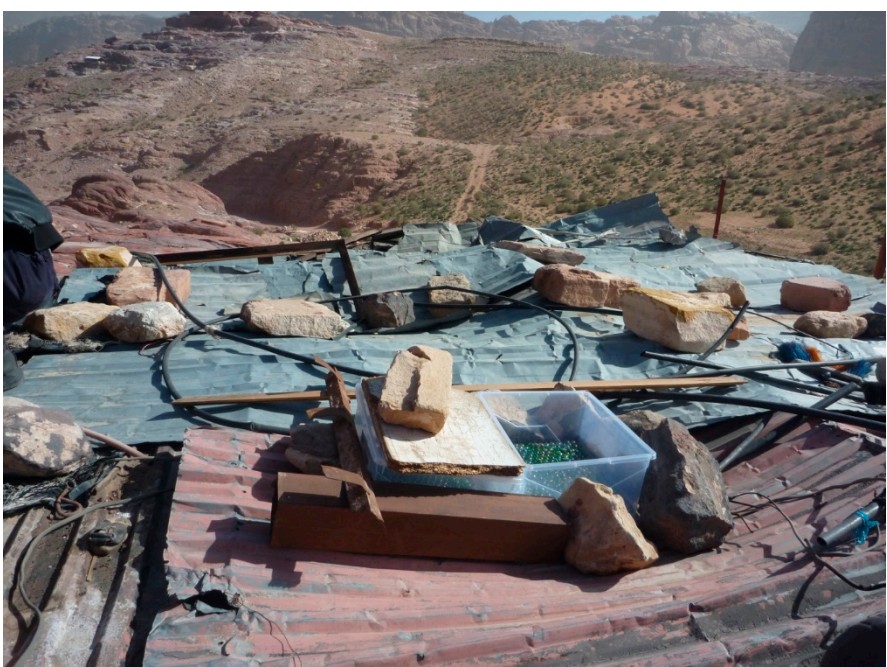

**Figure 7.** Dust sampler on the roof of Saleh Suleiman's house. Note the ridge of fine sediments in the background to the right, covered by small bushes and patches of grass.

As it was not possible to operate a weather station simultaneously with the sampling, weather conditions associated with deposition were gathered from the Israel Meteorological Service (https://ims.data.gov.il/) for Sede Boker, and from ventusky.com and field observations in the Petra region.

### 2.2. Methods of Sample Collection, Laboratory Analysis, Dating, and Statistical Evaluation

A detailed description of the methodologies of sample collection, laboratory analysis, dating, and statistical evaluation can be found in Appendix B. Soil profiles were classified according to [52] and

the horizons were described according to Soil Survey Staff, 2010. As deviation of the latter, we followed the concept of V-horizons, as proposed by [53] for vesicular layers. V-horizons are defined as follows:

"V horizons: Mineral horizons that have formed at the soil surface, or below a layer of rock fragments (e.g., desert pavement) or a physical or biological crust. They are characterized by the predominance of vesicular pores." ([53], p. 8).

Samples were analyzed in the laboratory regarding color, pH, electrical conductivity, organic matter, and contents of calcium carbonate. Three methods of $CaCO_3$ determination were used and various samples were analyzed with more than one method to ensure the comparability of results (see Appendix C, Table A1). Indicators of pedogenesis were evaluated by iron extraction, with oxalate and dithionite, and by calculation of weathering indices from total elements determined by X-ray fluorescence. Magnetic susceptibilities were also measured (see Appendix B for detailed method description).

Particle sizes were analyzed without acid-based removal of $CaCO_3$, with one exception to check the possible cementing role of secondary carbonates (sample *US 1*, see Table 3). Wet sieving and a Sedigraph were used for all samples, except the current dust. As amounts of the latter were often very small, a Malvern MasterSizer 3000 laser device had to be used. In order to check the comparability of results, the particle sizes of various samples were measured with both methods. Results suggest that they are well comparable for current dust samples, in particular if the silt-clay border of the laser measurements is adjusted to 5 μm (following [54]). For the ruin soils, however, sieving and Sedigraph seemed to produce more reliable results (see Appendix E).

For numerical dating, ages were determined by the radiocarbon method, Thermoluminescence (TL) of pottery sherds, and Optically Stimulated Luminescence (OSL) of sediments (see Appendix D). Archaeological ages of the investigated ruins were derived from typological dating of pottery and the building styles of the structures.

Statistical evaluation of particle size distribution was conducted using the Gradistat for Excel program [55] and using the EMMAgeo v0.9.4 R package for end-member analysis [56]. As well, we applied Principle Component Analysis (PCA) using the R-function prcomp [57] to describe similarities between the samples according to parameters of particle size distribution and magnetic susceptibilities. Furthermore, we utilized a non-parametric random forest approach using the R-package randomForest [58] to test whether particle sizes and magnetic susceptibilities are suited to distinguish between different types of deposit in this study (defined as current dust storm sediment, reference samples, and soils).

## 3. Results

Sedimentation in the ruins appeared as a continuous process, possibly with some variations, as described in detail in Appendix A. No indications of hiati could be observed. Diagnostic horizons show that all ruin soils are characterized by V-C profiles. The paleosols, in contrast, exhibited some initial development of brown color, which indicates weak formation of a B-horizon. All paleosols seem truncated and it is not possible to say whether a V-horizon was present at the surface during their formation. The natural and currently forming hilltop soil showed an A-C profile and no vesicular layer.

Profile classifications suggest that the ruin soils are indeed characterized by accumulating dust, which is indicated by the presence of V-horizons. Despite the presence of these vesicular layers, the soils do not possess yermic properties as defined by [52], as the organic matter contents of 0.7–1.2% are too high to match the definition of the additionally required aridic properties. As a vesicular layer is absent from the currently forming natural hilltop soil, it seems likely that less dust is incorporated there.

### 3.1. General Soil Properties

#### 3.1.1. Ruin Soils in the Negev

Table 2 shows a summary of the samples taken in the Negev, the soil types according to [52], the calibrated ages, and the results of color, pH, conductivity, and calcium carbonate contents. The colors of the ruin soils are pale brown colors, dominated by 10 YR. They are similar among each other and to the loess aprons, but different from the rock outcrops and the Turonian paleosol. The pH-values are of similar alkaline milieus between 8–9. Some differences of conductivities, organic matter, and $CaCO_3$ contents occur. Compared to the loess aprons, most of the ruin soils show strongly elevated conductivities. Organic matter contents of the ruin soils are elevated as well, but there is no direct correlation of organic matter and conductivities.

Apart from conductivity, the paleosol properties below the ruin in the central wadi of Horvath Haluqim are very similar to the paleosol in the nearby reference loess apron in Nahal HaRo'a. Both ruin soils exhibit higher $CaCO_3$ contents of 34%–42% than the loess paleosols, which reach 27%–29%, although there are no indications for plaster or other artificial additions rich in calcium carbonate. Higher conductivities and calcium carbonate contents could be due to the retaining walls of the ruins, as these might not only trap dust, but also rainwater that subsequently evaporates.

#### 3.1.2. Ruin Soils Near Petra

Table 3 summarizes the soil types according to [52], the calibrated ages, and the general soil properties of the sampled ruins soils and reference samples around Petra. While natural hilltop soils were not found in the investigation area in the Negev, the Protic Regosol (Arenic) developed on the flat sandstone plateau near Petra illustrates soil development without the protection of archaeological structures in that area (*Sandplateau* samples).

Except for a slight change of color and $CaCO_3$ contents, the Protic Regosol (Arenic) seems to resemble crushed bedrock (*Sandplateau* samples; *Sandplateau Stein* is the underlying bedrock). The sampled sandstone units show considerable variety. Color range is 2.5–10 YR and $CaCO_3$ contents ranges are 0%–4%, similar to previous investigations [59].

High $CaCO_3$ contents of 10%–18% are found in all ruin soils, except the one covering the monastery on Jabal Haroun, where 7% were found. This is in agreement with field observations that mud mortar from the walls of the monastery contributed to the substrate covering the courtyard (see detailed description of profiles in Appendix A). There were no indications that plaster or other artificial materials rich in calcium carbonate were added to the soils. On the contrary, additions of mortar made from mud apparently led to lower $CaCO_3$ contents.

The pH-values are alkaline, between 8–9, while conductivities and contents of organic matter range mostly below 100 μm/cm and 0.6% $C_{org}$, respectively. Only one sample from the monastery cover shows an elevated conductivity of 1296 μS/cm and the paleosol below Shkarat Msaied has 1375 μS/cm. The elevated conductivity in the monastery might be connected with remains of mud mortar, which was possibly mixed with organic garbage, while the higher conductivity of the Shkarat Msaied paleosol could be connected with its secondary carbonates or the possibility of a nearby spring in the past. Kinzel, M. [60] reported on water in ~1 m depth below the gravels of the nearby valley, which suggests occasional flooding of the paleosol during wetter periods, in particular if the past valley was less incised.

### 3.2. Ages and Sedimentation Rates

The bulk density of the soil covering the hilltop ruin at the western wadi in the Negev was determined as 1.074 g/cm$^3$, which is used as a reference for estimating weight-specific sedimentation rates. Despite various insecurities, described below, rates could be calculated as shown in Table 4.

**Table 4.** Average dust sedimentation rates calculated for the ruin soils in the Negev and Petra region.

| Site | mm/year | $g/m^2$ $year^{-1}$ | Comment |
|---|---|---|---|
| **Negev** | | | |
| Ruin near Western Wadi | 0.05 | 50 | Minimum rate: ruin filled completely in the past |
| Ruin near Central Wadi | 0.13–0.2 | 140–210 | Lower boundary unclear; occupation & collapse layers? |
| **Petra region** | | | |
| Jabal Farasha triclinium | 0.14 | 150 | Apparently constant deposition; consistent ages |
| Umm Saysaban | 0.02 | 24 | Minimum rate: ruin filled completely in the past |
| Jabal Haroun monastery | 0.10 | 110 | Minimum rate: lower boundary unclear |

### 3.2.1. Sedimentation Ages of the Negev Hilltop Ruin above the Western Wadi (HH-WW-R)

The wall remains of the hilltop ruin, overlooking the western wadi, were completely filled with sediment, which suggests that it may have stopped collecting sediment in the past. As OSL-dating methods could not be applied there, a rough age of the sediment was estimated from the age of the archaeological structure. Similar tumuli in the region were dated to the Intermediate Bronze Age of 2500–2000 years BCE [61]. As no indications for disturbances could be noticed, it seems likely that the sediment accumulated from the time of abandonment at least until the retaining walls had completely filled with sediment.

The sherd that was encountered at 15 cm depth belongs typologically to the hand-made 'Negebite pottery'. Following earlier descriptions of this type, Amiran, R. [62] summarized the hand-made wares and vessels found in the Negev highlands under this name, but it remained unclear whether they represented a distinctive ceramic tradition that may have appeared and disappeared during a time frame of possibly thousands of years [63]. Gunneweg et al. [64] found evidence, from provenance studies, that the Negebite wares were probably mostly not produced in the Negev, but near Wadi Feinan in Jordan and thus imported, and suggested the term "Coarse hand-made ware". Martin and Finkelstein [65] found, from further provenance studies, that the so-called Negebite pottery was connected with the ancient copper industry and occurred mainly during the 13th–9th century BCE. They interpreted it as household ceramics and a marker of pastoral-nomadic and tribal desert societies, which was mostly, if not all, imported from the Wadi Araba. Kleiman et al. [63] found a similar pattern in the Timna valley and suggested that this pottery is indicative for the second half of the 10th century BCE or slightly earlier.

The possible pottery production time was dated, by TL, to 970–470 BCE. If the median TL-age coincides with sedimentation, approximately 0.03 mm $a^{-1}$ of sediment were deposited from the construction of the monument until the pottery was laid down and around 0.06 mm $a^{-1}$ thereafter, if assuming that the ruin gathered sediment until today. The resulting time-normalized average sedimentation rate of 0.05 mm $a^{-1}$ must, however, be interpreted as a minimum since the ruin may have filled up and stopped gathering sediments long before today. In addition, the approximate construction time of the structure and the production age of the pottery can only provide rough estimates of sedimentation ages.

### 3.2.2. Sedimentation Ages of the Negev Ruin Overlooking the Central Wadi (HH-CW-R)

In the ruin above the central wadi a slight depression was present, which suggests ongoing sedimentation. Charred materials were identified in the upper unit at depths of 7, 14, and 20 cm. The fragment from 7 cm was selected from several not specifiable small Dicotyledon branches, while those from the layers below were selected amongst Chenopodiaceae fragments. They gave radiocarbon dates of ~1946 CE (the sample was too recent for calibration, as the age is located outside the calibration curve), 1643–1918 CE, and 1694–1918 CE. These suggest that this layer, characterized by a slightly platy structure, has been deposited rather recently (see Appendix A for a detailed description). The platy structure might be connected with temporarily standing rainwater that was retained by the walls or by some kind of activities leading to compaction, such as use of the place for Bedouin campfires. Trampling could have led to a collapse of the vesicles and thus the platy structure. Due to absence of

visible evidence of disturbances, we assume that the radiocarbon ages of the charred materials likely correspond to the deposition ages of the sediments.

An OSL-sample, acquired from the underlying layer at ~25 cm depth, yielded an age of 870–1020 CE. The following layer, 35–40 cm in depth, displayed sub-angular pebbles, possibly corresponding to a collapse of the archaeological structure or a former clast pavement that was buried by fines, analogous to buried desert pavements (see e.g., [66]). Another OSL-sample was taken from apparently undisturbed fine sediment at 55 cm depth from a layer, perhaps, of inhabitation time, which contained some pottery at 50 cm depth. The sediment yielded an OSL age of 2570–1990 BCE, which covers the Intermediate EB-MB archaeological period. The tempered pottery was dated by $^{14}$C on the organic, ashy temper and by TL, using quartz and feldspar. TL age was 830–300 BCE, and the radiocarbon age was 6210–6092 BCE. The differing ages suggest that older ashy material was mixed with sandy clay or that old carbon was present in the clay [67]. A loessial paleosol, occurring at ~70 cm below the archaeological structure, comprises a natural substrate and was dated by OSL (in 95 cm depth) to 23,980–20,580 BCE. This is in good agreement with the primary upper loess chronology of the Negev hilltops [12] and slightly pre-dates the onset of increased loess erosion [68].

The mismatch of the TL-age and the OSL-age pottery may indicate very limited sedimentation until deposition of the pottery piece, perhaps connected with maintenance and regular cleaning of the structure. Alternatively, some bioturbation or disturbance connected with the collapse of the building, which could not be observed in the profile, may have been involved in the deposition of the sherd. In the latter case it might belong to the final phase of use of the building. If assuming that the OSL-age in 55 cm corresponds to natural sedimentation, one would arrive at a sedimentation rate of 0.09 mm a$^{-1}$ for the period from approximately 2300 BCE–950 CE. Assuming that the median pottery TL-age in 50 cm marks the approximate onset of natural sedimentation, one arrives at an average sedimentation rate of 0.27 mm·a$^{-1}$.

It seems likely that the ruin was re-used for occasional campfires of Bedouins, which means that the platy top layer, with its different structure, resulted from compaction and has a slightly deviant sedimentation history than the lower part of the profile. Using mean ages, and assuming that the charcoal ages correspond to an ongoing sedimentation, the rate of the last ~220 years should be 0.46 mm a$^{-1}$. For the period from 950–1800 CE, sedimentation rates are estimated to be 0.19 mm a$^{-1}$.

Time-normalized average sedimentation rates, using mean ages over the whole profile, are 0.14 mm a$^{-1}$, when using the OSL-age of the sediments as lower boundary, and 0.2 mm a$^{-1}$ if the pottery TL-age is taken as start time of post-occupational sedimentation.

### 3.2.3. Sedimentation Ages in the Petra Region

The hilltop triclinium of Jabal Farasha had been built with sandstones placed on smoothed sandstone bedrock and no traces of mudbrick, plaster, or mortar made from mud could be observed. One OSL-age, from approximately 15 cm depth, yielded a deposition time of 870–1020 CE. If assuming that the triclinium was abandoned with the prohibition of private cults after the Roman annexation of the Nabatean kingdom, approximately 0.13 mm a$^{-1}$ were deposited from the 2nd until the 10th century CE. Since the walls of the triclinium are not yet completely covered by sediment, it seems possible that the structure is gathering sediment until today. Then a sedimentation rate of 0.14 mm a$^{-1}$ from the 10th–21st century, and a time-normalized average sedimentation rate over the whole profile of 0.14 mm a$^{-1}$ can be calculated.

Numerical dates are not available at the other studied sites near Jabal Haroun, but the archaeological context of the ruins gives some indication of the age of the sediment. At Umm Saysaban, dust deposition in the structure probably started ~2500 BCE after the orderly desertion of the place [69]. Sandstone wall remains had completely filled with sediments, which means that deposition may not have continued until sampling, but stopped or was greatly reduced at some time in the past. There are no indications of mudbrick or mortar made of mud. Assuming ongoing sedimentation, the rate is 0.02 mm·a$^{-1}$

(or 24 g m$^{-2}$·a$^{-1}$), but this is a minimum since it is unknown when the wall remains were covered and may have stopped collecting sediments.

Sediments covering the ruins of the monastery of Jabal Haroun are probably ~1000 years old, since human activities in the sampled northern courtyard apparently terminated at some time during the 10th century CE [70]. The ruin is assumed to have collected sediments at least until its excavation in 1998. Assuming that the 10 cm sediment unit that was studied with sample JH 1 accumulated during approximately 1000 years, a sedimentation rate of 0.1 mm·a$^{-1}$ (or 110 g m$^{-2}$·a$^{-1}$) can be calculated. However, this is only a minimum rate due to the diffuse lower boundary with the stone rubble of the monastery, which makes it difficult to determine the true depth of the profile. Voids between the stones filled first and it is difficult to assess the contribution of mortar made from mud that had been used to seal walls, and may thus have contributed to this sediment.

Regarding the paleosols below the PPNB buildings at Shkarat Msaied and Abu Suwwan, a minimum age of 10,000 years, or older, is reasonable. The excavated site of Shkarat Msaied was dated to 8340–7960 cal BCE [71]. In this context, the $^{14}$C age 12,385–12,131 BCE of the Dicotyledon charcoal fragment, from the ashy layer covering the ruins of the site, suggests a presence of older charred materials at the place, which was re-deposited after abandonment. Such re-deposited older ash deposits may also explain the large amount of ashy sediments covering the site. In contrast, the radiocarbon age, 2014–1916 BCE, of the conifer charcoal fragment from the (much smaller) ashy sediment covering the PPNB site of Abu Suwwan may indicate a potential re-use of the place during the Bronze Age.

*3.3. Grain Sizes*

Table 5 shows a summary of grain size analysis results for a selection of exemplary samples from both investigation regions. Full results appear in Tables A2–A4 in the Appendix C.

**Table 5.** Selected results of particle size analysis, including some statistical parameters calculated with the Gradistat program. Note that the very fine sand is part of the fine sand fraction. Note that the very fine sand (in italics) is part of the fine sand fraction. Skeleton is shown as well (in italics), and summarizing grain size units (in bold). Medium silt was not calculated from laser data. Full results: See Tables A2–A4 in Appendix C.

| Sample No. | Skeleton > 2 mm (%) | Sand % | Silt % | Clay % | Coarse Sand % | Medium Sand % | Fine Sand % | Very Fine Sand % | Coarse Silt % | Medium Silt % | Fine Silt % | Coarse Clay % | Medium Clay % | Fine Clay % | Mode 1 (μm) | Mode 2 (μm) | Mode 3 (μm) | Mean [μm] |
|---|---|---|---|---|---|---|---|---|---|---|---|---|---|---|---|---|---|---|
| | | | | | | | Negev ruin soils: Horvat Haluqim, ruin at Western Wadi (HH-WW-R1): Calcaric Leptosol (Protic, Siltic); ruin at Central Wadi (HH-CW-Ruin): protic regosol (Siltic) | | | | | | | | | | | |
| HH-WW R1 10 | *21* | **42** | **45** | **13** | 4 | 8 | 29 | *23* | 18 | 15 | 13 | 9 | 2 | 2 | 132 | 0 | 0 | 25 |
| HH-CW-Ruin 10 | *4* | **21** | **53** | **26** | 2 | 3 | 16 | *13* | 22 | 14 | 16 | 14 | 6 | 5 | 42 | 4 | 0 | 10 |
| HH-CW-R-60 | *24* | **36** | **51** | **13** | 5 | 5 | 27 | *23* | 19 | 18 | 14 | 9 | 3 | 1 | 132 | 0 | 0 | 22 |
| | | | | | | | Negev reference samples: Loess paleosol of loess apron in Nahal Haroa (NH-LA): Cambic Calcisol (Siltic); and chalk outcrop | | | | | | | | | | | |
| NH-LA-30cm | *5* | **27** | **58** | **15** | 2 | 2 | 23 | *22* | 35 | 12 | 10 | 8 | 6 | 1 | 42 | 0 | 0 | 20 |
| Haroa Farm - chalk | *n.a.* | **12** | **86** | **2** | 0 | 1 | 11 | *n.a.* | 1 | 7 | 78 | 1 | 0 | 2 | 4 | 0 | 0 | 5 |
| | | | | | | | Petra ruin soils: Jabal Haroun monastery (JH 1): Protic Arenosol (Aridic, Aeolic); Jabal Farasha triclinium (JF site 124/1): Calcaric Leptosol; Umm Saysaban: hilltop ruin: Calcaric Leptosol (Protic) | | | | | | | | | | | |
| JH 1 | *n.a.* | **75** | **13** | **12** | 4 | 31 | 39 | *13* | 5 | 4 | 4 | 4 | 3 | 5 | 132 | 0 | 0 | 68 |
| JF site 124/1 15 cm | *5* | **53** | **35** | **12** | 3 | 16 | 34 | *16* | 19 | 9 | 7 | 6 | 3 | 3 | 132 | 0 | 0 | 41 |
| Umm Saysaban 5 cm | *13* | **51** | **35** | **14** | 3 | 16 | 32 | *13* | 15 | 11 | 9 | 8 | 4 | 2 | 132 | 0 | 0 | 35 |
| | | | | | | | Petra region buried early Holocene paleosols: Abu Suwwan PPNB Ruins: Protic Regosol (Aeolic); Shkarat Msaied PPNB Ruins: Chromic Cambisol (Protocalcic) | | | | | | | | | | | |
| Abu Suwwan below nw 65 | *2* | **62** | **23** | **15** | 1 | 39 | 22 | *7* | 7 | 8 | 8 | 7 | 5 | 3 | 415 | 4 | 0 | 48 |
| Shakarat Msaid 1 | *0* | **52** | **26** | **22** | 0 | 30 | 21 | *5* | 11 | 7 | 8 | 8 | 6 | 8 | 415 | 4 | 0 | 27 |
| | | | | | | | Petra region natural hilltop soil with sandstone bedrock sample (*Stein*): Protic Regosol (Arenic) | | | | | | | | | | | |
| Sandplateau 1 | *0* | **80** | **12** | **8** | 3 | 59 | 18 | *n.a.* | 6 | 4 | 3 | 4 | 3 | 2 | 415 | 0 | 0 | 143 |
| Sandplateau Stein | *0* | **89** | **7** | **3** | 1 | 75 | 12 | *n.a.* | 2 | 2 | 4 | 3 | 1 | 0 | 415 | 0 | 0 | 248 |
| | | | | | | | Petra Region Reference Samples: Rocks and Current Fans | | | | | | | | | | | |
| Um Ishrin sandstone | *n.a.* | **81** | **12** | **7** | 18 | 55 | 9 | *n.a.* | 3 | 4 | 4 | 3 | 1 | 2 | 415 | 0 | 0 | 186 |
| JH limestone | *n.a.* | **48** | **46** | **6** | 29 | 14 | 5 | *n.a.* | 12 | 27 | 7 | 2 | 1 | 2 | 1315 | 13 | 0 | 74 |
| Beidha Fan | *9* | **69** | **21** | **11** | 3 | 43 | 22 | *n.a.* | 9 | 5 | 6 | 7 | 3 | 1 | 415 | 1 | 0 | 72 |
| | | | | | | | Current dust storm sediment samples from the Negev (HH – Horvat Haluqim) and the Petra region (JH – Jabal Haroun) | | | | | | | | | | | |
| 11-12-10-HH | *0* | **24** | **56** | **20** | 0 | 1 | 23 | *21* | 38 | n.a. | 18 | 19 | 1 | 0 | 42 | 1 | 0 | 16 |
| 29-02-12-HH | *0* | **30** | **58** | **12** | 0 | 0 | 30 | *28* | 46 | n.a. | 11 | 12 | 0 | 0 | 42 | 1 | 0 | 27 |
| JH-07.01.2017 | *0* | **83** | **13** | **4** | 10 | 62 | 11 | *5* | 7 | n.a. | 6 | 4 | 0 | 0 | 415 | 0 | 0 | 204 |
| JH-15.02.2017 | *0* | **44** | **43** | **13** | 1 | 24 | 19 | *11* | 22 | n.a. | 21 | 13 | 0 | 0 | 415 | 42 | 10 | 34 |
| JH-01.03.2017 | *0* | **46** | **46** | **8** | 0 | 11 | 35 | *22* | 30 | n.a. | 16 | 7 | 0 | 0 | 132 | 0 | 0 | 40 |

### 3.3.1. Negev

Table 5 shows the grain size distribution of exemplary samples from the Negev as well as some of the statistical parameters that were determined with the Gradistat program (for full results, see Table A2 in Appendix C). All textures are silt-dominated, but some differences can be observed, as follows: The ruin at the western wadi, and the lower part of the sediments in the ruin of the central wadi, contain relatively more sand and less clay than the paleosols and the top layer of the ruin in the central wadi. This might point to increased sand deposition during the Iron Age, as the pottery sherds were apparently deposited contemporaneously.

The sand in the ruins is coarser, as compared to the loess apron samples. Apart from the loess aprons, none of the reference geologic materials match the grain size distribution/texture in the ruins, suggesting that these outcrops contributed little substrate to the ruins. Gradistat analysis of the grain size shows that particle size distributions are unimodal or bimodal, with peaks around 132, 42, and/or 4 µm. Only sample HH-CW-Ruin 25, from the ruin near the central wadi, is trimodal with a third peak at 1315 µm, which reflects the layer change observed in the profile (see Table A2 and the detailed profile description in Appendix A). The fine sand peaks of the ruin soils are clearly reflected in the modes.

### 3.3.2. Petra Region

Table 5 also shows exemplary results of grain size analysis near Petra. Sand dominates throughout and the natural hilltop soil of the *Sandplateau* reference samples confirm that soils in the vicinity of Petra largely represent physically disintegrated sandstone. Despite some variations of sand and clay contents in the sandstone rocks, it is important to note that mainly medium sand is released from the stones (with the exception of the rock below the ruin on Jabal Farasha (JF site 124), which is dominated by coarse sand, see Table A2 in Appendix C). Although grain size results of the sandstone rocks are only estimates, due to the necessity to gently crush the rock for analysis, the dominance of medium sand was confirmed [59]. The sandstone parent rocks are generally defined as quartz arenite, indicating the near dominance of quartz as a constituent mineral. However, thin sections of some samples of the Disi Sandstone, for example, show well-preserved remnants of mica (Figure 8). The grain size also shows some variability.

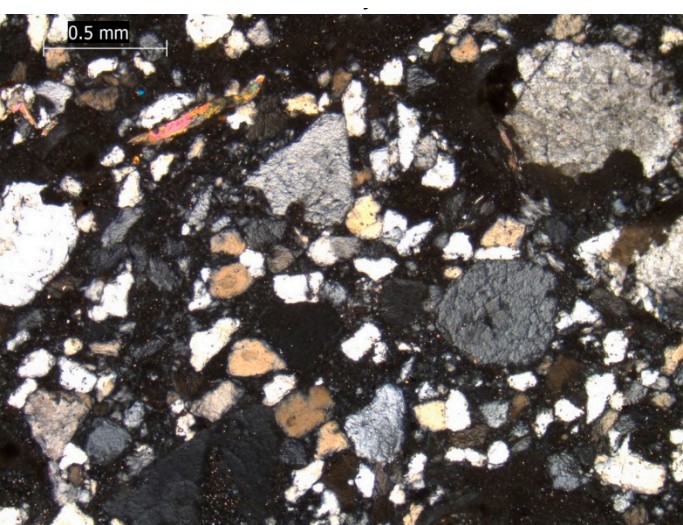

**Figure 8.** Thin section from a Disi Sandstone specimen from Beidha. Notable in it are relatively angular and poorly sorted quartz grains and a well-preserved mica particle in the upper left corner. XPL.

In contrast to the sandstones and natural hilltop soil, fine sand and coarse silt dominate in the ruins, with the only exception being sample *FJHP Site 1, trench R.* This is the sediment cover of the

monastery of Jabal Haroun, which probably contains significant amounts of mud mortar. As well, contents of very fine sand are relatively high in the ruins, with 13%–19%. The paleosols below the sites of Abu Suwwan and Shkarat Msaeid, as well as the cemented paleosol of sample US 1, take some intermediary position; they contain a higher share of medium sand than the ruin soils, but less than the current surface soils.

Gradistat analysis of grain sizes of the Petra region samples shows that more or less all the reference samples are unimodal, with a peak around 415 μm. This mode also dominates the paleosols, but they are mostly bimodal, with a second peak around 4 μm, and characterized by much smaller mean grain sizes. The ruin soils, in contrast, are mostly unimodal, with a peak around 132 μm.

### 3.4. Current Dust Samples

3.4.1. Deposition Conditions, Total Amounts, Colors, and CaCO$_3$ Contents of Current Dust

The dust samples that were collected from 2003–2018 at the Sede Boker Campus in the Negev (only a few km from Horvat Haluqim and from the nearby Meteorological Station at Kibbutz Sede Boker) and from 2016–2017 at the two sampling sites on and near Jabal Haroun, which are listed in Table 6. A total of 12 samples from the Negev and 11 from Petra were gathered. The total amounts of dustfall varied greatly. Four samples consisted of just around 1 g, which severed the analyses potential.

The available information on weather conditions during the deposition of the dust samples does not show strong relationships with the collected material. Precipitation might play some role in the Petra region, the snowstorm that could be sampled on Jabal Haroun brought the second largest dust sample. The largest individual sample amounts were deposited on Jabal Haroun during relatively quiet wind conditions on 5 August 2017. A closed sampler could be opened for this occasion and collected 56 g, compared to 66 g in the sampler that had remained open and was not emptied since March 2017. This suggests that around 10 g of dust accumulated during relatively quiet conditions in the summer and without rain, while one dry storm brought more than three times as much material than could be collected in the snowstorm.

At this time, it was possible to sample the collector on Jabal Haroun and the one at Saleh's house near the Snake Monument simultaneously. Interestingly, the amount of material deposited in the collector at the house was much lower than on the mountain. This might be related to lower wind speeds at the foot of the mountain and it seems possible that a dust devil was involved in the deposition of the large samples on Jabal Haroun. This illustrates the possibly rather erratic character of the roles of wind speeds and local conditions. Unfortunately, no direct measurements are available.

In the valley near the collector on Saleh's house, there are extensive exposures of reddish sandy sediments. They possibly represent slack water deposits of large paleofloods [72]. A contribution of these sediments to the dust sampler on the house seems evident, since these are the only current dust samples showing colors of 5YR. All other samples are characterized by colors of 10YR, with pale brown and yellowish-brown colors near Petra.

The CaCO$_3$ contents of samples collected during dry deposition on Jabal Haroun vary between 2%–5%, not exceeding the possible variation of the sandstones. However, those collected during wet deposition show CaCO$_3$ contents of 11%–15%. The situation is different at Saleh's house. Here, two samples collected during dry deposition show carbonate contents of 11% and 12%, respectively. It should be mentioned that the nearby valley sediments/slack water deposits partly contain rizoliths and other secondary carbonates, while the reference sample of the alluvial Beidha fan has 16% calcium carbonate. These CaCO$_3$ contents support the impression that local sources from the valley bottom contributed to the dust sampler at Saleh's house, while the CaCO$_3$ contents of samples collected on top of the mountain seem to depend more on precipitation.

**Table 6.** List of the collected dust samples, sampling dates and places, collected weights, colors, CaCO$_3$ contents (estimated from Ca %), and prevailing weather conditions from the Israel Meteorological Service (https://ims.data.gov.il/) for Sede Boker, and from ventusky.com and field observations in the Petra region.

| Sample No. | Date of Sample Collection | Place | Max. Wind Speed (km/h) | Average Wind Speed (km/h) | Main Wind Direction from | Precipitation | Weight (g) | Munsell Dry | CaCO$_3$ % from Ca% |
|---|---|---|---|---|---|---|---|---|---|
| **Negev** | | | | | | | | | |
| 25-03-03-HH dry spot | 25.03.2003 | Midreshet Ben-Gurion | 70 | 20–25 | W | rain, but dry deposition | 12 | 10YR 7/6 yellow | n.d. |
| 25-03-03-HH-incl.-washout | 25.03.2003 | Midreshet Ben-Gurion | 70 | 20–25 | W | rain | 14 | 10YR 7/6 yellow | 35 |
| 11-12-10-HH | 11.12.2010 | Midreshet Ben-Gurion | 80 | 25–30 | SW | dry | 15 | 10YR 7/4 very pale brown | 38 |
| 12-12-10-HH | 12.12.2010 | Midreshet Ben-Gurion | 80 | 25–30 | SW | dry | 27 | 10YR 7/4 very pale brown | 38 |
| 29-02-12-HH | 29.02.2012 | Midreshet Ben-Gurion | 83 | 25–30 | SW | rain | 9 | 10YR 7/4 very pale brown | 34 |
| 18-04-12-HH | 18.04.2012 | Midreshet Ben-Gurion | 64 | 10–15 | NW | dry | 16 | 10YR 7/4 very pale brown | 30 |
| 20-12-12-HH | 20.12.2012 | Midreshet Ben-Gurion | 68 | 15–20 | WSW | dry | 1 | n.d. | 33 |
| 22-03-13-HH | 22.03.2013 | Midreshet Ben-Gurion | 84 | 15–20 | W | dry | 5 | 10YR 7/4 very pale brown | 44 |
| 11-02-15-HH | 11.02.2015 | Midreshet Ben-Gurion | 81 | 25–30 | SW | very light rain | 3 | 10YR 7/4 very pale brown | 40 |
| 01-12-16-HH | 01.12.2016 | Midreshet Ben-Gurion | 60 | 15–20 | W | rain | 1 | n.d. | n.d. |
| 05-01-18-HH | 05.01.2018 | Midreshet Ben-Gurion | 80 | 25–30 | W | rain | 1 | 10YR 7/6 yellow | n.d. |
| 28-03-18-HH | 28.03.2018 | Midreshet Ben-Gurion | 73 | 15–20 | S | dry | 3 | 10YR 7/6 yellow | n.d. |
| **Petra Region** | | | | | | | | | |
| JH-25.11.2016 | 25.11.2016 | Jabal Haroun | 60 | 30–35 | SE | dry | 1 | 10YR 5/3 brown | n.d. |
| JH-19-12-2016 | 19.12.2016 | Jabal Haroun | 61 | 30–35 | W | very light rain | 2 | n.d. | 3 |
| JH-31-12-2016 | 31.12.2016 | Jabal Haroun | 42 | 15–20 | NW | dry | 1 | n.d. | n.d. |
| JH-07.01.2017 | 07.01.2017 | Jabal Haroun | 47 | 15–20 | W | dry | 8 | 10YR 6/3 pale brown | 6 |
| JH-15.02.2017 | 15.02.2017 | Jabal Haroun | 45 | 25–30 | NW | snow | 16 | 10YR 6/3 pale brown | 15 |
| JH-01.03.2017 | 01.03.2017 | Jabal Haroun | 39 | 5–10 | S | rain | 3 | 10YR 5/2 grayish brown | 11 |
| 05-08-17-JH | 05.08.2017 | Jabal Haroun | 30 | 15–20 | NW | dust devil? | 66 | 10YR 5/4 yellowish brown | 5 |
| 05-08-17-JH closed box | 05.08.2017 | Jabal Haroun | 30 | 15–20 | NW | dust devil? | 56 | 10YR 5/4 yellowish brown | 4 |
| Saleh.25.11.2016 | 25.11.2016 | Saleh's house | 60 | 30–35 | SE | dry | 12 | 5YR 5/3 reddish brown | 2 |
| 20-06-16-Saleh | 20.06.2017 | Saleh's house | 51 | 25–30 | NW | dry | 13 | 10YR 5/4 yellowish brown | 12 |
| 05-08-17-Saleh | 05.08.2017 | Saleh's house | 30 | 15–20 | NW | dry | 13 | 5YR 6/4 light reddish brown | 11 |

3.4.2. Grain Sizes of Current Dust Samples

Table 5 includes selected grain size analysis results of current dust samples. Analysis was done with a Malvern MasterSizer 3000 because most dust quantities were too small for the Sedigraph. As the current dust samples potentially resemble the material stored in the archaeological ruins, their grain sizes were compared. Following [54], we decided to apply the 5 μm clay-silt border and to mathematically eliminate the fraction of medium silt in the laser measurements. In order to check the comparability specifically for our material, a few dust samples where sufficient amounts were available were analyzed with the Sedigraph too. These results and a discussion of comparability are presented in Appendix E.

Particle sizes from the Negev show that samples are strongly silt-dominated, but can contain sand fractions of up to 30%. The sand consists nearly purely of very fine sand. It should be noted that the sampler was placed at about 80 cm height above ground level, which should strongly limit the deposition of heavier and larger sand grains.

In contrast to the report by [73], higher clay contents do not seem associated with precipitation. The samples with the highest clay contents that could be gathered are associated with dry storms of relatively high wind speeds (see Table A4 in Appendix C). This might, however, be connected with the position of the sampler that, with one exception, was protected from rain. Colors and $CaCO_3$ contents of dust samples in the Negev match the range encountered in the ruin soils and loess aprons. Samples from the Negev show no discernible role of precipitation or wind speed for total deposition amounts or for $CaCO_3$ contents.

Most dust samples near Petra, in contrast to the Negev finds, are sand-dominated. Some samples even contain coarse sand. This particularly true for the dust sampler on the mountain, which was placed on ground level. However, some coarse sand and significant amounts of medium sand were also collected on the roof of Saleh's house, i.e., approximately 2 m above ground level (see Table A4 in Appendix C). There are significant variations of medium sand in the samples from Petra, contents vary between 11%–65%. We interpret these large sand grains as sign for the deposition of material derived from the local sandstone detritus by strong winds.

The two following samples in the Petra region are silt-dominated: The snowstorm sample of 15 February 2017 and the sample deposited with rain on Jabal Haroun on 1 March 2017. These two samples illustrate that wet deposition can make a major difference for the character of the deposited dust. It should be noted that two samples of dry deposition at Saleh's house, from 20 June and 5 August 2017, contain significant amounts of silt, too. These samples also contain relatively high amounts of $CaCO_3$ which supports that the above mentioned nearby valley sediments contributed material.

Gradistat statistics show very homogeneous properties of the Negev dust samples, which are all characterized by a main mode of ~42 μm. Most show a second peak in the clay fraction around 1 μm and mean grain sizes vary between 9–27 μm.

In contrast to the Negev dust, samples in the Petra region show stronger variations between wet and dry deposition. Most samples from the Petra region are unimodal with main modes in the sand fractions of 415 or 132 μm. The two samples deposited with rain and snow exhibit markedly smaller mean grain sizes of 34–40 μm, while the samples of dry deposition vary between 63–204 μm. The sample deposited with snow is very different, showing two additional modes in the silt fractions of 42 and 10 μm.

*3.5. Statistical Analysis of Grain Size Distributions*

Seven end-members could be modeled with EMMAgeo for the whole dataset and are presented in Figure 9. The samples were numbered from 1–57 during statistical analysis and the robust end-member scores shown in Figure 9 can be attributed to individual samples by their numbers. A detailed list of the samples can be found in Appendix F. Table 7 summarizes the modeled end-members and the potential sources and processes that they might represent.

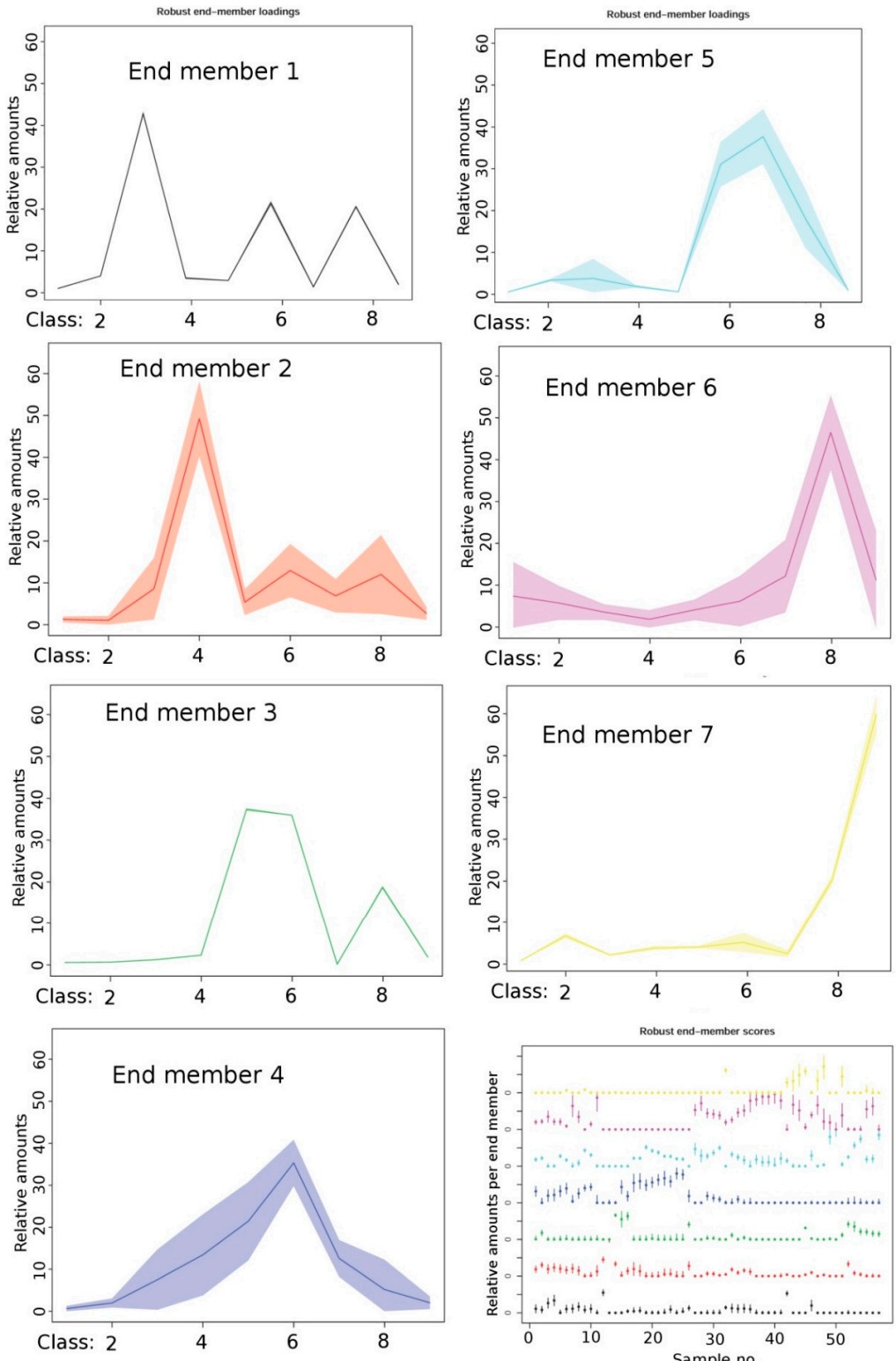

**Figure 9.** Particle size distributions of 7 end-members modeled with EMMAgeo and charts showing the respective end-member scores (relative amounts from 0–1) for each individual sample. Grain-size classes range from 1 (fine clay) to 9 (coarse sand). The chart to the bottom right, showing the robust end-member scores for each sample, displays end-member 1 at the bottom and the subsequent end-members upwards, until end-member 7 at the top. For a detailed list of the samples that were used in calculation of the statistics, including respective individual sample numbers, see Appendix F.

**Table 7.** Summary of end members that were statistically modeled with EMMAgeo. The main modes are displayed in bold.

| No. | Variance Expl. % | Composition | Present in | Possible Processes |
|---|---|---|---|---|
| 1 | 4.5 | **Coarse clay**, coarse silt, medium sand | Soils in both regions, reference samples, Turonian paleosol | In-situ weathering, local redistribution |
| 2 | 8.5 | **Fine silt** (& coarse silt, medium sand) | All Negev samples, ruin soils & snow and rainstorm dust near Petra | Long-range dust transport |
| 3 | 9 | **Medium-coarse silt**, medium sand | Negev references, one Negev dust storm, Petra dust storms, sandstones | Locally re-distributed rock particles |
| 4 | 19 | **Coarse silt** | All Negev soils and dust storms, Petra soils and dust storms with rainfall | Long-range and intermediate dust transport |
| 5 | 18 | **Coarse silt-fine sand**, coarse clay | Negev soils & dust storms, Petra soils & dust storms | Long-range and intermediate dust transport |
| 6 | 31 | **Medium sand**, fine clay | Ruin soils in both regions, most samples of Petra region | Rock weathering and short-range transport from sandstones |
| 7 | 10 | **Coarse sand** (& coarse silt, medium clay) | Some sandstone rocks near Petra and few dry dust storms | Rock weathering and strong winds e.g., by dust devils |

The seven modeled end-members can be described as follows:

1. End member 1 (black line) explains 4.5% of the variance and shows trimodal properties, with a main peak in the coarse clay and smaller modes in the coarse silt and medium sand fractions. It is present in the ruins soils in both areas, some reference samples of rocks, and the Turonian paleosol;

2. End member 2 (red line) explains 8.5% of the variance and shows trimodal properties, with a main mode in the fine silt fraction and minor peaks in the coarse silt and medium sand fractions. It is present in all Negev samples, to some degree in the ruin soils in the Petra region, and in some of the dust storms in the Petra region, particularly those associated with precipitation;

3. End member 3 (green line) explains 9% of the variance and has bimodal properties, with a main mode in the medium-coarse silt fraction and a minor peak in the medium sand fraction. This end member is present mainly in the Negev reference samples, one of the Negev dust storms, one of the Petra sandstone rocks, and in the dust storms in the Petra region;

4. End member 4 (blue line) explains 19% of the variance and has unimodal properties with a peak in the coarse silt fraction. It is present in all Negev soils and in all Negev dust storms, but not in the reference samples. In the Petra region, it is found to a very limited amount in the ruin soils and in the dust storms associated with precipitation;

5. End member 5 (cyan line) explains 18% of the variance and is bimodal, with a main mode in the coarse silt-fine sand fraction and a very minor peak in the coarse clay fraction. It is found in most Negev ruin soils and Negev dust storms and in most Petra ruin soils and Petra dust storms. With the exception of the limestone outcrop, it is absent in all reference samples;

6. End member 6 (purple line) explains 31% of the variance, which exceeds by far all other computed end members. It is bimodal with a strong peak in the medium sand and a small peak in the fine clay fraction. This end member is present in the ruin soils in both investigation areas and in most samples from the Petra region. It is absent in Negev dust storms and reference samples. In the Petra region, it is absent from the limestone outcrop, the dust storms associated with precipitation, one dry dust storm, and two sandstones;

7. End member 7 (yelow line) explains 10% of the variance and has trimodal properties. Its main peak is in the coarse sand fraction, with two minimal other peaks in the coarse silt and medium clay fraction. It is absent in the Negev samples and in the Petra region is only present in most sandstone rocks and a few dry dust storms, including the two samples that may have been associated with a dust devil.

Results suggest that the ruin soils in both investigation regions have similarly complex mixtures of local and remote sources. Coarse sand, medium sand, medium-coarse silt that seems often associated

with medium sand, and coarse clay could be derived from rock weathering and local redistribution processes and explain about 54.5% percent of the variance. This means that probably half of the substrate of the ruin soils can be explained with local sources.

End members consisting of coarse silt and coarse silt-fine sand associated with some coarse clay explain about 37% of the variance. These are not associated with the reference samples of either investigation region, but with the ruin soils and current dust storms, and thus likely represent intermediate-range dust transport.

One end member consists mainly of fine silt and is present in all Negev samples, but only in the ruin soils and the dust storms that were associated with precipitation in the Petra region. It might represent the contribution of long-range dust deposition to the sediments and explain 8.5% of the variance.

### 3.6. Pedogenesis

An important question is whether pedogenesis affected particle size distributions. As there are mostly no B-horizons discernible in the profiles, soil development appears weak and laboratory analyses are required for quantification of weathering intensities. We approached this question by extraction of pedogenic iron oxides, calculation of various weathering indices, and measurement of magnetic susceptibilities. Table 8 presents selected results, including the clay and calcium carbonate contents. For full results and additional interpretation of pedogenesis, see Table A5 in Appendix C.

### 3.6.1. Extractable Iron

Compared to the loess apron samples, which are characterized by very similar weathering intensities, the Negev samples from the ruins at the western wadi (HH-WW-R1) and central wadi (HH-CW-R) are characterized by lower contents of dithionite-extractable iron. In the Petra region, total concentrations of Fe(d) show low values too. Taking the natural plateau soil of the *Sandplateau* samples as reference, iron oxide concentrations are higher in the ruin soils. The highest concentrations of extractable iron are present on Jabal Farasha and Umm Saysaban. In contrast, amounts of dithionite-extractable iron in two of the three studied sandstone samples are minimal, but strongly elevated in the sandstone underlying the triclinium of Jabal Farasha (sample *JF site 124/rock*). This suggests an occasional presence of pre-weathered iron in sandstones and illustrates the variability of the sandstone rocks.

However, the soils overlying the respective sandstones show very different contents of extractable iron, indicating that the iron oxides of the overlying soils are not inherited from their bedrocks. This indicates an allochthonous source of iron, likely moved to the ruin soils by aeolian processes. Plotting the total amount of dithionite-extractable iron Fe(d) against the ratio of dithionite-extractable to total iron Fe(d/t) (Figure 10a), a strong correlation can be observed for the loess aprons and all ruin soils except the monastery of Jabal Haroun. This points to in-situ weathering, or rather, continuous deposition of pre-weathered iron during the genesis of the ruin soils.

When the paleosols in the Petra region are included in the Fe(d)-Fe(d/t) plot, the correlation is less strong (Figure 10b). This supports that these paleosols contain a higher share of disintegrated sandstone rock. Figure 10c shows the Fe(d)-Fe(d/t) relationship for all studied samples from both investigation regions. No correlation is possible, which confirms that sandstone iron sources are reflected by an irregularly scattered pattern.

**Table 8.** Table summarizing indicators of pedogenesis for selected samples. For full results, see Table A5 in Appendix C. CIA refers to the index of chemical alteration according to [74]. χ is the weight-specific magnetic susceptibility, $\chi_{FD}$ is the frequency-dependent change of susceptibility, $\chi_{Ox-diff}$ is the change of susceptibility after oxalate extraction, and $\chi_{Di-diff}$ after iron extraction with dithionite.

| Sample No. | Fe(o) (mg/g) | Fe(d) (mg/g) | Fe (d/t) | CIA Quotient A | CIA Quotient B | K/Al$_2$O$_3$-Ratio | CaCO$_3$ % | Clay % | χ (m$^3$/kg) E-8 | χ$_{FD}$ % | χ$_{Ox-diff}$ % | χ$_{Di-diff}$ % |
|---|---|---|---|---|---|---|---|---|---|---|---|---|
| | | | | | | **Negev** | | | | | | |
| | | | | | | Ruin Western Wadi | | | | | | |
| HH-WW R1 10 cm | 0.21 | 2.52 | 0.14 | 0.92 | 0.80 | 0.09 | 35 | 13 | 44 | 4 | 8 | 31 |
| | | | | | | Ruin Central Wadi | | | | | | |
| HH-CW-R 10 cm | 0.49 | 4.27 | 0.20 | 0.91 | 0.80 | 0.16 | 40 | 26 | 64 | 6 | | |
| HH-CW-R 60 cm | 0.23 | 1.84 | 0.11 | 0.92 | 0.80 | 0.13 | 32 | 13 | 61 | 5 | 2 | 35 |
| | | | | | | Haroa Loess Aprons | | | | | | |
| NH-LA-30 cm | 0.38 | 4.64 | 0.20 | 0.92 | 0.73 | 0.16 | 27 | 15 | 61 | 3 | 10 | 23 |
| | | | | | | **Petra region** | | | | | | |
| | | | | | | Monastery Ruin Soil on Jabal Haroun | | | | | | |
| FJHP Site 1, Trench R | 0.05 | 3.79 | 0.50 | 0.97 | 0.62 | 0.11 | 7 | 8 | 18 | 7 | | |
| | | | | | | Jabal Farasha Hilltop Ruin | | | | | | |
| JF site 124/1 15 cm | 0.30 | 6.72 | 0.45 | 0.91 | 0.49 | 0.07 | 10 | 12 | 26 | 6 | | |
| | | | | | | Umm Saysaban | | | | | | |
| Umm Saysaban 5 cm | 0.25 | 5.56 | 0.34 | 0.91 | 0.52 | 0.06 | 12 | 14 | 28 | 7 | | |
| | | | | | | Abu Suwwan Paleosol below Ruins | | | | | | |
| Abu Suwwan below nw 65 cm | 0.14 | 4.02 | 0.39 | 0.96 | 0.76 | 0.07 | 18 | 15 | 82 | 6 | | |
| | | | | | | Shkarat Msaied Paleosol below Ruins | | | | | | |
| Shkarat Msaied 1 | 0.27 | 4.42 | 0.31 | 0.95 | 0.69 | 0.14 | 15 | 22 | 22 | 5 | | |
| | | | | | | Natural Soil on Sandstone Plateau | | | | | | |
| Sandplateau 1 | 0.05 | 2.47 | 0.41 | 0.96 | 0.32 | 0.06 | 2 | 8 | 20 | −4 | | |
| Sandplateau Stein (sandstone) | 0.00 | 0.14 | 0.20 | 0.96 | 0.03 | 0.01 | 0 | 4 | 0 | | | |
| Baja Sandstein Oberfläche (sandstone) | 0.03 | 0.23 | 0.15 | 0.86 | 0.09 | 0.01 | 1 | 17 | 0 | | | |

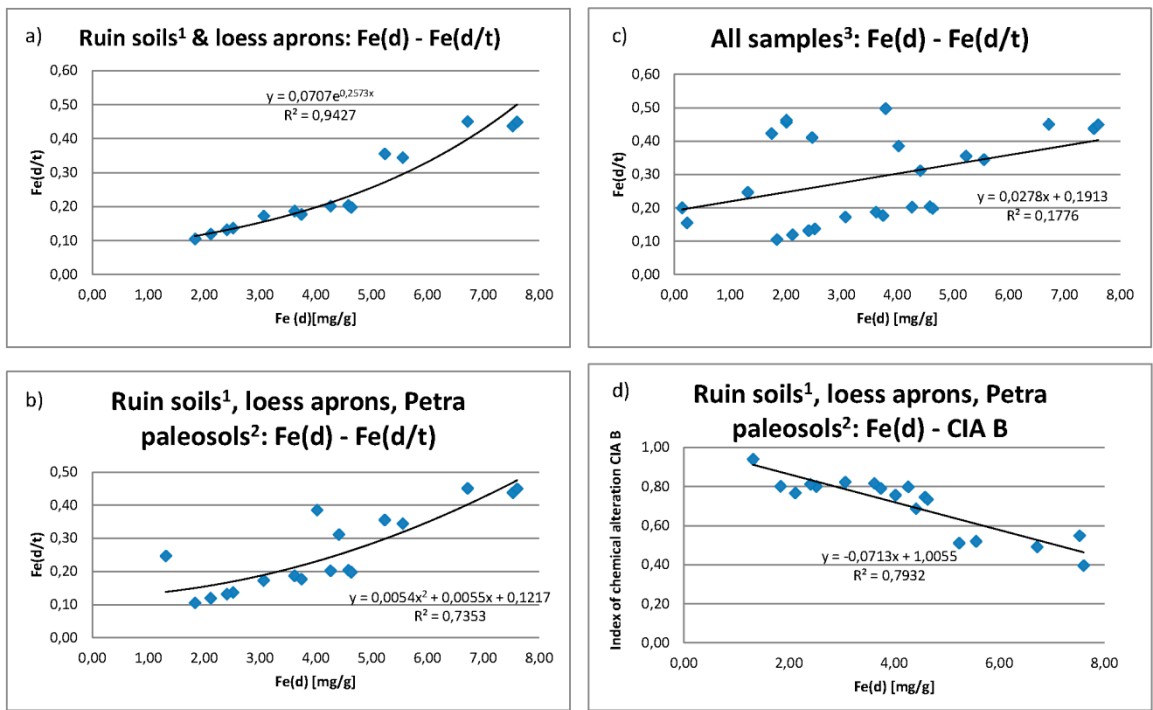

¹Except the monastery of Jabal Haroun. ²Paleosols include Abu Suwwan and Shkarat Msaied. ³Except sandstone sample *JF site 124/1 rock* which is off-scale.

**Figure 10.** (**a**) Ratio of dithionite-extractable iron Fe(d) to dithionite-extractable to total iron Fe(d/t) for the ruin soils and loessial aprons in the Negev; (**b**) also including the paleosols in the Petra region; (**c**) including the reference samples as well. (**d**) Correlation of Fe(d) and the Index of Chemical Alteration (CIA) B.

Other weathering indicators mirror pedogenic iron. The Index of Chemical Alteration B, which refers to the degree of feldspar breakdown [74], corresponds to Fe(d) (Figure 10d). The extraction of pedogenic iron shows some further results that deserve to be mentioned as follows:

- The higher amounts of extractable iron in the loess aprons in Nahal HaRo'a suggest that they are not the source of the material in the ruin soils, despite the similarity of grain sizes.
- Extractable iron is not elevated in the paleosols that contain secondary carbonates, which puts a question mark on their assumed inheritance from periods of intense pedogenesis as proposed by [6].

### 3.6.2. Magnetic Susceptibilities χ

Susceptibilities of the sandstone samples are zero or very low. The current soil of the *Sandplateau*-samples reaches magnetic susceptibilities of 10–20 m³/kg E-8. This is close to the levels of the ruins soils of the Petra region, where amounts of 20–30 m³/kg E-8 are found. In the Negev, the loess aprons and ruin soil in the central wadi show susceptibilities of 50–60 m³/kg E-8, while the ruin soil in the western wadi shows somewhat smaller values of 44 m³/kg E-8. The relative frequency-dependent changes of susceptibilities of all studied samples are similar and indicate that 3%–10% of the magnetic minerals show superparamagnetic behavior. These are very small grains of nanometer scale that are mostly formed during pedogenesis (or fire) and can carry a large part of the magnetic signal [75]. The deviant reductions of magnetic susceptibilities, due to the two types of iron extraction, suggest that the loess aprons in Nahal HaRo'a are characterized by a different set of magnetic minerals than the ruin soils.

Extraction by dithionite reduces susceptibilities of the ruin soils by 31%–54%. This is reflected in a fair correlation of magnetic susceptibilities and amounts of dithionite-extractable iron (Figure 11a). However, there is also an unexpected connection of calcium carbonate contents and susceptibilities, with

higher contents of calcium carbonate corresponding to higher susceptibilities (Figure 11b). This suggests that higher susceptibilities do not only result from pedogenesis, which should be connected with leaching of $CaCO_3$. That such leaching takes place is suggested by an inverse connection of calcium carbonate and extractable iron contents (Figure 11c).

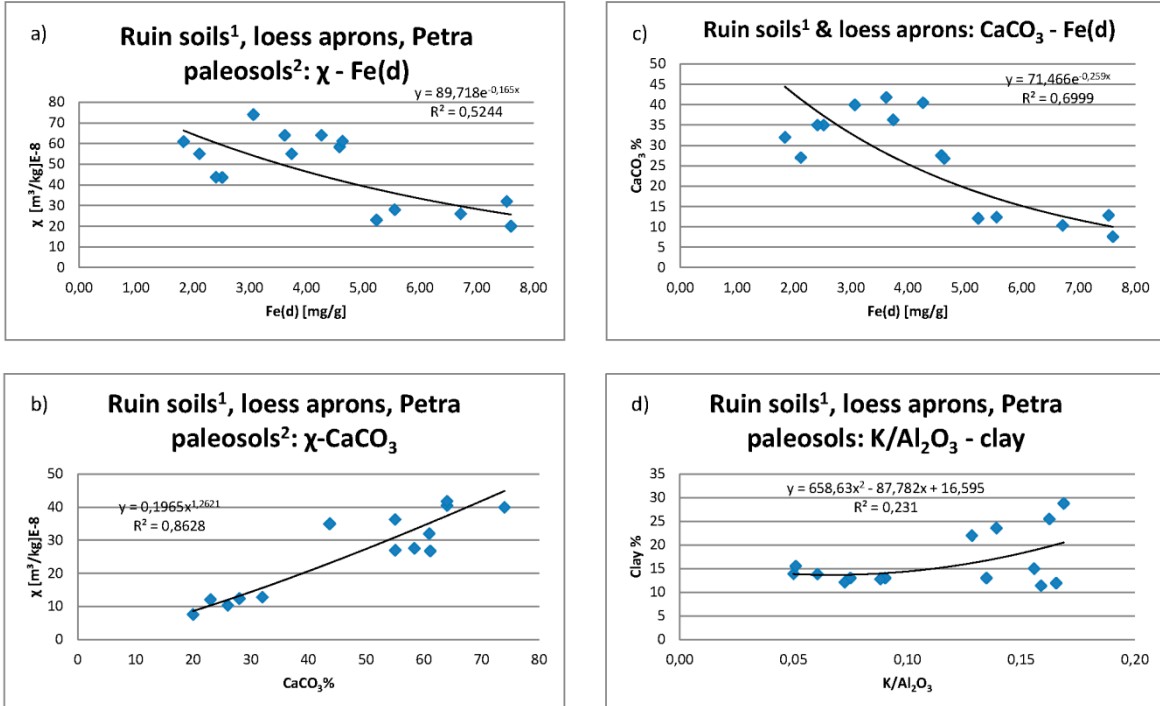

$^1$Except the monastery of Jabal Haroun. $^2$Paleosols include Abu Suwwan and Shkarat Msaied.

**Figure 11.** (**a**) Correlation of magnetic susceptibility $\chi$ and dithionite-extractable iron Fe(d) for the ruin soils, loessial aprons, and paleosols; (**b**) connection of magnetic susceptibility $\chi$ with calcium carbonate content for the ruin soils, loess aprons, and paleosols; (**c**) plot of $CaCO_3$-content against dithionite-extractable iron for the ruin soils and loessial aprons; and (**d**) relation of clay content and $K/Al_2O_3$-ratio for the ruin soils, loess aprons, and paleosols.

Clay and calcium carbonate contents are not connected and clay contents do not show any significant correlation with chemical weathering indices. This suggests that pedogenesis was not strong enough to alter particle sizes. Only the $K/Al_2O_3$-ratio, which indicates formation of secondary minerals accompanied with feldspars (illitization), shows some weak relation to clay contents (Figure 11d).

Plotting the susceptibilities of the dust storms against the ruin soils, it can be observed that they move in a very similar range, with current dust storms often exhibiting higher susceptibilities than the associated ruin soils (Figure 12). The magnetic signal in the ruin soils thus seems to be connected with dust properties and, therefore with calcium carbonate.

### 3.7. Principal Component Analysis (PCA) Based on Grain Sizes and Magnetic Susceptibilities

In order to describe similarities between the samples according to the nine grain-size classes, we used a Principle Component Analysis (PCA) based on the nine grain size classes, magnetic susceptibilities, and statistical parameters related to particle-size distribution, such as the main modes and mean grain sizes. The resulting PCA biplot is shown in Figure 13. Samples are presented in a 2-dimensional Euclidean space with similar samples located close to each other and samples with different properties located in relative larger distance.

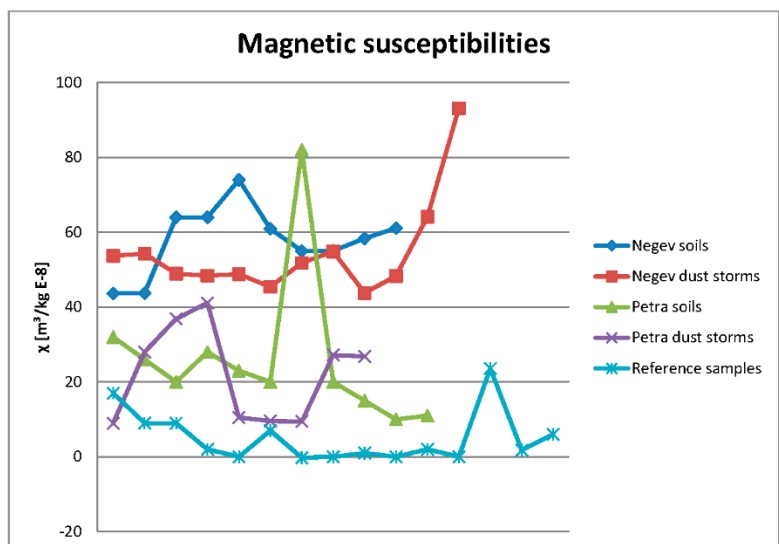

**Figure 12.** Variations of magnetic susceptibilities in the soils, sediments gathered in dust collectors (dust storms), and reference samples of the investigation regions. The x-axis represents individual samples of the respective categories (not numbered; see Table A5 in Appendix C for respective results).

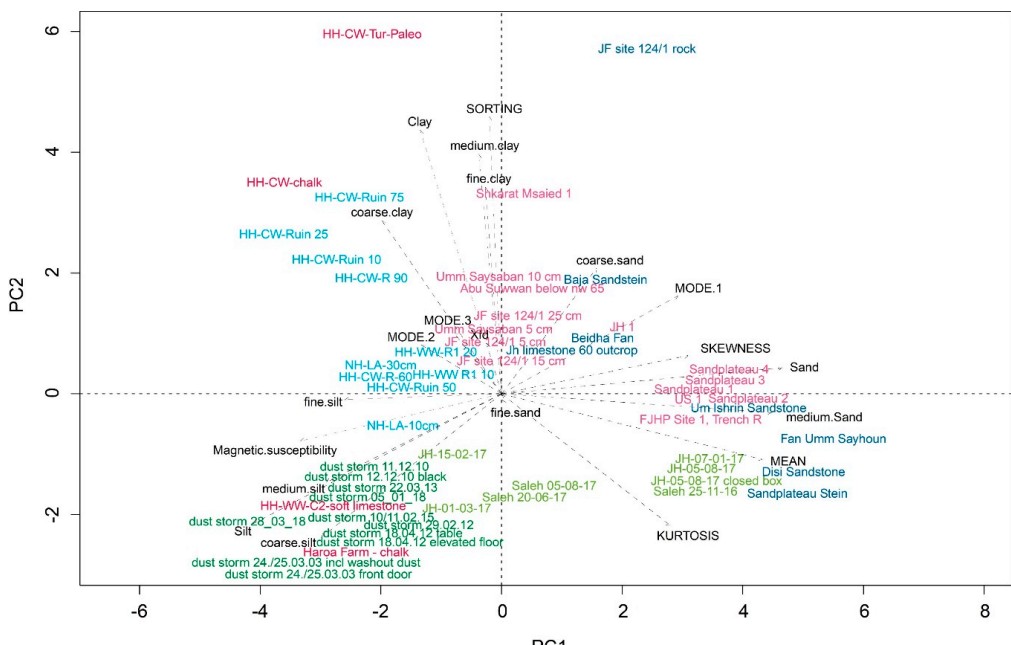

**Figure 13.** PCA biplot based on the nine grain size classes, magnetic susceptibilities, and statistical parameters related to particle-size distribution, such as the main modes and mean grain sizes. The relative locations of the samples reflect their respective similarity with regard to the first and second principal component. The parameters that are most relevant for this distribution are shown in black, along with dashed lines referring to their relative spatial direction, with regard to the principal components.

Three groups of samples can be discerned. The first group consists of the sandstones near Petra, the fans, most of the current dust samples from the Petra regions, and sample FJHP Site 1, trench R from the cover of the monastery on Jabal Haroun, which apparently contains a significant proportion of mud mortar. The second group with negative values of PC 1 and 2 are the Negev dust storms, the loess aprons, and the dust gathered during the snowstorm (15 February 2017) and wet deposition (1 March 2017) near Petra. The main parameters for their distribution are the silt contents. The third

group of samples are the ruin soils of both investigation regions and the paleosol of the loess apron, which cluster between the first two groups.

We used a non-parametric random forest modelling approach [76] to test our hypotheses. First, we modelled the assumed type of deposit (dust collected during current storms, reference samples, and soils) for the Negev samples. The model has an error rate of 16%. Dust storms and soils could be classified correctly in 100% of the cases.

Second, we modelled the assumed type of deposit (dust gathered during current storms, reference samples, and soils) for the Petra samples. The model has an error rate of 10%. Dust storm could be classified correctly in 100% of the cases. The soils were correctly classified 13 times and once assigned to references.

Summing up, these results suggest that dust storms and soils differ. As already seen in the general properties and end-member analysis, the ruin soils seem to be a mixture of reference samples and dust storms. This is true for both investigations regions and leads to an unexpected similarity of the composition of the ruin soils in general, despite the differences, e.g., in individual particle size distributions or potential sediment sources.

## 4. Discussion

### 4.1. Current Dust Storms Sediments vs. Ruins Soils

It seems that not only the composition of the dust storms, but also deposition processes, play a role for the formation of the ruin soils. This might be true for the Pleistocene too, as the loess apron samples from Nahal HaRo'a are very similar to the ruin soils. In this context, differences of pedogenic iron contents and magnetic properties, as well as contents of $CaCO_3$, suggest that the ruin soils do not resemble re-deposited Pleistocene loess from the vicinity—at least not from the aprons that we analyzed. Based on their observation of a haboob dust storm, [77] suggested that part of the currently suspended and deposited dust in Israel is recycled Pleistocene loess, which is released due to agriculture in the northwestern Negev. While this might be true for current dust storms, it cannot be confirmed for the ruin soils that we studied—despite the very similar particle size distributions of Pleistocene loess and Holocene ruin soils.

Our dust sampler, which was continuously open in the Petra region and sampled from November 2016–August 2017, yielded a total of 135 g dust, which would result in a deposition of 843.75 g/m$^2$. Without the event on 5 August 2017, which was possibly affected by the occurrence of a dust devil, 79 g were collected, which would result in 493.75 g/m$^2$. In comparison, the estimate of annual dust deposition in the triclinium of Jabal Farasha, which is the highest and probably most reliable figure, yields only 150 g/m$^2$ a$^{-1}$. The best estimate from the ruin in the Central Wadi in the Negev, 0.13–0.2 mm/a or 140–210 g/m$^2$ a$^{-1}$, indicates that similar rates could be present in the Negev. These values are in good agreement with the estimates of [9], who found average rates of 0.2–0.3 mm/a of dust accretion in ancient ruins.

These comparatively small sedimentation rates mean that only a part of the dust that can trapped with dry standard marble dust collectors, namely ~30%, is fixed in the ruins. Therefore, it seems unsurprising that the properties of current dust storm sediments and ruin soils are different. In this context, [9] suggested that archaeological ruins represent very effective dust traps due to their rough surface. This is supported by [21], who showed that dust deposition happens preferentially on the lee sides of slopes, or in areas where obstacles, such as the remains of walls, slow down wind speed. However, [9] argued that natural dust traps, such as stone covers, should be equally effective in harvesting dust with regard surface roughness, but did apparently not store much dust during the Holocene. This means that other factors must play a role in the ruins.

## 4.2. Sources of Sediments in the Ruin Soils

Gerson and Amit [9] proposed that a contribution of debris and artificial sources could be one option to explain the striking absence of Holocene sediments in natural dust traps compared to the archaeological ruins. In this context, all ruin soils show elevated amounts of small stones and coarse sand, probably from disintegrating stone architecture, but only the soil on the monastery of Jabal Haroun was found affected by artificial substrates, such as mud mortar. The latter can well be identified and distinguished from the dust-derived sediment. It is reflected by lower contents of calcium carbonate (7% vs. 8%–15% in the ruin soils) and a different particle size distribution, in particular by higher contents of medium sand (31%–47% vs. 14%–18% in the ruin soils). The ruins of Umm Saysaban and the triclinium on Jabal Farasha in the Petra region, as well as the two hilltop ruins of the Negev, apparently contain mostly wind-blown sediments. They show a similar composition and no changes in the individual profiles, pointing to rather constant deposition and soil formation from probably uniform (allochthonous) parent material.

The number of samples, 57, that were fed into the statistical model of EMMAgeo is rather small. However, since it significantly exceeds the amount of modeled particle size classes, the approach is feasible. Small inconsistencies, such as the minimal presence of medium sand that was suggested to be associated with the end-member consisting of fine silt, likely represent statistical errors. Additional samples might lead to better representativity of the model, but some clues can be deduced already now.

The ruin soils seem to reflect a complex mixture of local and remote sources, but do not represent the remains of artificial substrates. This is confirmed by principle component analysis (PCA). Local sources matter, as indicated by the significant presence of eroded clastic bedrock sediment in the Petra region, which differs from a comparative lack of carbonate detritus in the Negev sections. However, despite the different local settings and probably also divergent dust sources, PCA suggests that the complex composition of the ruins soils in both investigation regions is rather similar. This suggests that factors other than surface roughness or debris must be considered to explain the relatively strong dust accumulation in the ruins, in contrast to the weak deposition in potential natural traps.

## 4.3. Vegetation, Biocrusts, and Clast Pavements

Gerson and Amit [9] proposed that vegetation could play a very important role for the accumulation of dust. In this context, some clues might be derived from the Early Holocene hilltop paleosols that were preserved below the pre-pottery Neolithic ruins in the Petra region. These contain higher amounts of medium sand than the ruin soils, which probably mark the contribution of weathering sandstones in the Petra region. Accordingly, their Fe(d)-Fe(d/t) ratio shows a more scattered relationship. Calcium carbonate contents, however, are high, even higher than in the ruin soils. As there are no indications for plaster or other artificial additions, it seems likely that the $CaCO_3$ in the soils from the Early Holocene is derived from dust sources, too. In this context, secondary carbonate deposits at Wadi Namaleh near Shkarat Masaied were concluded to have calcium derived from wind-blown sources [78]. According to [79], enrichment of secondary calcium carbonate in soils of the Petra region is connected with Ca-rich dust and can be interpreted a marker of (past) elevated moisture. Soil $CaCO_3$ nodules, such as those present in the paleosol below the ruin of Shkarat Msaied, could have formed rather rapidly during some decades [80].

Compared to the current natural hilltop soil on the sandstone plateau near Petra, the paleosols are more weathered and contain a larger share of allochthonous material, i.e., dust. One possible explanation could be denser vegetation cover than today, which would be expected from environmental reconstructions suggesting moister conditions in southern Jordan during the Early Holocene [81]. Similarly, Baruch and Goring-Morris [82] reported that the Negev was covered by more mesic and denser woody vegetation during the Terminal Pleistocene, due to higher humidity. Denser vegetation would more efficiently trap sand particles [83]. This probably includes some sand from weathering of the underlying sandstones in the Petra region. As described in Appendix A, natural hilltop soil in the Petra region was only found in areas of the sandstone plateaus where Juniperus trees and some grasses

grew. This underlines the potential importance of the vegetation for soil development. In this context, it should be mentioned that all ruin soils were associated with some small, scattered shrubs (<30 cm) and grasses. Vegetation cover was sparse (perhaps covering 30% of the ruins surface), but present.

Weathering intensity indicators in the ruin soils point to stronger weathering than in the Early Holocene paleosols, although the Middle and Late Holocene are not expected to be characterized by moister conditions. This indicates that the mechanism of dust trapping in the ruin soils was different from the Early Holocene paleosols. As outlined by [83], convincing evidence of undercanopy dust enrichment still has to be published, but biocrusts are known to trap and fix dust <50 μm [84]. The answer for the divergent properties of the ruin soils could therefore be connected with the presence of biocrusts. The ruins not only reduce or prevent runoff, but might also retain rainwater to some degree and show little vegetation cover. Therefore, they could offer ideal conditions for biocrusts.

Kidron [83] outlined how different forms of biocrust will be established dependent on moisture availability. In addition, Kidron et al. [85] showed how crust might deteriorate and be eroded under drought conditions. The most frequent cause of crust mortality, in this context, was found to be burial by sand. Once crusts died, they lose their resistance to wind erosion, and prolonged droughts can lead to the erosion of previously stable landscapes. As there are no indications of hiati in the ruin sediments, it seems possible that the conditions provided by the ruins protected their crusts from being eroded during drought periods.

A first survey in the Petra region found biocrusts in some areas, but not everywhere. At least on the soil cover of the monastery on Jabal Haroun, biocrusts could not be confirmed [86]. This might be due to disturbances during the excavation, but suggests that other factors could be involved in the accretion of fines in the ruin soils. These could be clast pavements. McFadden et al. [87,88] proposed that aeolian infall could be trapped by clasts on the surface, such as stones and pottery that are present as dense cover on nearly all studied ruin soils (see Appendix A). Turk et al. [53,89] showed that vesicular layers trap air due to the crust-sealed surfaces, which expands when water infiltrates. Microorganisms might contribute to air expansion by gas emissions [89]. This expansion forces clasts to migrate upwards [53], maintaining them at the surface, and might take place under various degrees of clast coverage [90]. In this context, an enrichment of dust-derived calcium carbonate and iron oxides in the fine sediments accumulating under desert pavements, as observed by [88], seems to match our results from the ruin soils. Although McFadden et al. [88] did not observe an association of desert pavements and biocrusts in the Cima volcanic field, Turk [89] found that clast pavements and vesicular layers can be associated with biological and physical crusts.

Finally, the relative paucity of vegetation and the remote location of the hilltop ruins could have reduced disturbances, e.g., by keeping grazing animals to a minimum. Kidron et al. [85] showed that well-nourished crusts are able to fix sediments better. Therefore, the dust-trapping capacity of the ruin soils might, to some degree, also be connected with the weather.

*4.4. The Role of Climate*

Ganor [73] reported higher amounts of clay in dust settling with precipitation in Israel, but Kidron et al. [21] suggested that this might be an artifact created by the use of wet dust traps. Our data from the Negev does not indicate that dust deposition depends on rainfall or wind speed, but this could be due to the position of the dust collector which (except in one case) was protected from rain. In the Petra region, two dust samples that could be obtained during rain and snow are characterized by much higher contents of silt and $CaCO_3$. End-member modeling suggests that this is connected with the presence of a fine silt fraction that might represent long-range transport. The snow sample in particular seems characterized by this end-member. Although two samples do not yet represent a large database, they let it seem possible that more frequent rains and especially snow could lead to the deposition of significantly higher amounts of silt and $CaCO_3$.

To the best of our knowledge, there are so far no studies concerned with the possible importance of snow for dust deposition. However, it has been shown that snow can contain dust and act as agent of

sedimentation [91,92]. Our data from Petra let it seem possible that it might be an effective catalysator of dust deposition. In addition, snowfall in southern Jordan usually does not lead to strong runoff and erosion, but gentle water infiltration into the soil, providing maximum water storage. In this context, Lucke [50] suggested that the formation of paleosols and cemented valley fills in the Petra region at the end of the Pleistocence was connected with the common occurrence of snow. If frequent snowfalls took place in the Negev during the Pleistocene, this might be one explanation for high loess deposition as snow could enhance dust accretion in three aspects as follows:

- Snow would lead to gentle infiltration with maximum soil moisture, providing optimal conditions for biocrusts and vegetation;
- Snow would mostly be associated with very limited runoff and subsequent erosion;
- Snow could be effective in catching dust particles from the air. This hypothesis is based on the character of one snow storm in the Petra region on 15 February 2017 and needs further testing.

### 4.5. Ruin Soils as Potential Environmental Archives

Our study investigated a limited number of samples from 5 ruin soils and faces insecurities with regard to the dating. However, despite the still limited data, the general similarity of the ruin soils, with regard to their composition as complex mixtures of local and remote sources, allows for some first interpretations regarding environmental changes that might have been recorded by the sediments.

Comparing the two studied hilltop ruins in the Negev, it is interesting to note that the sediments at the bottom of the profiles are associated with similar, and significantly higher, fractions of fine sand than present in the upper part of the profile in the central wadi. In the profile in the western wadi, there is no upper section characterized by higher silt levels, which could be connected with the relatively small wall remains. They may simply have filled up; calculating with an estimated rate of 0.14 mm/a, sedimentation might have stopped or been greatly reduced after ~1500 years.

Although precise dating remains a challenge, the available ages from the archaeological context and pottery sherds that were associated with the ruins could suggest that the Bronze and Iron Age were associated with elevated amounts of fine sand deposition. In the context of the above-mentioned role of vegetation for the trapping of sands, this could be an indication for denser vegetation cover and/or more frequent winds capable to move sand grains. As strong winds are typically associated with cyclones travelling through the Negev, they could have been connected with elevated rainfall leading to more vegetation [21]. The coincidence of this sediment type with pottery as marker of human activity could be interpreted as an indication of moister conditions in the Negev during this period. The last 220 years, in contrast, might be associated with a marked increase of aeolian deposition of silt, since estimated sedimentation rates in the structure in the Central Wadi seem to rise sharply. If processes similar to those found by McFadden et al. [88] for the desert pavements in the Cima volcanic fields occurred in the Negev, this could indicate a marked increase of dust transport in the Negev during the late Ottoman period.

Further studies are needed to corroborate this suggestion, but it shows the potential of the ruin soils to serve as environmental archives. If the clasts covering the ruin soils play a similar role for catching dust as the stones of desert pavements, it seems possible that it is not the composition of the surface layer, but variations of dust influx that are decisive for changes in the deposited sediment [88].

## 5. Conclusions

The similarity of the complex composition of the ruin soils in the Negev and Petra regions, despite different bedrocks and probably varying dust sources [93], points to similar processes of sediment fixation and portrays the potential and significance of the ruin soils as environmental archives.

Our study of past and current dust deposition in archaeological hilltop ruins in the southern Levant shows that ruin soils can be effective traps for Holocene and current dustfall. Dust was found to be continuously deposited after the abandonment of the ancient structures. The importance of dust

for the development of these profiles is indicated by the presence of V-horizons, which could not be observed in natural and currently forming soils. With the exception of the soil on the monastery of Jabal Haroun, no traces of archaeological substrates were identified in the ruin soils.

Dust-derived sediments in the ruins seem to represent a complex mixture of local sand and clay, intermediate-range coarse silt and fine sand, and remote fine silt. The ruin soils in both study areas show similarly complex compositions, but differ from the sediments that can be gathered by dry marble dust collectors during current dust storms. Based on estimates of mean annual dust accretion rates in the ruins, in comparison with current dust collection rates, we calculate that only ~30% of the dust that can be trapped in dry marble collectors is fixed in the soil. This fixation seems connected with the presence of vegetation, crusts, and/or clast pavements.

The ruins seem to be effective traps for silt-sized and smaller aeolian sediment. The dust preservation could be due to sheltering strong winds, lee effects provided by wall remains, improved retention of rainwater in relation to open areas, and reduced runoff. Compared with paleosols from the Early Holocene in the Petra region, which probably formed during moister conditions, the ruin soils are characterized by a higher share of silt and less sand. Sand seems to be effectively trapped by vegetation, which could have been denser during the Early Holocene. Denser vegetation might also have led to a higher contribution of sandstone bedrock weathering to soil formation in the Petra region. Such increased contributions of sandstone detritus to soil formation during the Early Holocene could explain why the (younger) ruin soils in the Petra region appear more intensely weathered than the paleosols. Extractable iron oxides suggest that the ruin soils gathered a relatively high and constant share of pre-weathered iron during their development, or were subject to rather stable in-situ weathering.

Vegetation on the ruins is sparse, with respective reduced competition for light. Disturbances due to, e.g., grazing animals are minimal. The lack of disturbances offers good conditions for biocrust development that are effective in trapping silt and clay sized dust. It could not be ascertained to which degree biocrusts were present on the current surface, but no traces of buried biocrusts were observed. Crusts were present on all ruin soils and, in one case, it could be confirmed that it was a physical and not biological crust. In this context, nearly all the ruins showed clast covers of varying density at their surface. This suggests that processes similar to the formation of sediment bodies below clast pavements (e.g., desert pavements) contribute to the genesis of the ruin soils.

There were no indications that precipitation played a role for the composition of dust in the Negev, which could be related to a rain-protected location of the sampler. In contrast, rain and especially snow in the Petra region were characterized by very different properties of deposited sediment than dry storms. The sample collected with snow was especially large and showed a particle size distribution that was similar to the ruin soils. It seems possible that snow catalyzes dust deposition. Snowfall in the Petra region was associated with minimal runoff and good water infiltration into the soil, providing optimal conditions for vegetation and biocrust development, along with reduced erosion. During colder and moister times of the Pleistocene, snow was probably more abundant than during the Holocene and may have been an important driver for extensive deposition and preservation of loess in the highlands of the southern Levant.

**Author Contributions:** Conceptualization, B.L.; Data curation, B.L.; Formal analysis, B.L.; Funding acquisition, B.L. and H.J.B.; Investigation, B.L., H.J.B., N.A.-J., K.D., S.L., N.P., P.J.R., R.B., T.E.-G., and P.K.; Methodology, B.L., J.R., K.V., N.A.-J., K.D., S.L., N.P., and P.J.R.; Project administration, B.L.; Resources, B.L. and R.B.; Software, K.V.; Supervision, B.L.; Validation, B.L., J.R. and K.V.; Visualization, B.L.; Writing – original draft, B.L.; Writing – review & editing, B.L., J.R., K.V., H.J.B., N.A.-J., K.D., S.L., N.P., P.J.R., R.B., T.E.-G., and P.K.

**Funding:** This research was funded by the German Research Foundation (DFG), grant number LU 1552/2-1.

**Acknowledgments:** We gratefully acknowledge the financial support of the German Research Foundation (DFG) which made this work possible. We would like to thank the Department of Antiquities of Jordan and the Israel Antiquities Authority for their permission and support to study the sediments of archaeological sites. As well, we are indebted to Ulrich Hübner and the Natural History Society of Nuremberg for their support with sampling Umm Saysaban, and to Moritz Kinzel and the Shkarat Msaeid excavation team. We are grateful to two unknown

reviewers for a thorough check of the manuscript, and to Amir Sandler for a critical evaluation of a manuscript draft. Last but not least, we would like to thank Saleh Suleiman and Ghassem Mohammed for their indispensable help with collecting dust samples on Jabal Haroun.

**Conflicts of Interest:** The authors declare no conflict of interest. The funders had no role in the design of the study; in the collection, analyses, or interpretation of data; in the writing of the manuscript, or in the decision to publish the results.

## Appendix A. Detailed Description of Sampled Sites

*Appendix A.1. Sampled Profiles in the Negev*

The archaeological site of Horvath Haluqim consists mainly of the remains of a large oval building that was interpreted by [94] as a small Iron Age fortress with casemate walls. It is situated near the valley of Nahal HaRo'a, at the crossroads of important routes through the Negev, in the past and in the present (today roads 204 and 40, connecting Beer Sheva, Dimona, Mitzpe Ramon, and Eilat, pass the site). Three small tributaries discharge near the above-mentioned large oval building into the main wadi. In the valleys of the tributaries, numerous remains of ancient terraces were preserved, as well as the remains of a Roman watchtower, a hilltop cairn, and the ruins of some other round and rectangular buildings. As well, conduit channels apparently led runoff water into open basins, which are now mostly filled by sediment. Except the fills in the archaeological structures and in the bottom of the wadis, the slopes are devoid of soil cover and hard limestone is exposed. At some locations paleosol units of a continental phase during the Turonian [47] crop out and comprise reference sites for local sediment sources. Two hilltop ruins were sampled.

Appendix A.1.1. Western Wadi: Hilltop Ruin (Samples HH-WW R1)

At the westernmost of the three valleys, we dug a sondage into the soil cover of a ruin located on an adjacent summit (Figure 4a), which probably represents a Bronze Age tumulus grave (N 30.89151, E 34.79909). The ruin was not yet excavated and probably belonged to a cluster of tumuli that may have marked ancient routes and/or territorial claims. Although the cairn had largely collapsed, and was possibly already looted and destroyed during antiquity, approximately 20 cm of sediment cover had completely filled the remains of a wall that once surrounded the monument. In about 15 cm depth, non-diagnostic pottery of the 'Negebite' type was encountered and submitted to dating by thermoluminiscence (TL). The building had been constructed with hard limestone and there were no indications of plaster, mud mortar, or mudbrick remains.

A few scattered bushes < 30 cm height and patches of grass grew in the ruin. The current surface was covered by many small, subangular limestone fragments and a thin crust (~1 mm). It could not be determined whether the crust was physical or biological. The sondage showed no discernible layering. Regarding diagnostic horizons, a V-C profile was present and the top V-horizon, characterized by vesicular properties, was up to 5 cm thick. A crumbly, friable, and platy structure of small aggregates with many large pores in the C-horizon below suggests a slow build-up involving the formation of vesicular layers during aeolian sedimentation. The vesicular layers may later have collapsed to the platy structure. Hiati or layer changes could not be observed, but it seemed possible that the order of vesicular pore remains might roughly indicate a succession of old surfaces. Occasional small roots <1 mm diameter were present throughout the sondage. Scattered over the whole profile, various small stones, apparently broken off collapsed architecture, were present as well. No traces of bioturbation or buried crusts could be observed. Two samples for laboratory analysis were taken from 10 and 20 cm depths. The ruin soil in the hilltop of the western wadi is classified as Calcaric Leptosol (Protic, Siltic), according to [52].

Appendix A.1.2. Central Wadi: Hilltop Ruin (Samples HH-CW Ruin)

In the central wadi, the ruins of a small group of round structures, which were not yet excavated, cluster on the middle slope near the remains of terraces in the central part of the valley. They seem

to represent remains of houses or huts possibly connected with an agricultural use of the terraces. One circular structure was located on a small spur projecting into the valley, on a slightly elevated position where fluvial deposition appeared unlikely (N 30.88948, E 34.80015). A fallen big stone in the center of this ruin suggests that the building may have been sub-structured, e.g., for huts at the sides and an open space in the middle. A few scattered bushes <30 cm in height and patches of grass grew in the ruin.

A sondage was excavated in the center of the circular wall remains of the ruin (Figure 4b). The structure had filled with a minimum of 60 cm sediment and the spur turned out to be a small loess apron with buried paleosol containing secondary carbonate nodules, similar to the ones described by [8] from nearby Nahal HaRo'a. In difference to the first ruin profile, some layer changes could be observed. The surface was mostly free of stones and the uppermost layer (up to 5 cm thick) showed a loose, powdery consistency and a single-grain structure. It was not sampled, as we suspected, that relatively recent anthropogenic activities, such as Bedouin campfires, could be connected with these properties of the top layer. The removal of stones is typical for fireplaces and surface disruption due to trampling could be indicated by the structure of the top layer.

The following V-horizon (V1: 5–15 cm) was yellowish-brown and of platy structure, possibly from collapsed past vesicles. Under a diffuse horizontal boundary, a crumbly, friable, and less platy V-horizon with a slight increase of fine sand and various small stones followed (V2: 15–35 cm), and then a layer with many subangular pebbles in 35–40 cm. The latter might represent a collapse event of the structure or a former clast-covered surface analogous to buried desert pavements ([64], 2012). Charred materials were recovered in ~7, 14, and 20 cm depths, which could be used for $^{14}$C dating. Under an undulating boundary below the pebble-rich layer, yellowish-brown silt of a friable, slightly platy, C-horizon was encountered, which contained complete snail shells and some non-diagnostic pottery of the Negebite type in 50 cm depth. These items were perhaps deposited during occupation of the site and a pottery sherd was submitted for TL-dating.

A diffuse boundary to the bottom makes it difficult to decide where the geoarchaeological sediments end and where the buried loess apron begins. The substrate post-dating the erection of the structure might be mirrored by the occurrence of various small stones, which seem to concentrate in 70 cm depth, possibly indicating a former land surface. Below 70 cm, stones disappear and secondary carbonates are present. As the latter are typical for the Pleistocene loess stratigraphy of the area [8], it seems safe to assume that the archaeological sediment ends at 70 cm depth. We continued to excavate until a depth of 95 cm, which produced some large $CaCO_3$-nodules (loesskindl). The loess paleosol seemed characterized by darker color and more compact structure with increasing depth, apparently representing a Bwk-horizon.

Various small stones could be observed throughout the profile, apparently broken off the collapsed architecture. Occasional small roots < 1 mm in diameter were present throughout the sondage. No traces of bioturbation or buried biocrusts could be discerned. Samples for sedimentological analysis were taken at 10, 25, 50, 60, 75, and 90 cm depths. Samples for dating by Optical Stimulated Luminiscence (OSL) were taken from 25, 55, and 90 cm depths. The order of diagnostic horizons is as follows: A-V1-V2-C-2Bwk. The ruin soil at the central wadi is classified as Protic Regosol (Siltic) according to [52]. The loessial paleosol that was buried by the structure is classified as Cambic Calcisol (Siltic).

Appendix A.1.3. Loessial Apron in Nahal HaRo'a

A loessial apron in the nearby Nahal HaRo'a (N 30.30140, E 34.84296) had been exposed by a gully (Figure 4c). Various bushes <30 cm, occasional small stones, and patches of grass were present at the surface. A crust ~1 mm thick occurred on the current surface. It could not be determined whether it formed as biological or physical crust. Occasional small roots <1 mm in diameter were present throughout the profile. No traces of bioturbation or buried biocrusts could be discerned. Despite a friable and vesicular structure in the upper part of the profile, the apron material was much more

compact than the archaeological sediments. The upper 25 cm of the section represent a V-horizon characterized by yellow color (10 YR 8/6), while brownish-yellow color (10 YR 6/6) and secondary carbonates appeared in ~25 cm depth and below, suggesting a layer change and the presence of a buried, probably truncated, 2Bwk horizon. Truncation is supported by the presence of a few small subangular stones at the layer change, probably marking an old land surface, while stones are otherwise mostly absent from the profile. The soil developed in the upper 25 cm is classified as Protic Regosol (Siltic) and the buried paleosol as Cambic Calcisol (Siltic), according to [52]. Samples for laboratory analysis were taken at 10 and 30 cm depths.

Appendix A.1.4. Negev Reference Samples: Limestones, Chalk, and Turonian Paleosol

Various reference samples were taken from local outcrops that could have provided sediment to the archaeological structures. These include samples of soft limestone (N 30.89068, E 34.79859), hard limestone (N 30.89068, E 34.79859), chalk (N 30.88943, E 34.79994 from an outcrop in the central wadi, and N 30.89634, E 34.84516 from an outcrop in Nahal HaRo'a), and from the Turonian paleosol (N 30.88943, E 34.79994). These slope deposits might have been fluvially mobilized and sediments along wadis could have blown into nearby archaeological ruins.

*Appendix A.2. Sampled Profiles in the Petra Region*

Jabal Haroun is an area of religious significance where Aaron/Haroun, the brother of Biblical prophet Moses, is understood to be buried. Today there is an Islamic shrine on the mountaintop that is frequently visited by pilgrims and tourists. During antiquity, a chapel was built on the mountaintop, while a broad plateau was used for worship in a Nabatean pagan sanctuary that was later built over by a Christian monastery [95]. The place had been mentioned as an agriculturally productive area in the Petra papyri from Late Antiquity [42,95,96]. The Finnish Jabal Haroun Project (FJHP) conducted an extensive survey of archaeological sites and the remains of agricultural structures of this region, including collection of off-site material scatters and the mapping of ancient terrace systems. As well, the ruins of the Byzantine monastery were excavated.

Appendix A.2.1. The Monastery of Jabal Haroun

In the monastery, two courtyards or open spaces had been located that were characterized by a relatively limited amount of rubble and fallen architecture. Here, remains of the surrounding walls would have protected the area from fluvial deposition from the sandstone plateau. However, since mortar made from mud had been used to close gaps in the walls during their construction ([95] and own observations), caution had to be exercised with sampling the debris of the monastery. In order to avoid the mud mortar, one sample was carefully collected from the upper 10 cm of the debris cover of the northern courtyard. It seemed to represent a V-horizon and was characterized by a vesicular, crumbly, and friable structure and a strong reaction to HCl (sample *JH 1*, Figure 5a, N 30.31734, E 35.40418). Neither traces of bioturbation or buried biocrusts, nor indications of plaster or mudbrick remains, could be observed. Many small subangular sandstones were present throughout the profile. A few scattered bushes <30 cm in height and patches of grass grew in the ruin. The current surface was covered by many small sandstone fragments and occasional small roots <1 mm in diameter were present throughout the sondage. A crust ~1 mm thick occurred on the current surface. Since no traces of a biological soil crust could be identified with a magnifying glass [86], it seems to be a physical crust.

A second sample of the cover of the monastery taken during its excavation from the south-western corner of the building was provided by the Finnish Jabal Haroun Project (sample FJHP site 1, trench R, N 30.3171107, E 35.4038850). The soils of the sediments covering the monastery of Jabal Haroun exhibit V-C profiles and are classified as Protic Arenosols (Aridic, Aeolic), according to [52].

Appendix A.2.2. The Triclinium on Jabal Farasha

A Nabatean triclinium on the mountaintop of Jabal Farasha, facing Jabal Haroun in the south-west, was apparently used for cultic purposes. It has not yet been excavated, but according to surveys of material culture, activities lasted from the 1st century BCE till the 2nd century CE. These are assumed to have terminated due to the prohibition of private cults after the Roman annexation [96]. There is a foundation wall built of sandstone and no indications of plaster, mud mortar, or mudbrick remains could be observed. The area between the triclina seems to have remained open, most likely consisting of cleared smoothed sandstone rock during antiquity. This space is free of larger stones and covered by about 25 cm of fine sediment, which must have been transported there by wind since the position on the very top of the mountain leaves no possibility of fluvial deposition. The walls were not yet completely covered by sediments. The ruin soil covering the former open space of the triclinium has a crumbly, friable, and subangular aggregate structure throughout, with significant silt content and a strong reaction to HCl, but also a large portion of sand. Small voids seem to represent the remains of pores of a vesicular layer and are present in the upper 10 cm. This V-horizon is covered by a thin crust. It could not be determined whether it formed as physical or biological crust. No traces of bioturbation or buried crusts could be observed, and no stones were present throughout the profile, but occasional pieces of pottery, particularly at the surface and in the upper 15 cm. All were dated typologically to the Nabatean period. A few scattered bushes <30 cm in height and patches of grass grew in the ruin. The current surface was densely covered by various small sandstone fragments and pottery sherds.

The sandstone rock exposed at the bottom of the sondage was not smooth anymore, but appeared weathered, falling apart into small plates with some loose material in-between small cracks. It was possible to collect small rock plates just by hand. This suggests that some bedrock weathering took place since the site went out of use. Although the surrounding walls were made of the local sandstone, they did not look weathered. If sand was released from in-situ rock weathering, it was therefore probably mainly from the bottom, connected with soil formation processes possibly taking place while the ruin was covered with sediments. The surrounding walls probably withheld rainwater to some degree.

Three soil samples and one of the underlying rock were collected (JF site 124/1 5cm, 15 cm, 25 cm, rock; N 30.30445, E 35.40141, Figure 5c) and an OSL-sample was extracted as near to the top as seemed possible without contamination from the surface (from approximately 15 cm depth). The ruin soil of Jabal Farasha exhibits a V-C profile and is classified as Calcaric Leptosol (Protic), according to [52].

Appendix A.2.3. Umm Saysaban Hilltop Ruin

Umm Saysaban is one of the rare Bronze Age sites in the Petra area. It was constructed on a high, but relatively flat sandstone plateau north-west of Jabal Haroun, near an escarpment of the rift valley where a temporary waterfall drops down to the Wadi Araba during winter rains. It seems likely that the spectacular place was chosen for defensive purpose and numerous pits and storage jars suggest that the main purpose of the site was storing goods [97,98]. Since the storage was empty, it seems the place was orderly deserted. Umm Saysaban was constructed by large stone slabs that were placed on the narrow side and collapsed later. Most wall stones now lie next to their foundations. Mud mortar was used in the foundation walls, but apparently not on the standing wall stones. This means that the sediment that accumulated next to the fallen wall stones most probably represents aeolian deposits. The excavations revealed that the smoothed sandstone bedrock was the floor of the ancient buildings.

A few scattered bushes of <30 cm in height and patches of grass grew in the ruin. The current surface was covered by many small subangular sandstone fragments. Occasional small roots <1 mm in diameter were present throughout the sondage. The building had been constructed with sandstone and there were no indications of plaster or mudbrick remains. Occasional small subangular sandstone pebbles were present throughout the profile. Two samples were taken from a fresh exposure on the top of the settlement plateau, consisting of approximately 10 cm calcareous and silty fine sand (N 30.34595, E 35.43178, Figure 5b). The upper V-horizon was at least 5 cm thick (sample Umm Saysaban 5 cm)

and showed a friable, crumbly, and vesicular structure. The lower C-horizon (sample Umm Saysaban 10 cm) was slightly compacted and might have settled within a puddle in the remains of the fallen walls. A slightly platy structure and small voids might represent the remains of pores of a vesicular layer. The topsoil is covered by thin crust. It could not be determined whether it formed mainly as a biological or physical crust. No traces of bioturbation or buried biocrusts could be observed. The ruin soil of Umm Saysaban exhibits a V-C profile, and is classified as Calcaric Leptosol (Protic), according to [52].

Appendix A.2.4. Shkarat Msaied Hilltop Ruin

The site of Shkarat Msaeid is situated on a flat sandstone plateau c. 13 km north of Petra near the Wadi Namaleh, at the road from Petra to the Wadi Araba (N 30.44372, E 35.43917). It was a settlement of the Pre-Pottery Neolithic B (PPNB) period, excavated and dated to 9200–7700 BCE [71]. Apart from sandstone, Precambrian igneous rocks are exposed in the area. The upper levels of the sediment cover of the ruins of Shkarat Msaeid were disturbed by farming during antiquity and several large agricultural walls that crossed the area might have preserved the site from erosion. It was cut by gullying to the east and south. An outcrop to the south revealed silt-rich sediments in the contact zone of the sandstone rock to the lower limit of the foundations of the Neolithic site. On the sandstone, a calcareous crust had formed where the rock was in contact with the sediments and secondary carbonates could be found in the silt-rich sediments below the Neolithic walls. The PPNB site apparently buried and preserved a paleosol that might have covered large parts of the area, but is today eroded in the vicinity.

The profile below the Neolithic walls exposed some 60 cm of the silt-rich paleosol (Figure 6a) below the site. The bottom zone in 45–60 cm was rich in $CaCO_3$ and very compact. It represents a 3BCk-horizon that contained secondary calcium carbonate nodules, calcified root channels, and large pores that could represent either remains of root channels or a vesicular layer. The paleosol was free of stones and characterized by a friable, compact, and crumbly structure. In 20–45 cm depth below the walls, the color was more brownish, suggesting that a 3Bw-horizon was present. In 0–20 cm depth, the color turned grayish, which is interpreted as related to human activity, such as deposition of ash before the construction of the site. It seems likely that a former 3A-horizon was truncated. The paleosol below the ruin walls is classified as Cambic Calcisol (Protocalcic), according to [52]. Only the bright silt-rich horizon on top of the sandstone, possibly representing a BCk-horizon partly derived from aeolian deposits, was submitted to sedimentological analysis (sample *Shakarat Msaid 1*).

Above the archeological ruins, the sediments covering the remains of the Neolithic walls reach some 110 cm in size and apparently contain much ash and very fine sediment in the lower part (horizon 2C in 70–110 cm depth). The excavating archaeologists interpret this layer as a large fire that might have destroyed the settlement and led to an enrichment of ash [60], but it seems strange how such amounts could accumulate and why it was largely free of charcoal. A small Dicotyledon charcoal fragment could be separated by flotation and was submitted to [14]C dating (*sample Shakarat Msaid 4*). The upper 70 cm of the sediments covering the ruins, however, seemed rather disturbed by terrace farming during antiquity and recent bulldozing during construction of a road and were, thus, not further studied.

Appendix A.2.5. Abu Suwwan Hilltop Ruin

There is an unexcavated pre-pottery Neolithic B (PPNB) site called Abu Suwwan on Jabal al-Bara, in the rugged sandstone mountains between Jabal Haroun and Umm Saysaban, indicated by many lithics from this period. Remains of terrace walls in the area, tentatively dated to the Nabatean period by various sherds from this time, might be the reason why the site has not been eroded. The terrace builders could have been attracted by remains of fine sediments held in place by the ruins in the otherwise rocky environment. We excavated a sondage into a looter pit (N 30.33064, E 35.42297, Figure 6b).

At ~50 cm depth, a wall built of thin broad stone slabs was encountered, which was very different from the larger stones used for the terrace walls. That these slabs resemble a wall of the Neolithic site is supported by lithics typical for the PPNB that were associated with this wall. In the upper part of the profile, in contrast, there were no such lithics present, but some occasional Nabatean and Iron Age pottery. Below the Neolithic wall, we encountered a silt-rich soil that was much more compact than the sediments on top. The wall was built on this apparent paleosol, which was possibly, to some degree, truncated during construction of the site. The wall remains are covered by a debris layer extending from approximately 20–40 cm in the profile and containing ash, from which we extracted a charcoal sample that was submitted to $^{14}$C-dating. On top, crumbly sandy sediment with many stones covered the site. A few scattered bushes of <30 cm in height and patches of grass grew in the ruin. Occasional small roots <1 mm in diameter and small sandstones were present only in the upper 40 cm of the profile, but not in the paleosol. The paleosol was nearly devoid of stones. Soil structure was friable, crumbly, and of subangular shape throughout the paleosol, which represents a 3BC-horizon. Although each layer was sampled, only the paleosol was analyzed in the laboratory (sample *Abu Suwwan below nw 65 cm*) and a conifer charcoal fragment of the ashy layer dated. The paleosol of Abu Suwwan is classified as Protic Regosol (Aeolic), according to [52].

Appendix A.2.6. Natural Sandstone Plateau Soil

In contrast to our investigation area in the Negev, where no natural soils could be encountered in hilltop positions, there are rather flat sandstone plateaus in the study area near Petra that are sometimes covered by fines. On one such plateau, a 70 cm deep profile was excavated next to a *Juniperus* tree (N 30.41564, E 35.46117). Some bushes <30 cm in height and grass patches were present as well. Omnipresent roots in the profile suggest that soil formation was connected with a role of the vegetation for bedrock weathering and erosion protection (Figure 6c). Apart from a few soft (probably weathered) small sandstones, the profile was free of stones and showed a single-grain structure. Occasional small sandstones were present on the surface. Four samples at 10, 30, 50, and 70 cm were taken, as well as a sandstone rock sample from the bottom of the profile and a sandstone lying on the topsoil. This soil exhibits an A-C profile and is classified as Protic Regosol (Arenic), according to [52].

Appendix A.2.7. Petra Region Reference Samples

Apart from the above-mentioned profiles, various reference sites were sampled in order to tackle the potential contributions of local sources to the studied sediments and the role of the archaeological structures for the accumulation and preservation of sediment. Sandstones of the prevailing geological formations of the Petra area were collected and analyzed as follows: Abu Khushayba (N 30.31734, E 35.40418), Umm Ishrin (N 30.32192, E 35.43099), and Disi (N 30.34763, E 35.45844) sandstones. From a limestone of the Turonian Wadi as Sir formation that is exposed in a horst near Jabal Haroun, another reference sample was taken (N 30.31261, E 35.39468). Two active sediment fans near the villages of Beidha (N 30.37694, E 35.45327) and Umm Sayhoun (N 30.34793, E 35.45814) were sampled.

In addition, remains of a paleosol possibly containing aeolian sediments, and probably dating to the Pleistocene, could be observed near the village Umm Sayhoun (N 30.34090, E 35.45723). It seemed rich in silt and was weakly cemented by $CaCO_3$ (Figure 6d). Nearby remains of a paleochannel with partly cemented polymictic fanglomerate suggest erosion and colluvial re-distribution of a part of the parent material of this potential paleosol, which we tentatively call Umm Sayhoun formation. It contained calcified root channels and large pores that could represent either remains of root channels or vesicular layers. The structure was friable, crumbly, and of subangular aggregates. Stones of various sizes and types, probably derived from the fanglomerate, were present. It could not be decided whether the profile represents a truncated B- or a cemented V- or C-horizon. A sample of the paleosol (*US 1*) was analyzed in the laboratory. The paleosol was classified as Cambic Calcisol (Loamic), according to [52].

## Appendix B. Detailed Description of Methods

*Appendix B.1. Sample Collection*

Soil and sediment profiles were excavated until bedrock or rock debris were met, or until a stratigraphy similar to representative soil profiles in the area was exposed. Profiles were classified according to [52] and horizons described according to [99]. As deviation of the latter, we followed the concept of V-horizons as proposed by [53] for of vesicular layers. Samples were taken from areas of approximately 5 cm size in the center of defined layers or horizons and collected in plastic bags.

*Appendix B.2. Laboratory Analysis*

The samples were air-dried for 72 hours at 40 °C and then dry sieved by 2 mm. All further analyzes were conducted with the fraction <2 mm. For those analyzes where pulverized samples were needed, an agate ring mill type Retsch RS 200 was used to grind and homogenize them.

Dry and moist colors were recorded using the Munsell Soil Color Charts. The pH was determined with a glass electrode (pH-meter 530 by WTW, with electrode InLab 423 by Mettler-Toledo) in distilled water, in a soil:water solution of 1:2.5. Electrical conductivity was measured with a GMH 3410 conductivity meter by Greisinger electronic in a soil:water solution of 1:5. Contents of $CaCO_3$ and organic matter were determined using an Elementar vario EL cube C/N-analyzer in doubles, before and after ignition, for two hours at 500 °C. Ignition removes organic matter and it is assumed that the remaining carbon is bound in $CaCO_3$. For some samples, $CaCO_3$ contents were determined using a "Karbonat-Bombe", a device applying similar principles as the Scheibler-Apparatus [100]. It measures the pressure of released $CO_2$ when treating a sample of 0.70 g with 6n HCl. The Karbonat-Bombe was used for samples such as the rocks, where organic matter was not studied. The $CaCO_3$ contents of the current dust samples were calculated from Ca contents that were determined by total element analysis by XRF, since these samples were mostly very small which limited the range of methods that could be applied to them. For a comparison of results from these three methods, see Appendix C.

Particle sizes were analyzed without removal of $CaCO_3$, but samples were washed with distilled water in order to remove coagulating ions until conductivity fell below 200 μS/cm. The only exception was the paleosol of the Umm Sayhoun formation, where grain sizes were also determined after removing $CaCO_3$ with 10% HCl and washing the sample until conductivity fell below 200 μS/cm in order to check the impact of carbonate cement for particle sizes. After pretreatment, all samples were dispersed with 80 ml sodium hexametaphosphate ($Na_4P_2O_4$). Wet sieving determined the sand fractions according to DIN 19683 [101], while the smaller particles were analyzed with a Sedigraph 5100 (Micromeritics) [102]. Since the Sedigraph requires minimum weights of approximately 8 g (silt and clay fractions), it was not possible to analyze most of the current dust samples as the collected amounts were too small. Therefore, the current dust samples were measured with a Malvern Mastersizer 3000, in water using the Hydro-EV dispersion unit with 3000 rpm. A total of 80 ml sodium hexametaphosphate ($Na_4P_2O_4$) was added before entering a sample and a minimum of 5 mins of ultrasonic was applied (or until shadowing remained constant). The raw laser diffraction values were transformed into particle-size distributions using the Mie scattering model, a refractive index (RI) of 1.54 and adsorption (A) of 0.1. Laser results were mostly calculated as averages of 5 measurements, which showed good reproducibility. When sufficient amounts were available, some samples were analyzed by both particle size methods in order to check the comparability (see Appendix E). The fraction of the very fine sand (125–63 μm) could be calculated for the samples measured by the laser and was determined for selected Sedigraph samples by a dry sieving of the fine sand fraction with a 125 μm sieve after its separation by wet DIN 19683 sieving. Some of the soft rocks were gently crushed by hand, in order to simulate physical disintegration, and the particle sizes of the resulting fines were measured as described above in order determine an estimate of the particle sizes that could be released from the rocks.

For some samples where in-situ soil formation seemed possible, pedogenic oxides were dissolved with sodium dithionite at room temperature, according to Holmgren [103]. Weakly crystallized

pedogenic oxides were extracted in the dark using buffered (pH 3.25) oxalate-solution according to [104]. The iron, aluminum, and manganese contents were measured by ICP-OES (ICAP 6200) and, in case of the oxalate-extraction, silicium was analyzed as well. For total element contents, the loss of ignition was determined by weighing the powdered samples before and after drying, as follows: 1) For 12 h at 105 °C in a cabinet dryer and 2) 12 h at 1030 °C in a muffle furnace, melting samples in Pt-crucibles. Major element oxides and selected trace elements were then measured with an energy-dispersive Spectro XEPOS. Precision and accuracy are generally better than 0.9% (main elements) and 5% (trace elements).

The weight-specific magnetic susceptibility $\chi$ was examined with an Agico MFK1-FA multi-function Kappabridge device using three different frequencies (976 Hz, 3904 Hz and 15616 Hz) and field strengths varying between 10 and 700 A/m. For some samples, an Agico KLY-4S was used with 875 Hz and 300 A/m. In order to make results between the two devices directly comparable, 424 A/m and 976 Hz were used as reference setting of the MFK1-FA. For some of the samples where pedogenic oxides had been extracted, the extracts were dried and measured in the MFK1-FA in order to assess the role of pedogenic iron oxides for magnetic susceptibilities.

*Appendix B.3. Dating*

Samples UBA-38659 to 38661, UBA-38960, and UBA-39002 (see Table A6 in Appendix D) were radiocarbon dated at the [14]CHRONO Centre for Climate, the Environment and Chronology, Queen's University Belfast. The robust charcoal samples (UBA-38659, 38660, and 38661) were pretreated using a standard acid-alkali-acid (AAA) method. The samples were placed in 100 ml beakers (cleaned by baking at 450 °C) and immersed in hydrochloric acid (4%, 30–50 ml). The contents of the beaker were heated on a hotplate (800 °C for 2–3 h). The samples then received subsequent washes with deionized water until neutral. Sodium hydroxide (2%, at 800 °C for 1–2 h) was added to remove humic acids, followed by further rinsing with deionized water until neutral. The acid step was repeated to remove any $CO_2$ absorbed during the NaOH step, rinsed, and dried overnight at 60 °C. The sample of charcoal from the ashy layer at Shkarat Msaid (UBA-39002) appeared to be too fragile for the full AAA so it only received the first acid step. Likewise, the sample of the ashy layer from Abu Suwwan (UBA-38660) had an acid only pretreatment.

The dried samples were weighed into pre-combusted quartz tubes with an excess of copper oxide (CuO), sealed under vacuum, and combusted to carbon dioxide ($CO_2$). The $CO_2$ was converted to graphite on an iron catalyst using the zinc reduction method [105], except for sample UBA-39002 which, because it was slightly smaller, was converted to graphite using the hydrogen reduction method [106]. The graphite produced was analyzed on a 0.5 MV National Electrostatics compact accelerator mass spectrometer (AMS). The sample $^{14}C/^{12}C$ ratio was background corrected using measurements on anthracite and normalized to the HOXII standard (SRM 4990C; National Institute of Standards and Technology). The radiocarbon ages were corrected for isotope fractionation using the AMS measured $^{13}C/^{12}C$ ratios, which accounts for both natural and machine fractionation. The radiocarbon age and one standard deviation were calculated using the Libby half-life of 5568 years, following the methods of [107].

At the Curt-Engelhorn Centre for Archaeometry (CEZA), Mannheim, Germany, the radiocarbon sample MAMS 31135 of charred pottery temper was pretreated with 4% hydrochloric acid (HCl) to remove carbonates, 0.4% Sodiumhydroxide (NaOH) to remove the soluble humic acids, followed by another acid step to remove modern carbon attracted by the pretreatment with the base. Organics and carbonates were removed to remain with the minerals only. Acid steps were carried out at 60 °C and each step took one hour, with rinsing with Milli-Q water in between and at the end. The dried samples were combusted to $CO_2$ in an elemental analyzer (MicroCube, Elementar) and catalytically (iron) reduced to elemental C in a graphitization system [108]. The carbon was transferred into a target and measured in a MICADAS AMS system [109]. Results was fractionation corrected and calibrated using Swisscal1.0 and the IntCal13 dataset.

Pottery samples were dated using thermoluminescence (TL) at CEZA. In the dark lab, the light-exposed surface was removed (softly) and the samples crushed and sieved. Sample pretreatment included a step with acetic acid, to remove carbonates, followed by a step with hydrogen peroxide, to remove organic material. Grain sizes of 4–11 µm were extracted as polymineral fine grain fraction, according to [110]. The powder was pipetted in stainless steel discs and measured in a Risø TL-DA-20 system with a $^{90}$Sr beta source and a $^{241}$Am alpha source. Both methods, multiple aliquot additive after [111] as well as single aliquot regenerative [112], were used to determine the natural dose of the sample. TL data was analyzed with the software Analyst 4.31.9 [113]. The dose rate of the sample and the surrounding sediment was determined using the low-level Germanium Well detector (Mirion, formerly Canberra), with the software Genie2000. Final age calculation was performed with the software Adele2017 (add-ideas).

Selected sediment samples were dated by optically stimulated luminescence (OSL) at the Geological Survey of Israel (GSI). In the field, samples were collected without exposure to sunlight and all laboratory procedures were carried out under suitable dim orange-red light. Quartz in the range of 90–125 µm was extracted using routine protocols [20]. The single aliquot regenerative dose (SAR) protocol [112] was used to measure the equivalent dose (De) of the sample on 2 mm aliquots. Dose rates were evaluated from the concentrations of the radio-elements U, Th, and K and measured by inductively coupled plasma (ICP) instrumentation. Cosmic dose was calculated from current burial depths and final age calculations were performed using DataBase [114]. For field and laboratory data of all TL- and OSL-ages, see Table A7 in Appendix D.

*Appendix B.4. Statistical Evaluation*

For a first statistical assessment of particle size distribution, the Gradistat for Excel program [55] was used and characteristic parameters, according to [115] calculated such as modality, mean grain size, sorting, skewness, and kurtosis. As well, principal component analysis (PCA) and the EMMAgeo v0.9.4 R package for end-member analysis were applied [56].

In order to systematically compare samples from the datasets presented above and to trace potential sediment sources, we used the EMMAgeo statistical algorithm in R to unmix and describe possible end-members which could represent deposition processes or different source materials [116]. It is based on the mathematical concept of eigenspace analysis [117]. EMMAgeo uses all available sediment samples to identify the potential, so far unknown end-members, and their contributions to the final archive [56]. All particle size results from both investigation regions described above were compiled into one matrix for joint evaluation. In order avoid empty values of medium silt, we used laser grain size results with the regular clay-silt border of 2 µm for statistics with EMMAgeo (see Appendix E for an extended discussion on the comparability of methods).

An eigenspace is an attribute space of interrelated processes or sources recorded in a geoarchive. The axes (i.e., eigenvectors) of this space are the underlying processes or sources as linear combinations of measured sediment properties. EMMAgeo extracts end members from the eigenspace of a data set and thus statistically "unmixes" the sources of the sediment. The resulting end members consists of loadings representing the composition in the sample space [118]. Prior to end-member analysis, raw grain-size distributions were rounded to sum exactly to 100%. A transformation of percentage values is necessary [119], as large-scale contrasts may result in weak or hidden correlations. EMMAgeo applies a column-wise weight transformation, as suggested by [120] and [121], and not a log-ratio transformation, as too many zero values may exist within the grain-size distribution space resulting in numerical problems, such as artificial extremes and divisions by zero [56].

The true number of final end-members is unknown, but a minimum number of potential end-members can be estimated by testing whether the log-ratios of an error matrix E are normally distributed [122]. EMMAgeo defines the minimum number of potential end-members by an iterative loop taking at least as many eigenvectors into a VARIMAX rotation [123] as needed to explain more than 95% of total variance in the original data [56].

We applied principle component analysis (PCA) using the R-function prcomp [57] to describe similarities between the samples, according to parameters of particle size distribution and magnetic susceptibilities. Furthermore, we utilized a non-parametric random forest approach, using the R-package randomForest [58], to test whether these parameters are suited to distinguish between the different types of deposits in this study (i.e., dust storm, reference samples, and soils). The algorithm applies the techniques of decision tree learning and bootstrap aggregation (or bagging), which repeatedly selects a random sample with replacement of the training set and the resulting calculated predictions for the samples. A detailed list of the samples fed into EMMAgeo and PCA is presented in Appendix F.

## Appendix C. Detailed Results of Laboratory Analyses

**Table A1.** Detailed results of $CaCO_3$-analyses from measuring samples before and after ignition at 500 °C in the C/N-analyzer, from gas pressure measured by the Karbonat-Bombe and calculated from Ca-contents that were determined by XRF.

| Sample No. | $CaCO_3$ % from C/N | $CaCO_3$ % Karbonat-Bombe | $CaCO_3$ % from Ca% |
|---|---|---|---|
| **Negev Western Wadi Ruin: Calcaric Leptosol (Protic, Siltic)** | | | |
| HH-WW R1 10 | 35 | n.a. | 40 |
| HH-WW-R1 20 | 35 | n.a. | 42 |
| **Negev Central Wadi Ruin: Protic Regosol (Siltic)** | | | |
| HH-CW-Ruin 10 | 40 | 39 | 41 |
| HH-CW-Ruin 25 | 42 | 40 | 43 |
| HH-CW-Ruin 50 | 40 | 38 | 42 |
| HH-CW-R-60 | 32 | n.a. | 37 |
| HH-CW-Ruin 75 | 36 | 36 | 37 |
| HH-CW-R 90 | 27 | n.a. | 32 |
| **Negev Loess Apron in Nahal Haroa: Cambic Calcisol (Siltic)** | | | |
| NH-LA-10cm | 28 | n.a. | n.a. |
| NH-LA-30cm | 27 | n.a. | n.a. |
| **Negev Reference Samples from Horvath Haluqium** | | | |
| HH-CW-Tur-Paleo | n.a. | 43 | 45 |
| HH-CW-chalk | n.a. | 98 | 97 |
| Haroa Farm chalk | n.a. | 98 | n.a. |
| Soft limestone | n.a. | 93 | 98 |
| Hard limestone | n.a. | 93 | 98 |
| **Natural Hilltop Soil in Petra Region: Protic Regosol (Arenic)** | | | |
| Sandplateau 1 | 2 | n.a. | n.a. |
| Sandplateau 2 | 1 | n.a. | n.a. |
| Sandplateau 3 | 0 | n.a. | n.a. |
| Sandplateau 4 | 0 | n.a. | 0 |
| Sandplateau Stein | 0 | n.a. | 0 |
| **Jabal Haroun Monastery Ruin: Protic Arenosol (Aridic, Aeolic) - Petra** | | | |
| FJHP Site 1, Trench R | 7 | n.a. | 7 |
| JH 1 | 7 | n.a. | 7 |
| **Petra Region, Jabal Farasha Hilltop Ruin: Calcaric Leptosol (Protic)** | | | |
| JF site 124/1 5 cm | 13 | n.a. | 13 |
| JF site 124/1 15 cm | 10 | n.a. | 11 |
| JF site 124/1 25 cm | 8 | n.a. | 9 |
| JF site 124/1 rock | 2 | n.a. | 3 |
| **Petra region, Umm Saysaban Hilltop Ruin: Calcaric Leptosol (Protic)** | | | |
| Umm Saysaban 5 cm | 12 | n.a. | 13 |
| Umm Saysaban 10 cm | 12 | n.a. | 13 |
| **Petra region, Abu Suwwan PPNB Ruins: Protic Regosol (Aeolic)** | | | |
| Abu Suwwan ashy layer 30 cm | n.a. | n.a. | n.a. |
| Abu Suwwan below nw 65 | 18 | 17 | 18 |
| **Petra Region, Shkarat Msaied Ppnb Ruins: Chromic Cambisol (Protocalcic)** | | | |
| Shkarat Msaeid 4 | n.a. | n.a. | n.a. |
| Shkarat Msaied 1 | 15 | 15 | 15 |
| **Reference Samples from Petra Region** | | | |
| Ba'ja Sandstein (rock) | 1 | n.a. | 2 |
| Disi Sandstone | 4 | n.a. | 2 |
| Um Ishrin sandstone | 2 | n.a. | 2 |
| Abu Khushayba sandstone JH | 2 | n.a. | 1 |
| JH limestone outcrop | 97 | 98 | 57–100 |
| Beidha Fan | 16 | n.a. | n.a. |
| Fan Umm Sayhoun | 5 | n.a. | n.a. |
| Umm Seihun formation (US 1) | 34 | n.a. | 36 |

The similarity of results suggests that most if not all Ca is bound in $CaCO_3$ bearing minerals. A mismatch between the different methods of $CaCO_3$ determination suggests the presence of partly dolomitic limestone in sample *Jh limestone outcrop*, which could be confirmed by X-ray diffraction [124]. The Wadi As Sir limestone formation is known to contain partly dolomitic limestone [48]. A total of 11% of Mg were present in the rock, which is more than 10 times the amount than in any of the studied ruin soil samples. Therefore, this outcrop cannot be their source of $CaCO_3$.

The effects of $CaCO_3$ removal prior to grain size analysis can be illustrated with sample *US 1*. HCl pretreatment led to a strong increase of the fine clay fraction and a subsequent reduction of the coarse sand fraction. This suggests that the relatively high coarse sand content of sample *US 1* might result from cementation by $CaCO_3$, but the strong increase of fine clay after acid pretreatment is probably an artifact created by dissolution of primary calcareous minerals [125]. Despite acid treatment, the sample remains dominated by medium sand, which points to a local origin from the sandstones and not to a loess-like deposit despite its 'silty' appearance in the field.

In contrast to the Petra region, the samples that could be collected simultaneously with and without rain in the Negev on 25 March 2003, show nearly no differences. It should be noted, though, that general deposition conditions of this storm were the same; just the positions of the samplers differed. Yellow and very pale brown colors characterize all dust samples that were collected in the Negev and the $CaCO_3$ contents range between 30%–44%.

**Table A2.** Results of particle size analysis of the Negev samples with sieving and the Sedigraph and selected statistical parameters, such as the main modes and average grain size calculated with the Gradistat program. Note that the very fine sand (in italics) is part of the fine sand fraction. Skeleton is shown as well (in italics), and summarizing grain size units (in bold).

| Sample No. | Skeleton > 2 mm (%) | Sand % | Silt % | Clay % | Coarse Sand % | Medium Sand % | Fine Sand % | Very Fine Sand % | Coarse Silt % | Medium Silt % | Fine Silt % | Coarse Clay % | Medium Clay % | Fine Clay % | Mode 1 (μm) | Mode 2 (μm) | Mode 3 (μm) | Mean (μm) |
|---|---|---|---|---|---|---|---|---|---|---|---|---|---|---|---|---|---|---|
| Western Wadi Ruin: Calcaric Leptosol (Protic, Siltic) | | | | | | | | | | | | | | | | | | |
| HH-WW R1 10 | *21* | **42** | **45** | **13** | 4 | 8 | 29 | *23* | 18 | 15 | 13 | 9 | 2 | 2 | 132 | 0 | 0 | 25 |
| HH-WW-R1 20 | *16* | **45** | **42** | **13** | 5 | 11 | 30 | *24* | 12 | 16 | 13 | 8 | 3 | 2 | 132 | 13 | 0 | 28 |
| Central Wadi Ruin: Protic Regosol (Siltic) | | | | | | | | | | | | | | | | | | |
| HH-CW-Ruin 10 | *4* | **21** | **53** | **26** | 2 | 3 | 16 | *13* | 22 | 14 | 16 | 14 | 6 | 5 | 42 | 4 | 0 | 10 |
| HH-CW-Ruin 25 | *19* | **23** | **53** | **24** | 4 | 3 | 17 | *11* | 18 | 16 | 19 | 16 | 5 | 3 | 4 | 42 | 1315 | 11 |
| HH-CW-Ruin 50 | *9* | **41** | **47** | **12** | 3 | 6 | 32 | *22* | 20 | 14 | 13 | 7 | 3 | 2 | 132 | 0 | 0 | 26 |
| HH-CW-R-60 | *24* | **36** | **51** | **13** | 5 | 5 | 27 | *23* | 19 | 18 | 14 | 9 | 3 | 1 | 132 | 0 | 0 | 22 |
| HH-CW-Ruin 75 | *12* | **20** | **51** | **29** | 1 | 4 | 15 | *15* | 28 | 9 | 14 | 11 | 7 | 11 | 42 | 4 | 0 | 8 |
| HH-CW-R 90 | *10* | **27** | **51** | **22** | 2 | 2 | 23 | *21* | 22 | 14 | 14 | 11 | 7 | 4 | 132 | 4 | 0 | 13 |
| Loess Paleosol Of Loess Apron in Nahal Haroa: Cambic Calcisol (Siltic) | | | | | | | | | | | | | | | | | | |
| NH-LA-10cm | *10* | **32** | **57** | **11** | 3 | 3 | 25 | *24* | 39 | 10 | 8 | 7 | 4 | 0 | 42 | 0 | 0 | 27 |
| NH-LA-30cm | *5* | **27** | **58** | **15** | 2 | 2 | 23 | *22* | 35 | 12 | 10 | 8 | 6 | 1 | 42 | 0 | 0 | 20 |
| Reference Samples from Horvath Haluqium | | | | | | | | | | | | | | | | | | |
| HH-CW-Tur-Paleo | *14* | **20** | **42** | **38** | 2 | 6 | 11 | *n.a.* | 12 | 16 | 14 | 8 | 9 | 22 | 13 | 0 | 0 | 3 |
| HH-CW-chalk | *n.a.* | **25** | **39** | **36** | 5 | 9 | 11 | *n.a.* | 1 | 0 | 38 | 35 | 0 | 2 | 4 | 132 | 0 | 8 |
| Haroa Farm - chalk | *n.a.* | **12** | **86** | **2** | 0 | 1 | 11 | *n.a.* | 1 | 7 | 78 | 1 | 0 | 2 | 4 | 0 | 0 | 5 |
| HH-WW-C2-soft limestone | *n.a.* | **16** | **80** | **4** | 4 | 3 | 8 | *n.a.* | 7 | 53 | 20 | 2 | 1 | 1 | 13 | 132 | 0 | 14 |

**Table A3.** Results of particle size analysis of the samples from the Petra region by sieving and Sedigraph and respective statistical parameters. Note that the very fine sand (in italics) is part of the fine sand fraction. Skeleton is shown as well (in italics), and summarizing grain size units (in bold).

| Sample No. | Skeleton > 2 mm (%) | Sand % | Silt % | Clay % | Coarse Sand % | Medium Sand % | Fine Sand % | Very Fine Sand % | Coarse Silt % | Medium Silt % | Fine Silt % | Coarse Clay % | Medium Clay % | Fine Clay % | Mode 1 (µm) | Mode 2 (µm) | Mode 3 (µm) | Mean (µm) |
|---|---|---|---|---|---|---|---|---|---|---|---|---|---|---|---|---|---|---|
| **Jabal Haroun Monastery Ruin: Protic Arenosol (Aridic, Aeolic)** | | | | | | | | | | | | | | | | | | |
| FJHP Site 1, Trench R | *n.a.* | **84** | **8** | **8** | 2 | 47 | 36 | *3* | 3 | 2 | 3 | 3 | 2 | 2 | 415 | 0 | 0 | 178 |
| JH 1 | *n.a.* | **75** | **13** | **12** | 4 | 31 | 39 | *13* | 5 | 4 | 4 | 4 | 3 | 5 | 132 | 0 | 0 | 68 |
| **Jabal Farasha hilltop Ruin: Calcaric Leptosol (Protic)** | | | | | | | | | | | | | | | | | | |
| JF site 124/1 5 cm | *3* | **51** | **36** | **13** | 3 | 17 | 30 | *15* | 18 | 11 | 8 | 6 | 4 | 3 | 132 | 0 | 0 | 38 |
| JF site 124/1 15 cm | *5* | **53** | **35** | **12** | 3 | 16 | 34 | *16* | 19 | 9 | 7 | 6 | 3 | 3 | 132 | 0 | 0 | 41 |
| JF site 124/1 25 cm | *21* | **57** | **29** | **14** | 4 | 14 | 39 | *19* | 15 | 8 | 6 | 6 | 5 | 2 | 132 | 1 | 0 | 38 |
| JF site 124/1 rock | *n.a.* | **60** | **18** | **22** | 48 | 10 | 2 | *n.a.* | 3 | 8 | 8 | 10 | 9 | 3 | 1315 | 1 | 13 | 89 |
| **Umm Saysaban Hilltop Ruin: Calcaric Leptosol (Protic)** | | | | | | | | | | | | | | | | | | |
| Umm Saysaban 5 cm | *13* | **51** | **35** | **14** | 3 | 16 | 32 | *13* | 15 | 11 | 9 | 8 | 4 | 2 | 132 | 0 | 0 | 35 |
| Umm Saysaban 10 cm | *12* | **54** | **30** | **16** | 5 | 18 | 32 | *13* | 13 | 9 | 8 | 8 | 5 | 3 | 132 | 0 | 0 | 36 |
| **Abu Suwwan PPNB Ruins: Protic Regosol (Aeolic)** | | | | | | | | | | | | | | | | | | |
| Abu Suwwan below nw 65 | *2* | **62** | **23** | **15** | 1 | 39 | 22 | *7* | 7 | 8 | 8 | 7 | 5 | 3 | 415 | 4 | 0 | 48 |
| **Shkarat Msaied PPNB Ruins: Chromic Cambisol (Protocalcic)** | | | | | | | | | | | | | | | | | | |
| Shakarat Msaid 1 | *0* | **52** | **26** | **22** | 0 | 30 | 21 | *5* | 11 | 7 | 8 | 8 | 6 | 8 | 415 | 4 | 0 | 27 |
| **Natural Hilltop Soil on Sandstone Plateau: Protic Regosol (Arenic)** | | | | | | | | | | | | | | | | | | |
| Sandplateau 1 | *0* | **80** | **12** | **8** | 3 | 59 | 18 | *n.a.* | 6 | 4 | 3 | 4 | 3 | 2 | 415 | 0 | 0 | 143 |
| Sandplateau 2 | *5* | **82** | **9** | **9** | 2 | 61 | 19 | *n.a.* | 4 | 3 | 3 | 3 | 3 | 3 | 415 | 0 | 0 | 169 |
| Sandplateau 3 | *9* | **82** | **9** | **9** | 2 | 61 | 19 | *n.a.* | 4 | 2 | 3 | 3 | 3 | 3 | 415 | 0 | 0 | 161 |
| Sandplateau 4 | *0* | **82** | **8** | **10** | 1 | 62 | 18 | *n.a.* | 3 | 2 | 3 | 3 | 3 | 4 | 415 | 0 | 0 | 160 |
| Sandplateau Stein | *0* | **89** | **7** | **3** | 1 | 75 | 12 | *n.a.* | 2 | 2 | 4 | 3 | 1 | 0 | 415 | 0 | 0 | 248 |
| **Reference Samples from Petra Region: Rocks, Current Fans, and Cemented Paleosol: Cambic Calcisol (Loamic)** | | | | | | | | | | | | | | | | | | |
| Ba'ja sandstone | *n.a.* | **72** | **11** | **17** | 14 | 31 | 27 | *n.a.* | 3 | 3 | 6 | 17 | 0 | 0 | 415 | 1 | 0 | 56 |
| Disi sandstone | *n.a.* | **90** | **6** | **4** | 11 | 67 | 12 | *n.a.* | 1 | 2 | 2 | 2 | 1 | 1 | 415 | 0 | 0 | 281 |
| Um Ishrin sandstone | *n.a.* | **81** | **12** | **7** | 18 | 55 | 9 | *n.a.* | 3 | 4 | 4 | 3 | 1 | 2 | 415 | 0 | 0 | 186 |
| Abu Khushayba sandstone JH | *n.a.* | **79** | **11** | **10** | 3 | 63 | 13 | *n.a.* | 2 | 3 | 5 | 5 | 2 | 2 | 415 | 0 | 0 | 101 |
| JH limestone outcrop | *n.a.* | **48** | **46** | **6** | 29 | 14 | 5 | *n.a.* | 12 | 27 | 7 | 2 | 1 | 2 | 1315 | 13 | 0 | 74 |
| Beidha Fan | *9* | **69** | **21** | **11** | 3 | 43 | 22 | *n.a.* | 9 | 5 | 6 | 7 | 3 | 1 | 415 | 1 | 0 | 72 |
| Fan Umm Sayhoun | *7* | **91** | **3** | **6** | 16 | 66 | 9 | *n.a.* | 0 | 1 | 1 | 2 | 1 | 3 | 415 | 0 | 0 | 338 |
| Cemented paleosol US 1 | *n.a.* | **82** | **14** | **4** | 26 | 40 | 16 | *8* | 4 | 4 | 5 | 3 | 0 | 1 | 415 | 0 | 0 | 230 |
| US 1after removal of CaCO$_3$ | *n.a.* | **68** | **9** | **23** | 13 | 42 | 13 | *n.a.* | 3 | 3 | 3 | 4 | 3 | 16 | 415 | 0 | 0 | 29 |

**Table A4.** Results of particle size analysis of the current dust samples from Petra region and the Negev by laser grain size analysis with assumed clay-silt border of 5 µm in order to maximize comparability with Sedigraph results and respective statistical parameters. Note that the very fine sand (in italics) is part of the fine sand fraction. Skeleton is shown as well (in italics), and summarizing grain size units (in bold). The medium silt fraction is mathematically eliminated due to the adapted clay border.

| Sample No. | Skeleton > 2 mm (%) | Sand % | Silt % | Clay % | Coarse Sand % | Medium Sand % | Fine Sand % | Very Fine Sand % | Coarse Silt % | Medium Silt % | Fine Silt % | Coarse Clay % | Medium Clay % | Fine Clay % | Mode 1 (µm) | Mode 2 (µm) | Mode 3 (µm) | Mean [µm] |
|---|---|---|---|---|---|---|---|---|---|---|---|---|---|---|---|---|---|---|
| **Negev** | | | | | | | | | | | | | | | | | | |
| 25-03-03-HH dry spot | *0* | **4** | **83** | **13** | 0 | 0 | 4 | *3* | 44 | n.a. | 39 | 13 | 0 | 0 | 42 | 0 | 0 | 12 |
| 25-03-03-HH-incl.-washout | *0* | **2** | **84** | **14** | 0 | 0 | 2 | *2* | 39 | n.a. | 44 | 14 | 0 | 0 | 42 | 0 | 0 | 11 |
| 11-12-10-HH | *0* | **24** | **56** | **20** | 0 | 1 | 23 | *21* | 38 | n.a. | 18 | 19 | 1 | 0 | 42 | 1 | 0 | 16 |
| 12-12-10-HH | *0* | **22** | **59** | **19** | 0 | 0 | 22 | *21* | 41 | n.a. | 18 | 19 | 1 | 0 | 42 | 1 | 0 | 16 |
| 29-02-12-HH | *0* | **30** | **58** | **12** | 0 | 0 | 30 | *28* | 46 | n.a. | 11 | 12 | 0 | 0 | 42 | 1 | 0 | 27 |
| 18-04-12-HH | *0* | **26** | **61** | **13** | 0 | 1 | 25 | *23* | 48 | n.a. | 13 | 12 | 0 | 0 | 42 | 1 | 0 | 24 |
| 20-12-12-HH | *0* | **23** | **64** | **13** | 0 | 0 | 23 | *22* | 50 | n.a. | 14 | 13 | 0 | 0 | 42 | 1 | 0 | 18 |
| 22-03-13-HH | *0* | **24** | **60** | **16** | 0 | 1 | 23 | *21* | 44 | n.a. | 16 | 16 | 0 | 0 | 42 | 1 | 0 | 17 |
| 11-02-15-HH | *0* | **22** | **60** | **18** | 0 | 0 | 22 | *21* | 44 | n.a. | 15 | 17 | 1 | 0 | 42 | 1 | 0 | 16 |
| 01-12-16-HH | *0* | **16** | **68** | **16** | 0 | 0 | 16 | *15* | 51 | n.a. | 18 | 15 | 0 | 0 | 42 | 1 | 0 | 17 |
| 05-01-18-HH | *0* | **21** | **63** | **16** | 0 | 3 | 17 | *15* | 45 | n.a. | 18 | 15 | 1 | 0 | 42 | 1 | 0 | 9 |
| 28-03-18-HH | *0* | **6** | **72** | **22** | 0 | 1 | 6 | *5* | 34 | n.a. | 38 | 21 | 0 | 0 | 42 | 0 | 0 | 12 |
| **Petra** | | | | | | | | | | | | | | | | | | |
| JH-19-12-2016 | *0* | **89** | **8** | **3** | 0 | 36 | 54 | *20* | 4 | n.a. | 4 | 3 | 0 | 0 | 132 | 0 | 0 | 157 |
| JH-07.01.2017 | *0* | **83** | **13** | **4** | 10 | 62 | 11 | *5* | 7 | n.a. | 6 | 4 | 0 | 0 | 415 | 0 | 0 | 204 |
| JH-15.02.2017 | *0* | **44** | **43** | **13** | 1 | 24 | 19 | *11* | 22 | n.a. | 21 | 13 | 0 | 0 | 415 | 42 | 10 | 34 |
| JH-01.03.2017 | *0* | **46** | **46** | **8** | 0 | 11 | 35 | *22* | 30 | n.a. | 16 | 7 | 0 | 0 | 132 | 0 | 0 | 40 |
| 05-08-17-JH | *0* | **80** | **14** | **6** | 4 | 63 | 14 | *4* | 7 | n.a. | 7 | 5 | 0 | 0 | 415 | 0 | 0 | 170 |
| 05-08-17-JH closed box | *0* | **82** | **13** | **5** | 3 | 65 | 13 | *3* | 6 | n.a. | 6 | 5 | 0 | 0 | 415 | 0 | 0 | 182 |
| Saleh.25.11.2016 | *0* | **86** | **10** | **4** | 0 | 52 | 34 | *9* | 5 | n.a. | 5 | 4 | 0 | 0 | 415 | 0 | 0 | 184 |
| 20-06-17-Saleh | *0* | **59** | **33** | **8** | 0 | 19 | 41 | *24* | 23 | n.a. | 10 | 8 | 0 | 0 | 132 | 1 | 0 | 63 |
| 05-08-17-Saleh | *0* | **65** | **28** | **7** | 1 | 23 | 42 | *24* | 19 | n.a. | 8 | 6 | 0 | 0 | 132 | 1 | 0 | 85 |

**Table A5.** Table summarizing indicators of pedogenesis in the studied hilltop soils. CIA refers to the index of chemical alteration according to [74]. $\chi$ is the weight-specific magnetic susceptibility, $\chi$FD is the frequency-dependent change of susceptibility, $\chi$Ox-diff is the change of susceptibility after oxalate extraction, and $\chi$Di-diff after iron extraction with dithionite.

| Sample No. | Fe(o) (mg/g) | Fe(d) (mg/g) | Fe (d/t) | CIA quotient A | CIA quotient B | $K/Al_2O_3$-ratio | $CaCO_3$ % | Clay % | $\chi$ (m³/kg) E-8 | $\chi$FD % | $\chi$Ox-diff % | $\chi$Di-diff % |
|---|---|---|---|---|---|---|---|---|---|---|---|---|
| **Negev** | | | | | | | | | | | | |
| *Ruin Western Wadi* | | | | | | | | | | | | |
| HH-WW R1 10 cm | 0.21 | 2.52 | 0.14 | 0.92 | 0.80 | 0.09 | 35 | 13 | 44 | 4 | 8 | 31 |
| HH-WW-R1 20 cm | 0.20 | 2.41 | 0.13 | 0.92 | 0.81 | 0.07 | 35 | 13 | 44 | 4 | 6 | 36 |
| *Ruin Central Wadi* | | | | | | | | | | | | |
| HH-CW-R 10 cm | 0.49 | 4.27 | 0.20 | 0.91 | 0.80 | 0.16 | 40 | 26 | 64 | 6 | | |
| HH-CW-R 25 cm | 0.50 | 3.62 | 0.19 | 0.92 | 0.82 | 0.14 | 42 | 24 | 64 | 7 | | |
| HH-CW-R 50 cm | 0.54 | 3.07 | 0.17 | 0.92 | 0.82 | 0.17 | 40 | 12 | 74 | 8 | | |
| HH-CW-R 60 cm | 0.23 | 1.84 | 0.11 | 0.92 | 0.80 | 0.13 | 32 | 13 | 61 | 5 | 2 | 35 |
| HH-CW-R 75 cm | 0.32 | 3.74 | 0.18 | 0.92 | 0.79 | 0.17 | 36 | 29 | 55 | 3 | | |
| HH-CW-R 90 cm | 0.21 | 2.12 | 0.12 | 0.92 | 0.77 | 0.13 | 27 | 22 | 55 | 3 | 6 | 38 |
| *Haroa Loess Aprons* | | | | | | | | | | | | |
| NH-LA-10 cm | 0.40 | 4.59 | 0.20 | 0.92 | 0.75 | 0.16 | 28 | 11 | 58 | 3 | 14 | 25 |
| NH-LA-30 cm | 0.38 | 4.64 | 0.20 | 0.92 | 0.73 | 0.16 | 27 | 15 | 61 | 3 | 10 | 23 |
| **Petra Region** | | | | | | | | | | | | |
| *Monastery Ruin Soil on Jabal Haroun* | | | | | | | | | | | | |
| FJHP Site 1, Trench R | 0.05 | 3.79 | 0.50 | 0.97 | 0.62 | 0.11 | 7 | 8 | 18 | 7 | | |
| *Jabal Farasha Hilltop Ruin* | | | | | | | | | | | | |
| JF site 124/1 5 cm | 0.21 | 7.53 | 0.44 | 0.91 | 0.55 | 0.09 | 13 | 13 | 32 | 5 | 0 | 54 |
| JF site 124/1 15 cm | 0.30 | 6.72 | 0.45 | 0.91 | 0.49 | 0.07 | 10 | 12 | 26 | 6 | | |
| JF site 124/1 25 cm | 0.19 | 7.61 | 0.45 | 0.90 | 0.40 | 0.05 | 8 | 14 | 20 | 6 | | |
| JF site 124/1 rock (sandstone) | 0.21 | 52.20 | 0.63 | 0.72 | 0.09 | 0.02 | 2 | 22 | 7 | 14 | | |
| *Umm Saysaban* | | | | | | | | | | | | |
| Umm Saysaban 5 cm | 0.25 | 5.56 | 0.34 | 0.91 | 0.52 | 0.06 | 12 | 14 | 28 | 7 | | |
| Umm Saysaban 10 cm | 0.17 | 5.24 | 0.36 | 0.90 | 0.51 | 0.05 | 12 | 16 | 23 | 7 | | |
| *Abu Suwwan Paleosol bel. Ruins* | | | | | | | | | | | | |
| Abu Suwwan below nw 65 cm | 0.14 | 4.02 | 0.39 | 0.96 | 0.76 | 0.07 | 18 | 15 | 82 | 6 | | |
| *Shakarat Msaid Paleosol bel. Ruins* | | | | | | | | | | | | |
| Shkarat Msaied 1 | 0.27 | 4.42 | 0.31 | 0.95 | 0.69 | 0.14 | 15 | 22 | 22 | 5 | | |
| *Natural Soil on Sandstone Plateau* | | | | | | | | | | | | |
| Sandplateau 1 | 0.05 | 2.47 | 0.41 | 0.96 | 0.32 | 0.06 | 2 | 8 | 20 | −4 | | |
| Sandplateau 2 | 0.04 | 2.01 | 0.46 | 0.97 | 0.26 | 0.05 | 1 | 9 | 15 | −6 | | |
| Sandplateau 3 | 0.00 | 2.01 | 0.46 | 0.97 | 0.15 | 0.04 | 0 | 9 | 10 | −2 | | |
| Sandplateau 4 | 0.00 | 1.75 | 0.42 | 0.97 | 0.07 | 0.04 | 0 | 10 | 11 | 4 | | |
| Sandplateau Stein (sandstone) | 0.00 | 0.14 | 0.20 | 0.96 | 0.03 | 0.01 | 0 | 4 | 0 | | | |
| Baja Sandstein Oberfläche (sandstone) | 0.03 | 0.23 | 0.15 | 0.86 | 0.09 | 0.01 | 1 | 17 | 0 | | | |
| *Cemented Paleosol* | | | | | | | | | | | | |
| US 1 | 0.03 | 1.32 | 0.25 | 0.98 | 0.94 | 0.07 | 34 | 4 | 6 | 10 | | |

Additional Interpretation of Pedogenesis:

Absolute values of oxalate-extractable iron Fe(o) and dithionite-extractable iron Fe(d) show mostly low concentrations of oxides. The much higher contents of Fe(d), compared to Fe(o), suggest that iron oxides occur predominantly in well-crystalline form (probably mainly goethite and perhaps small amounts of hematite). The effect of additions of disintegrating sandstone in a developing soil can be estimated with the sample *FJHP 1, trench R* from the monastery cover on Jabal Haroun, which was found to contain mortar made from mud. Substrate derived from the sandstones mostly reduces the contents of extractable iron. The very similar amounts of extractable iron in the samples of each ruin profile suggest either a constant supply of similar allochthonous material or a balance of in-situ weathering and dust deposition.

Pedogenesis, or fire, can lead to higher magnetic susceptibilities, which usually depend on the concentrations of magnetite and maghemite. While magnetite is usually of lithogenic origin, maghemite can be formed during pedogenesis in Jordan [50]. Extraction by oxalate and dithionite removes superparamagnetic grains [126]. Dithionite very effectively removes all pedogenic maghemite [127], but only fine magnetite particles (<1 μm, [128]). However, some forms of maghemite are readily soluble in oxalate but only slightly in dithionite [129].

The exceptionally elevated susceptibility of the Abu Suwwan paleosol might be explained with charred materials from the settlement. The reference samples, in contrast, show only negligible magnetic susceptibilities in both investigation regions. The only exceptions are the Turonian paleosol (with 17 $m^3$/kg E-8) in the Negev, but it is a soil of the past, and the Beidha fan (with 24 $m^3$/kg E-8) in the Petra region. The latter likely contains eroded soil.

## Appendix D. Dating Results

**Table A6.** Summary of radiocarbon dating results.

| Labcode | Sample | Material | ¹⁴C yrs BP | ± | cal BCE/CE 1 σ | cal BCE/CE 2 σ | C (%) | Software/Dataset |
|---|---|---|---|---|---|---|---|---|
| UBA-38659 | HH-CW-R 13-15 | charcoal | 229 | 35 | 1643–1949* CE | 1527–1949* CE | 76.8 | CALIB7.04/IntCal13 |
| UBA-38660 | HH-WC-R-20 | charcoal | 106 | 27 | 1694–1918 CE | 1682–1935 CE | 75.7 | CALIB7.04/IntCal13 |
| UBA-38661 | HH-WC-R-5-8 | charcoal | 4 | 24 | age - 2 sigma outside calibration age range | | 82.8 | |
| UBA-38980 | AS ashy layer 30 | charcoal | 3599 | 28 | 2014–1916 BCE | 2026–1891 BCE | 0.62 | CALIB7.04/IntCal13 |
| UBA-39002 | Shakarat Msaid 4 | charcoal | 12288 | 59 | 12385–12131 BCE | 12666–12077 BCE | 65.8 | CALIB7.04/IntCal13 |
| MAMS 31135 | HH CW R1 50 TL temper (ash) | charcoal | 7282 | 29 | 6210–6092 BCE | 6219–6072 BCE | 1.4 | SwissCal 1.0/IntCal13 |

* impinges on end of calibration curve. Calibration references: CALIB7.04: [130]; IntCal13: [131].

**Table A7.** Summary of OSL- and TL-dating results.

| Labcode | Sample | Depth Below Surface (cm) | Water % | Th μg/g | ± | U μg/g | ± | K % | ± | Mean DR mGy/a | Cosm. DR mGy/a | OSL/TL | OD (%) | N | De Gy | ± | a-Value | Age yrs | ± | Year BCE/CE yrs | Remarks |
|---|---|---|---|---|---|---|---|---|---|---|---|---|---|---|---|---|---|---|---|---|---|
| MAL 10341 | HH-WW-R-TL 15 | 15 | 5 ± 5 * | 7.46 | 0.35 | 5.49 | 0.22 | 1.90 | 0.09 | 4.38 ± 0.4 | 0.26 ± 0.03 | TL | | | 11.53 | 0.3 | 0.05 | 2736 | 252 | 970–470 BCE | Fraction 4–11 μm |
| MAL 10342 | HH CW R1 50 TL | 50 | 5 ± 5 * | 5.81 | 0.17 | 2.12 | 0.21 | 2.47 | 0.4 | 4.25 ± 0.4 | 0.21 ± 0.02 | TL | | | 10.38 | 0.5 | 0.1 | 2586 | 264 | 830–300 BCE | Fraction 4–11 μm |
| HLK-12 | HH-CW-Ruin 25 | 25 | 5 ± 3 | 4.7 | | 1.5 | | 1.08 | | 1.82 ± 0.07 * | | OSL | 35 | 30/31 | 2.0 | 0.1 | | 1070 | 80 | 870–1020 CE | Fraction 90–125 μm |
| HLK-11 | HH-CW-Ruin 50 | 55 | 8 ± 3 | 3.4 | # | 1.3 | # | 0.73 | # | 1.36 ± 0.05 * | | OSL | 34 | 32/33 | 5.9 | 0.4 | | 4300 | 290 | 2570–1990 BCE | Fraction 90–125 μm |
| HLK-10 | HH-CW-Ruin 90 | 90 | 8 ± 3 | 6.3 | # | 2.1 | # | 1.31 | # | 2.21 ± 0.09 * | | OSL | 32 | 17/19 | 54 | 3 | | 24300 | 1700 | 23,980–20,580 BCE | Fraction 90–125 μm |
| PET-20 | JF site 124/1 15 | 15 | 5 ± 2 | 10.2 | # | 2.42 | # | 0.56 | # | 2.02 ± 0.07 * | | OSL | 27 | 19/19 | 2.2 | 0.1 | | 1060 | 80 | 870–1040 CE | Fraction 90–125 μm |

DR = dose rate, De = Paleodose, Th = Thorium, U = Uranium, K = Potassium. * including the cosmic dose. # errors on radioelements: Th- 10%; U- 8%; K- 5%. OSL De values and errors calculated using the central age model. OD—overdispersion. N—number of aliquots used for age calculations out of those measured.

**Appendix E. Comparison of Grain Size Analysis Results**

The problematic comparability of laser and sedimentation methods is a well-known and unsolved problem of grain size analysis (see e.g., [4,125,132–139]). There is no universal solution available, such as a general correction factor, and it is possible that comparability depends on the type and amount of material [136]. Recommendations vary as follows: While some researchers proposed factors to correct misinterpretation of irregular shapes by the laser (e.g., [140]), others stated that it is necessary to conduct a calibration comparison each time when a new type of material/sample region is investigated [132]. Some authors suggested that not an optical misinterpretation by the laser, but the formation of aggregates and magnetic properties of the samples were the main problem. Vdovi'c et al. [141] proposed pre-treatment with $H_2O_2$ as solution, while [137] suggest a careful and intense pretreatment procedure with dispersant, ultrasonic, and other measures, depending on the material. Other researchers doubt that a correction and direct comparison of the two methods is possible at all [136,139].

Most proposed correction factors apply modifications of the clay-silt border in order to compensate a possible optical underestimation of clay particles by the laser. This could be due to the platy and non-spherical shapes of phyllosilicates, which the laser interprets as spheres. The proposed correcting clay-silt borders range from 8 μm [140] over 6.2 μm [138] and 5 μm [11,54] to 4 μm [142]. Following previous studies of loess in the Negev [11], we decided to apply the 5 μm border. We set the fine silt border to 20 μm and thus mathematically eliminated the fraction of medium silt in the laser measurements. In order to check the comparability specifically for our material, a few dust samples where sufficient amounts were available were analyzed with the Sedigraph, too. As well, some of the samples from the ruin soils were measured with the laser. Results of the method comparison on the dust samples are shown in Table A8 and, for the ruin soils, in Table A9.

Compared with the Sedigraph, the laser underestimates the clay and overestimates the silt fraction of the dust samples. Applying the 5 μm clay border, these effects are reduced and results are much more similar to the Sedigraph. Only for the sample of the Negev dust storm on 21 December 2010, results are still to some degree different, as the laser measured a higher sand content.

For the samples of the ruin soils that were studied by both methods, the laser underestimates the sand contents by approximately 10% for all samples (Table A9). Applying the normal clay border of 2 μm, there is an underestimation of clay contents discernible as well. The laser seems not to detect fine and medium clay. This does not change when applying the 5 μm correction of the clay-silt border. Total sand contents continue to be underestimated, very different sub-fractions of the sand are detected, and the clay contents are now higher than with the Sedigraph and seem over-corrected. Applying other correction factors from the literature (with mostly higher clay borders) will not solve this problem, since clay values would increase even further, and it is not possible to correct the different sand values.

For some samples it would have been a possibility to combine wet sieving with laser analysis, but there was a risk that the silt and clay fractions recovered after sieving were too small for further analyses. In addition, laser results from the literature always include direct measurements of the sand fraction. Therefore, we did not combine wet sieving with laser analysis of silt and clay.

It can be concluded that the tested 5 μm correction of the clay-silt border worked with the dust samples, but not with the ruin soils. We therefore measured only the dust samples with the laser.

The ruin soils can possibly not reliably be measured with a laser device. The divergent result could be related to sample size, particularly with regard to the sand fractions. As it is necessary to select a much smaller sample for laser analysis than for sieving and sedimentation, choosing a comparable sample may pose a problem, as the absence or presence of a few large sand grains could skew results significantly. However, since the repetitions of the laser measurements showed good reproducibility, a systematic difference related to the measurement principle seems possible. In this context, our divergent results for the ruin soils seem to match the findings of [136] for samples from natural environments that represent complex mixtures of diverse grain sizes and varying mineralogy. Interactions of particles of different sizes and adsorption-reflection properties with the laser light,

as well detector thresholds and saturation problems, might pose severe limitations for reliable and reproducible measurements of ruin soils with the laser device.

However, comparison seems well possible for the dust samples, possibly because of a less complex composition of this material. Since applying the 5 µm clay-silt border seemed to improve comparability with the Sedigraph, the dust results were recalculated accordingly and used for the comparison with the ruin soils.

**Table A8.** Comparison of results of particle size analysis of sediment samples collected during current dust storms by wet sieving and Sedigraph with the Malvern Mastersizer 3000 laser device, applying clay-silt borders of 2 and 5 μm for laser results.

| Sample No. & Method | Coarse Sand % | Medium Sand % | Fine Sand % | Coarse Silt % | Medium Silt % | Fine Silt % | Coarse Clay % | Medium Clay % | Fine Clay % | Sand % | Silt % | Clay % | Very Fine Sand (125–63 μm) % |
|---|---|---|---|---|---|---|---|---|---|---|---|---|---|
| | | | | | Sedigraph | | | | | | | | |
| 25-03-03 HH incl washout (Sedigraph) | 0 | 0 | 2 | 14 | 51 | 18 | 7 | 5 | 3 | **3** | **82** | **15** | *n.a.* |
| 12-12-10-HH (Sedigraph) | 0 | 0 | 11 | 38 | 16 | 12 | 10 | 6 | 6 | **12** | **66** | **22** | *n.a.* |
| 05-08-17 JH (Sedigraph) | 0 | 48 | 36 | 4 | 4 | 3 | 2 | 1 | 1 | **84** | **11** | **5** | *n.a.* |
| 05-08-17 JH closed box (Sedigraph) | 0 | 47 | 36 | 4 | 4 | 3 | 2 | 1 | 1 | **84** | **11** | **5** | *n.a.* |
| | | | | | Laser, Clay Border 2 μm | | | | | | | | |
| 25-03-03 HH incl. washout (Laser 2 μm) | 0 | 0 | 2 | 39 | 41 | 12 | 6 | 0 | 0 | **2** | **92** | **6** | *2* |
| 12-12-10-HH (Laser 2 μm) | 0 | 0 | 22 | 41 | 15 | 14 | 8 | 1 | 0 | **23** | **69** | **8** | *21* |
| 05-08-17-JH (Laser 2 μm) | 4 | 63 | 14 | 7 | 6 | 4 | 2 | 0 | 0 | **81** | **17** | **2** | *4* |
| 05-08-17-JH closed box (Laser 2 μm) | 3 | 65 | 13 | 6 | 5 | 4 | 2 | 0 | 0 | **82** | **16** | **2** | *3* |
| | | | | | Laser, Clay Border 5 μm | | | | | | | | |
| 25-03-03 HH incl. washout (Laser 5 μm) | 0 | 0 | 2 | 39 | n.a. | 44 | 14 | 0 | 0 | **3** | **83** | **14** | *2* |
| 12-12-10-HH (Laser 5 μm) | 0 | 0 | 22 | 41 | n.a. | 18 | 19 | 1 | 0 | **22** | **59** | **19** | *21* |
| 05-08-17-JH (Laser 5 μm) | 3 | 65 | 13 | 6 | n.a. | 6 | 5 | 0 | 0 | **82** | **13** | **5** | *3* |
| 05-08-17-JH closed box (Laser 5 μm) | 3 | 65 | 13 | 6 | n.a. | 6 | 5 | 0 | 0 | **82** | **13** | **5** | *3* |

**Table A9.** Comparison of results of particle size analysis of ruin soil samples by wet sieving and Sedigraph with the Malvern Mastersizer 3000 laser device, applying clay-silt borders of 2 and 5 μm for laser results.

| Sample No. & Method | Coarse Sand % | Medium Sand % | Fine Sand % | Coarse Silt % | Medium Silt % | Fine Silt % | Coarse Clay % | Medium Clay % | Fine Clay % | Sand % | Silt % | Clay % | Very Fine Sand (125–63 μm) % |
|---|---|---|---|---|---|---|---|---|---|---|---|---|---|
| | | | | | | Sedigraph | | | | | | | |
| HH-WW R1 10 (Sedigraph) | 21 | 4 | 8 | 29 | 18 | 15 | 13 | 9 | 2 | **42** | **45** | **13** | *n.a.* |
| HH-WW-R1 20 (Sedigraph) | 16 | 5 | 11 | 30 | 12 | 16 | 13 | 8 | 3 | **45** | **42** | **13** | *n.a.* |
| JF site 124/1 5 cm (Sedigraph) | 3 | 3 | 17 | 30 | 18 | 11 | 8 | 6 | 4 | **51** | **36** | **13** | *n.a.* |
| JF site 124/1 15 cm (Sedigraph) | 5 | 3 | 16 | 34 | 19 | 9 | 7 | 6 | 3 | **53** | **35** | **12** | *n.a.* |
| JF site 124/1 25 cm (Sedigraph) | 21 | 4 | 14 | 39 | 15 | 8 | 6 | 6 | 5 | **57** | **29** | **14** | *n.a.* |
| | | | | | | Laser, Clay Border 2 μm | | | | | | | |
| HH-WW R1 10 (Laser 2 μm) | 0 | 6 | 25 | 28 | 18 | 15 | 8 | 1 | 0 | **31** | **60** | **9** | *19* |
| HH-WW-R1 20 (Laser 2 μm) | 1 | 8 | 26 | 26 | 15 | 14 | 9 | 1 | 0 | **35** | **56** | **9** | *19* |
| JF Site 124/1 5cm (Laser 2 μm) | 0 | 16 | 24 | 24 | 16 | 13 | 6 | 0 | 0 | **40** | **53** | **7** | *14* |
| JF Site 124/1 15cm (Laser 2 μm) | 0 | 17 | 26 | 23 | 15 | 12 | 6 | 0 | 0 | **44** | **50** | **6** | *15* |
| JF Site 124/1 25cm (Laser 2 μm) | 0 | 16 | 27 | 22 | 15 | 12 | 6 | 1 | 0 | **43** | **50** | **7** | *15* |
| | | | | | | Laser, Clay Border 5 μm | | | | | | | |
| HH-WW R1 10 (Laser 5 μm) | 0 | 6 | 25 | 28 | n.a. | 21 | 20 | 1 | 0 | **31** | **49** | **20** | *19* |
| HH-WW-R1 20 (Laser 5 μm) | 1 | 8 | 26 | 26 | n.a. | 19 | 19 | 1 | 0 | **35** | **45** | **20** | *19* |
| JF Site 124/1 5cm (Laser 5 μm) | 1 | 16 | 24 | 24 | n.a. | 19 | 16 | 0 | 0 | **41** | **42** | **17** | *14* |
| JF Site 124/1 15cm (Laser 5 μm) | 0 | 17 | 26 | 23 | n.a. | 18 | 15 | 0 | 0 | **43** | **41** | **16** | *15* |
| JF Site 124/1 25cm (Laser 5 μm) | 0 | 16 | 27 | 22 | n.a. | 18 | 16 | 1 | 0 | **43** | **41** | **16** | *15* |

## Appendix F. Detailed Sample List for EMMAgeo

**Table A10.** Numbers, sample names, and rough classification of the sample types that were used for end-member modeling of grain sizes with EMMAgeo, as well as for principal component analysis (PCA) based on grain sizes and magnetic susceptibilities. The sample numbers match those of the robust end member scores presented in Figure 9. "Soil" summarizes ruin soils, paleosols, loessial aprons in the Negev, and the natural plateau soil in the Petra region. "References" geological units and "dust storms" comprise the samples collected in dust collectors. These types were utilized for the non-parametric random forest approach to test whether grain size distributions and magnetic susceptibilities are suited for predictions of the sample type. Background colors refer to sample types: yellow for Negev soils, orange for Negev references, red for Negev dust storms, brown for Petra region soils, blue for Petra region reference samples, and gray for Petra region dust storm sediments.

| No. | Sample Name | Type | No. | Sample Name | Type | No. | Sample Name | Type |
|---|---|---|---|---|---|---|---|---|
| 1 | HH-WW R1 10 | Negev soil | 20 | dust storm 18.04.12 elevated floor | Negev dust storm | 39 | Sandplateau 3 | Petra soil |
| 2 | HH-WW-R1 20 | Negev soil | 21 | dust storm 18.04.12 table | Negev dust storm | 40 | Sandplateau 4 | Petra soil |
| 3 | HH-CW-Ruin 10 | Negev soil | 22 | dust storm 20.12.12 | Negev dust storm | 41 | Sandplateau Stein | Petra reference |
| 4 | HH-CW-Ruin 25 | Negev soil | 23 | dust storm 22.03.13 | Negev dust storm | 42 | Baja Sandstein | Petra reference |
| 5 | HH-CW-Ruin 50 | Negev soil | 24 | dust storm 10/11.02.15 | Negev dust storm | 43 | Disi Sandstone | Petra reference |
| 6 | HH-CW-R-60 | Negev soil | 25 | dust storm 05_01_18 | Negev dust storm | 44 | Um Ishrin Sandstone | Petra reference |
| 7 | HH-CW-Ruin 75 | Negev soil | 26 | dust storm 28_03_18 | Negev dust storm | 45 | Jh limestone 60 outcrop | Petra reference |
| 8 | HH-CW-R 90 | Negev soil | 27 | FJHP Site 1, Trench R | Petra soil | 46 | Beidha Fan | Petra reference |
| 9 | NH-LA-10cm | Negev soil | 28 | JH 1 | Petra soil | 47 | Fan Umm Sayhoun | Petra reference |
| 10 | NH-LA-30cm | Negev soil | 29 | JF site 124/1 5 cm | Petra soil | 48 | US 1 | Petra soil |
| 11 | HH-CW-Tur-Paleo | Negev reference | 30 | JF site 124/1 15 cm | Petra soil | 49 | Saleh 25-11-16 | Petra dust storm |
| 12 | HH-CW-chalk | Negev reference | 31 | JF site 124/1 25 cm | Petra soil | 50 | JH-19-12-16 | Petra dust storm |
| 13 | Haroa Farm - chalk | Negev reference | 32 | JF site 124/1 rock | Petra reference | 51 | JH-07-01-17 | Petra dust storm |
| 14 | HH-WW-C2-soft limestone | Negev reference | 33 | Umm Saysaban 5 cm | Petra soil | 52 | JH-15-02-17 | Petra dust storm |
| 15 | dust storm 24./25.03.03 front door | Negev dust storm | 34 | Umm Saysaban 10 cm | Petra soil | 53 | JH-01-03-17 | Petra dust storm |
| 16 | dust storm 24./25.03.03 incl washout dust | Negev dust storm | 35 | Abu Suwwan below nw 65 | Petra soil | 54 | Saleh 20-06-17 | Petra dust storm |
| 17 | dust storm 11.12.10 | Negev dust storm | 36 | Shkarat Msaied 1 | Petra soil | 55 | JH-05-08-17 | Petra dust storm |
| 18 | dust storm 12.12.10 | Negev dust storm | 37 | Sandplateau 1 | Petra soil | 56 | JH-05-08-17 closed box | Petra dust storm |
| 19 | dust storm 29.02.12 | Negev dust storm | 38 | Sandplateau 2 | Petra soil | 57 | Saleh 05-08-17 | Petra dust storm |

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
