# Peer review of "Character, Rates, and Environmental Significance of Holocene Dust Accumulation in Archaeological Hilltop Ruins in the Southern Levant"

_geosciences, doi:10.3390/geosciences9040190_

Round 1

Reviewer 1 Report

This is very good contribution to understanding the role of ruins in trapping the dust. Paper is soundly written and supported by many detailed analysis. However, paper is too long in this form and needs significant shortening. I made some suggestions in edited manuscript. Other possible way of reducing the length of paper and making it better is to replace several pages of sediment descriptions (grain size analyses, tables, etc.) with two generalized profiles of ruin soils, one from Negev and other from Petra, side by side. Then, on this same figure explain or point to similar components within the two profiles (eg. fine silt content from distant source(s), and distinguish these from sediments derived from local sediment sources.

Author Response

Dear reviewers, dear editors,

thank you very much for the detailed revision of our manuscript. We followed all your suggestion.

The references and sections marked in the pdf were checked and corrected. We replaced the figures showing statistics results with pictures with larger letters, and edited the PCA biplot by hand in order to make the sample names better legible. As well, we improved the respective captions and hope the figures are now more clear. However, there is a limit of what can be done with regard to the letter size of the PCA biplot since too large letters might shift the position too much, and the similarity of samples is expressed by their relative position.

We consider the statistical results essential for the paper and would like to include these respective figures. Their legibility depends also on the final editing of the article and we hope the editors can find a solution to include them in sufficiently large size, or make them available as additional high-resolution download.

The references are now numbered and presented according to the geosciences template. We did our best to follow the examples there.

#Reviewer 1: in order to shorten the paper, we summarized the results of particle sizes in only one table showing only results of selected samples from both investigation regions. The three tables showing full results of particle sizes were moved to appendix C. Similarly, we shortened the table of pedogenesis results to those of the selected samples and moved the full results to appendix C. As well, part of the text on the results of pedogenesis and statistical evaluation were either moved to appendix C or deleted.

This shortened the results text significantly, but we would like to keep all figures in as figures usually allow readers to quickly grasp the essence of results, and make articles better legible. We hope that the editorial team will manage to place them with good resolution in a way that will permit a swift flow of reading without occupying much page space.

We trust that the presentation and download option of the appendix section will be handled by the editorial team in a way that permits to keep the main paper short, but make the appendix easily accessible. When preparing the additional data for the appendix, we followed the instructions for authors and some earlier communication with the editorial team, and assume that the current presentation of the material fits the journal's procedures.

We would be glad if the paper was now ready for publication, thanks very much,

Bernhard Lucke on behalf of all authors

Reviewer 2 Report

Very interesting and carefully prepared study on Holocene dust/loess accumulation in in Archaeological Hilltop Ruins in the Southern Levant. Authors provide many data and discuss very detailed this problem. 

However, there are some misunderstandings in the text which should be corrected.

Average sedimentation rate calculated in g/m2/year (Table 4) is 10 times underestimated. 0,1 mm/year gives 100 000 mm3/m2/year. It is 100 cm3 because 1 cm3 = 1000 mm3.

Check it and take it into account in further part of the manuscript.

Crouvi et al 2008 do not eliminate the fraction of medium silt in the laser measurements but only the name of "medium silt". They use so called 20-μm grain ratio [defined as (20–200 μm %)/(5–20 μm %)] as a proxy for the different transport mechanisms and the dust-source variability in the loess sequences.

References are not properly sorted (Kronberg & Nesbitt, 1981).

Author Response

Dear reviewers, dear editors,

thank you very much for the detailed revision of our manuscript. We followed all your suggestion.

The references and sections marked in the pdf were checked and corrected. We replaced the figures showing statistics results with pictures with larger letters, and edited the PCA biplot by hand in order to make the sample names better legible. As well, we improved the respective captions and hope the figures are now more clear. However, there is a limit of what can be done with regard to the letter size of the PCA biplot since too large letters might shift the position too much, and the similarity of samples is expressed by their relative position.

We consider the statistical results essential for the paper and would like to include these respective figures. Their legibility depends also on the final editing of the article and we hope the editors can find a solution to include them in sufficiently large size, or make them available as additional high-resolution download.

The references are now numbered and presented according to the geosciences template. We did our best to follow the examples there.

#Reviewer 2: we made indeed a calculation mistake when transforming current dust deposition rates from cm² to m² by the mentioned factor of 10 – thanks very much for your attention on that! Values and the respective discussion were corrected accordingly.

We revised the citation of Crouvi et al. (2008) and described our procedure more precisely, and sorted the references now in the order of appearance in the text as outlined in the geoscience template/instructions for authors.

We would be glad if the paper was now ready for publication, thanks very much,

Bernhard Lucke on behalf of all authors